# Dissecting gene regulatory networks governing human cortical cell fate

Jingwen W. Ding[1,2], Chang N. Kim[3,4], Megan S. Ostrowski[1,2], Yashodara Abeykoon[1,2], Bryan J. Pavlovic[1,2], Jenelle L. Wallace[1,2], Nathan K. Schaefer[1,2], Tomasz J. Nowakowski[1,3,4,5,6] & Alex A. Pollen[1,2,6 ✉]

Human cortical neurogenesis involves conserved and specialized developmental processes during a restricted window of prenatal development. Radial glia (RG) neural stem cells shape cortical cell diversity by giving rise to excitatory neurons, oligodendrocytes and astrocytes, as well as olfactory bulb interneurons (INs) and a recently characterized population of cortical INs[1,2]. Complex genetic programs orchestrated by transcription factor (TF) circuits govern the balance between self-renewal and differentiation, and between different cell fates[3–8]. Despite progress in measuring gene regulatory network activity during human cortical development[9–12], functional studies are required to evaluate the roles of TFs and effector genes in human RG lineage progression. Here we establish a human primary culture system that allows sensitive discrimination of cell fate dynamics and apply single-cell CRISPR interference (CRISPRi) screening[13,14] to examine the transcriptional and cell fate consequences of 44 TFs active during cortical neurogenesis. We identified several TFs with new roles in cortical neurogenesis, including *ZNF219*−previously uncharacterized−that represses neural differentiation and *NR2E1* and *ARX* that have opposing roles in regulating RG lineage plasticity and progression across developmental stages. We also detected convergent effector genes downstream of multiple TFs enriched in neurodevelopmental and neuropsychiatric disorders and observed conserved mechanisms of RG lineage plasticity across primates. We further uncovered a post-mitotic role for *ARX* in safeguarding IN subtype specification through repressing *LMO1*. Our study provides a framework for dissecting regulatory networks driving cell fate consequences during human neurogenesis.

Human radial glia (RG) evolved an increased proliferative capacity, altered cell fate potential and protracted period of maturation, supporting the increased number and complexity of daughter cells[2,15,16]. Gene regulatory networks governing RG self-renewal, differentiation and maturation have long been implicated in cortical expansion[17,18]. RG sequentially produce distinct subtypes of neuron followed by glial cell types[19,20]. Although cortical inhibitory neurons (INs) are generated mainly in the ganglionic eminences and migrate to the cortex[21–23], recent lineage tracing studies[1,24–27] and developmental cell atlases[9,12] have demonstrated that human RG also produce cortical-like INs at late stages of neurogenesis that share transcriptional signatures with caudal ganglionic eminence (CGE) and lateral ganglionic eminence derived INs. However, the role of TFs in regulating RG differentiation decisions, lineage plasticity to produce INs and maturation remains largely unexplored.

Genetic perturbations combined with measurements of single-cell gene expression provide a powerful approach, termed Perturb-seq, for high-throughput dissection of gene function[13,14]. Cas9-based CRISPR loss-of-function perturbation approaches have been applied recently to study gene regulatory networks influencing cortical development using induced pluripotent stem (iPS)-cell-derived organoid[28–30] and mouse[31,32] models. However, Cas9-induced double-stranded DNA breaks can cause cytotoxicity, influencing proliferation and differentiation decisions[33,34], whereas acquired genetic and epigenetic variation in source iPS cells[35,36], and patterning biases[37] and cell stress[38] during differentiation, can influence cell-type fidelity and fate specification in organoid models. CRISPR interference (CRISPRi) screens using dCas9–KRAB enable efficient and uniform repression of target genes with limited cellular toxicity[39,40], whereas primary cortical culture[41,42] captures physiological aspects of human cortical neurogenesis, supporting sensitive detection of cell fate choices.

Here we established a primary cell model system that recapitulates in vivo differentiation dynamics and performed Perturb-seq to measure the transcriptional and cell fate consequences of repressing 44 TFs that are expressed robustly in the human cortical RG lineage. Extending human primary cell culture approaches[1,42,43], we targeted TFs in a homogeneous RG population and then removed growth factors to permit

[1]The Eli and Edythe Broad Center of Regeneration Medicine and Stem Cell Research, University of California San Francisco, San Francisco, CA, USA. [2]Department of Neurology, University of California San Francisco, San Francisco, CA, USA. [3]Department of Neurological Surgery, University of California San Francisco, San Francisco, CA, USA. [4]Department of Anatomy, University of California San Francisco, San Francisco, CA, USA. [5]Department of Psychiatry and Behavioral Sciences, University of California San Francisco, San Francisco, CA, USA. [6]Weill Institute for Neurosciences, University of California San Francisco, San Francisco, CA, USA. ✉e-mail: alex.pollen@ucsf.edu

cortical neurogenesis, cell fate choice and early subtype specification. Our screening revealed the role of *ZNF219*—not previously described in cortical development—in repressing neuronal differentiation, opposing roles for *NR2E1* and *ARX* in regulating the balance of human excitatory versus inhibitory neurogenesis, and the role of *ARX* in safeguarding IN subtype specification through transcriptional repression of downstream transcription cofactor *LMO1*. Intersecting dysregulated genes under different perturbations revealed candidate hub effector genes downstream of several TFs enriched for roles in neurodevelopmental and neuropsychiatric disorders. Coupling CRISPRi screening with barcoded lineage tracing demonstrated the potential to engineer lineage plasticity and developmental tempo of individual RG through TF perturbation. Together with parallel screening in rhesus macaque, our data illuminate conserved mechanisms governing cortical RG lineage progression across primates.

## Systematic TF repression during human corticogenesis

We designed a primary cell model of neurogenesis and lineage progression to evaluate the impacts of TF repression on cell fate choice during human cortical development. To direct gene targeting to RG at the start of differentiation, we first enriched for RG isolated from primary human tissue samples by adding epidermal growth factor (EGF) and fibroblast growth factor 2 for 5 days before infecting with an all-in-one CRISPRi lentivirus (Fig. 1a), and we further expanded RG for 1 week allowing for target gene knockdown (KD) before differentiation[39] (Extended Data Fig. 1a–c). We then replaced growth factors with brain-derived neurotrophic factor (BDNF) to support spontaneous differentiation of perturbed RG. Consistent with recent studies[1,42,43], this model recapitulates in vivo RG lineage progression and generation of excitatory neurons (ENs) and INs (Fig. 1a and Extended Data Fig. 1a,b), enabling detection of perturbations that affect differentiation dynamics and cell fate choice. Repression of *PAX6* and *EOMES* recapitulated the effects of these TFs described in other model systems in promoting excitatory neurogenesis, highlighting the potential of our system to uncover additional regulators[44–47] (Extended Data Fig. 1d).

To systematically identify regulators of human cortical neurogenesis, we first prioritized TFs using single-cell RNA sequencing (scRNA-seq) and single-cell assay for transposase-accessible chromatin sequencing (scATAC-seq) multiome data[10,12]. We selected 44 TFs based on robust expression, motif accessibility, gene regulatory network size, target gene expression and predicted transcriptional consequences in the RG lineage (Supplementary Table 1; Methods). We then adapted an all-in-one CRISPRi vector co-expressing green fluorescent protein (GFP)[48] by inserting a capture sequence in the single guide RNA (sgRNA) scaffold, enabling direct capture of sgRNAs during scRNA-seq using a 10x Genomics platform[49], thereby supporting screening in primary cell culture systems[50]. We synthesized and cloned a library containing 164 sgRNAs, targeting the 44 TFs, and included 20 non-targeting control (NT) sgRNAs. For each TF, we targeted active promoters with accessible chromatin during human cortical neurogenesis[9] using three sgRNAs per promoter[51,52] (Supplementary Table 2). We derived primary human cultures from cryopreserved cortical tissue of four individuals at stages of peak neurogenesis from gestational weeks (GW) 16–18 (ref. 41). We delivered the CRISPRi library by lentivirus into primary RG, targeting less than 30% infection rate to generate libraries with mostly singly infected cells (Extended Data Fig. 1a and Supplementary Table 3), removed growth factors, and confirmed efficient gene repression before and throughout differentiation (Extended Data Fig. 1c,e).

scRNA-seq confirmed a 95% population of cycling RG on day 0 of differentiation, marked by co-expression of RG marker *HOPX* and proliferation marker *MKI67* (Fig. 1b–f). By day 7, three principal cortical cell class trajectories emerged—EN, IN and oligodendrocyte lineages marked by *NEUROD2*, *DLX2* and *BCAS1*, respectively (Fig. 1c–e)—in addition to a continuum from RG to astrocytes marked by *AQP4* (Extended

Data Fig. 1f). Integrating both experimental timepoints highlighted homogeneity of day 0 populations with minimal spontaneous differentiation (Fig. 1c and Extended Data Fig. 1g), as well as comparable representation of cells derived from independent technical and biological replicates across cell types with minimal batch effects (Fig. 1b,d and Extended Data Fig. 1h).

Pearson correlation and reference mapping to a developmental cortical cell atlas[12] confirmed the recapitulation of in vivo-like gene expression, cell types, states, differentiation dynamics and cell fate choice in the primary culture differentiation system (Extended Data Fig. 1i,j). Most neurons exhibited immature states (Extended Data Fig. 1j), consistent with recent generation from RG after differentiation. These differentiation dynamics were also recovered by RNA velocity and pseudotime analysis (Fig. 1g and Extended Data Fig. 1l,m). Notably, comparison with iPS-cell-based models revealed a near twofold increase in the Pearson correlation coefficients of marker gene expression compared with in vivo cell types in the primary two-dimensional (2D) system, supporting improved fidelity to normal development (Extended Data Fig. 1i,k). In addition, primary culture in 2D and organotypic slice both showed reduced transcriptional signatures of cellular stress across principle dimensions, including glycolysis, endoplasmic reticulum stress, oxidative stress and apoptosis[53–55] (Extended Data Fig. 1n), supporting the physiological relevance of the primary cell models.

We assigned sgRNAs to 118,456 cells (Methods) across timepoints, with a mean of 200 singly infected cells per sgRNA on differentiation day 7 (Fig. 1h and Extended Data Fig. 1o). KD efficiency was calculated in each cell class and 18 sgRNAs with less than 25% KD (log$_2$ fold change (FC) > −0.4) were removed from downstream analyses (Fig. 1i and Supplementary Table 4; Methods). Remaining active sgRNAs exhibited a median KD efficiency of 72%, with comparable transcriptional responses to independent sgRNAs targeting the same promoter (Fig. 1i and Extended Data Fig. 2). For further analysis, we collapsed all active sgRNAs sharing the same target TF, which yielded a median KD efficiency of 80% and a mean of 600 cells per gene (Extended Data Fig. 1o,p).

## Transcriptional regulators of human corticogenesis

We next examined the consequences of TF repression on gene expression and cell-type composition by differentiation day 7 (Fig. 2a and Supplementary Table 5; Methods). Gene expression changes were correlated between individual sgRNAs and genes (average Pearson $R = 0.88$) and cell-type abundance was preserved upon downsampling of cell number (average Pearson $R = 0.94$), supporting the power of the screen to detect changes in both modalities (Extended Data Fig. 3a,b).

We observed a positive correlation between the effects of individual TFs on cell-type composition and on gene expression across cell types (Fig. 2b) and within cell classes (Extended Data Fig. 3c). Comparing the extent of changes in both dimensions prioritized TFs whose depletion caused the strongest phenotypes: *NR2E1*, *ARX*, *ZNF219*, *SOX2*, *SOX9*, *CTCF*, *NEUROD2* and *PHF21A* (Fig. 2b). These TFs also showed the strongest impact on other molecular phenotypes, including Euclidean distance and energy distance[56] to NT, which compare the average expression between groups and the distance between and within groups, respectively, as well as maximum composition change among all cell types in the RG lineage (Extended Data Fig. 3d). Notably, the TFs with strong cellular and transcriptional phenotypes have been implicated previously in neurological disorders: *ARX* in X-linked lissencephaly[57], epilepsy[58,59] and intellectual disability[60,61], and autism spectrum disorder (ASD)[58]; *NR2E1* in schizophrenia (SCZ)[62]; *SOX2* in intellectual disability and epilepsy[63,64]; *CTCF* in intellectual disability with microcephaly[65,66]; *NEUROD2* in intellectual disability[67] and early infantile epileptic encephalopathy[68]; *PHF21A* in intellectual disability with epilepsy and ASD[69] and *ZNF219* in a case of low IQ ASD[70]. Although we cannot rule out that additional TFs may impact differentiation or

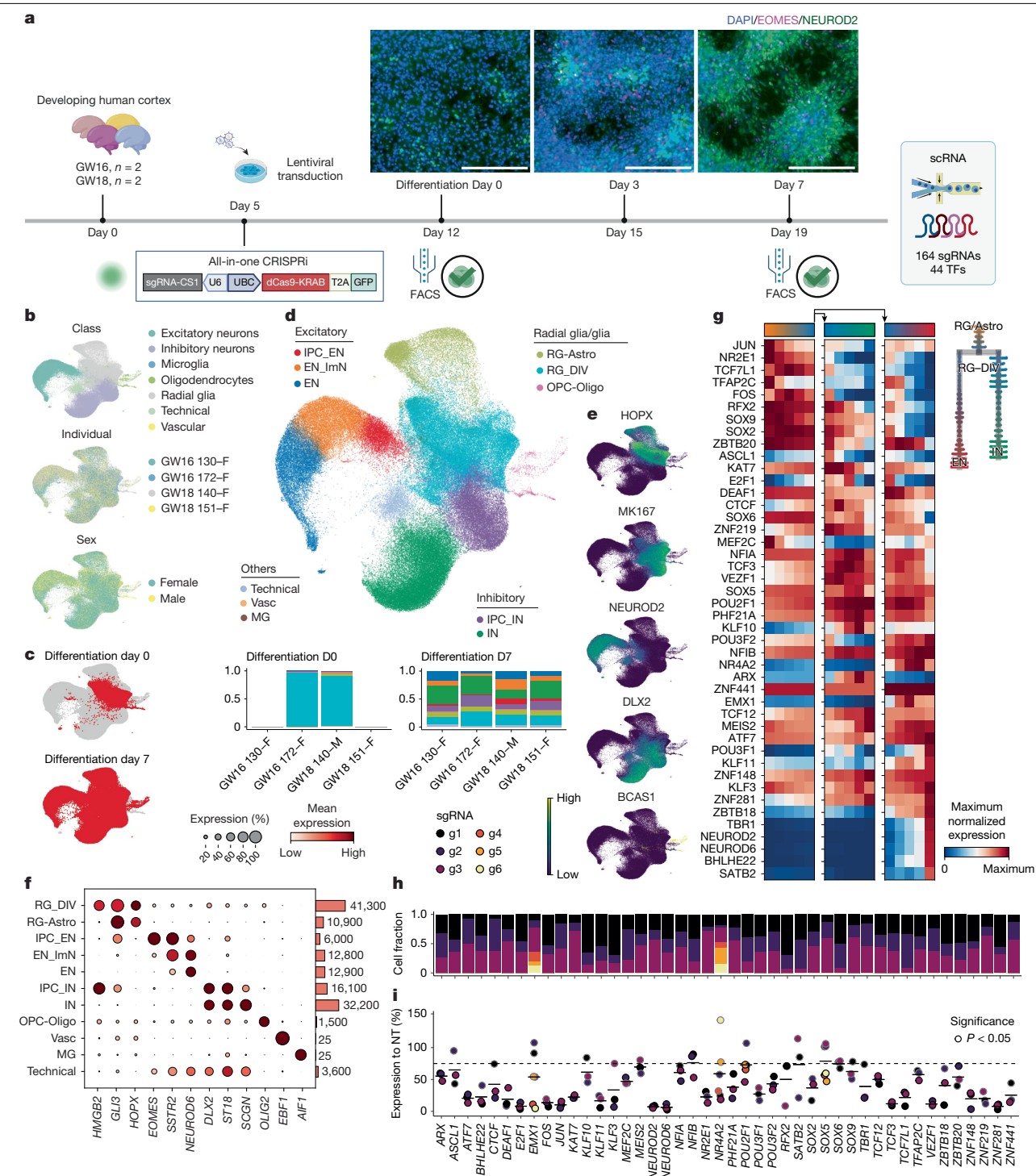

**Fig. 1 | Single-cell Perturb-seq on 44 TFs in primary human neuronal differentiation. a**, Experimental design for high-throughput perturbation of 44 TFs in primary human cortical progenitors from four individuals, with immunocytochemical labelling of *EOMES* and *NEUROD2* in in vitro 2D culture of human cortical RG before (day 0) and after (day 3, day 7) induced differentiation. Representative images from four individuals are shown. FACS, fluorescence-activated cell sorting. **b**, Uniform manifold approximation and projections (UMAPs) of cells collected on day 0 (21,151 cells, *n* = 2 individuals) and day 7 (116,166 cells, *n* = 4 individuals), coloured by cell class, individual and sex. **c**, UMAPs highlighting cells from different timepoints. **d**, UMAP coloured by supervised cell type, with stacked barplots (bottom) showing cell-type distributions across individual at each timepoint. **e**, UMAPs showing expression of *HOPX, MKI67, NEUROD2, DLX2* and *BCAS1*. **f**, Left, dotplot of marker gene expression across annotated clusters; right, barplot of cell numbers. Dot size denotes the expressing-cell fraction and colour denotes mean expression.

**g**, Heatmap of normalized expression of 44 TFs targeted in this study along pseudotime with branches for IN and EN (left) lineages, with a tree plot showing cell-type distribution along pseudotime (right). Pseudotime was calculated using NT cells on day 7. **h**, Stacked barplot showing sgRNA distributions per TF target on day 7, coloured by individual sgRNA. **i**, Dotplot showing target gene KD efficiency compared with NT cells on day 7, filled by individual sgRNA with borders highlighting significance. log$_2$FC values were calculated with DEseq2 in each cell class in *n* = 4 individuals and the lowest value for each sgRNA was used for visualization and filtering. sgRNAs with less than 25% reduction were removed from downstream analyses; remaining active sgRNAs showed a median 72% KD (28% residual expression). Astro, astrocytes; DIV, dividing; ImN, immature neuron; MG, microglia; OPC, oligodendrocyte progenitor cell; Vasc, vascular. Scale bars, 200 μm. Panel **a** created in BioRender. Nowakowski, T. (https://BioRender.com/7cr6o12).

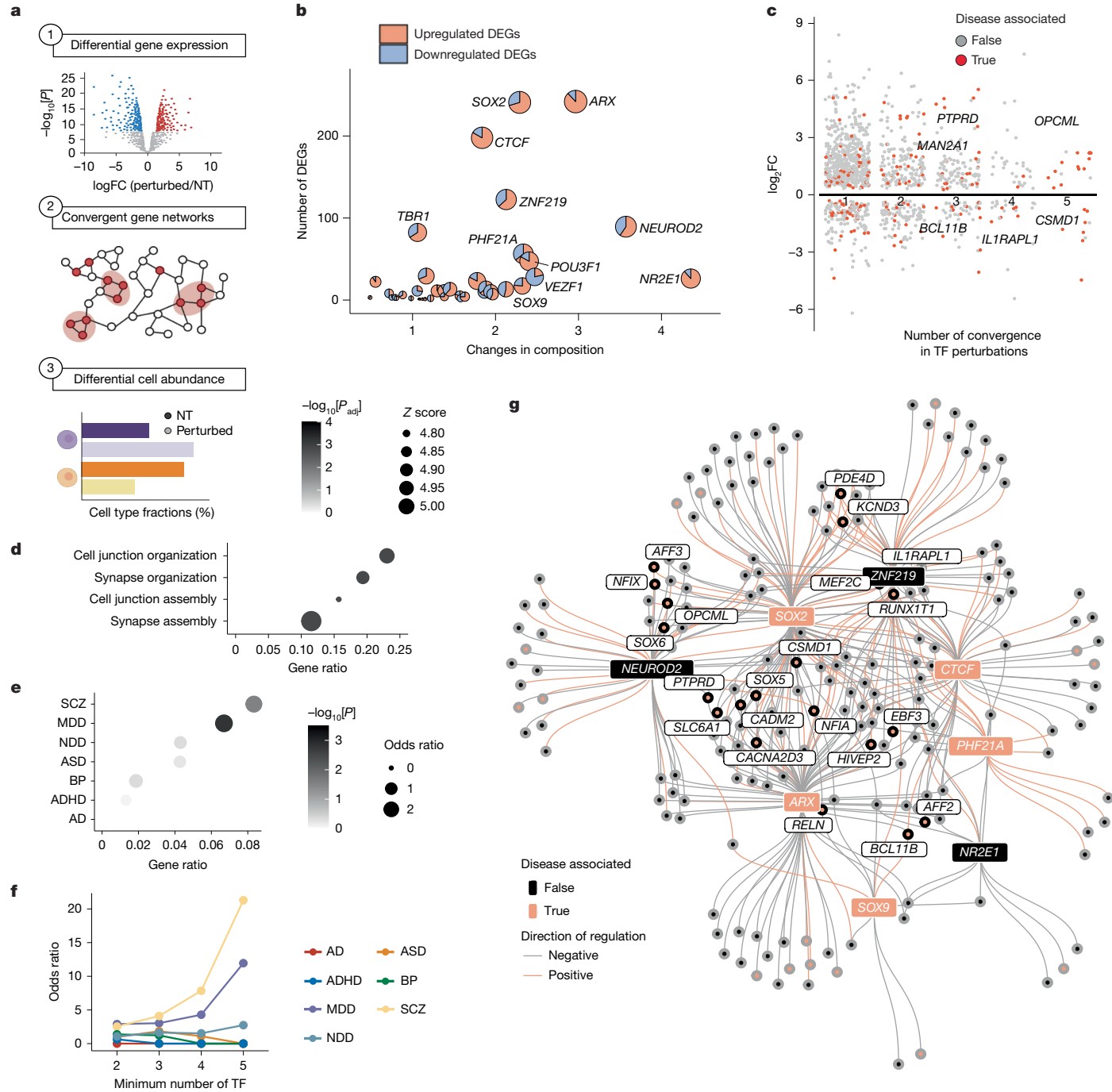

**Fig. 2 | Convergent transcriptional changes reveal neuropsychiatric-disorder-enriched hub effector genes. a**, Schematics showing analytical workflow applied to the day 7 Perturb-seq dataset obtained from Fig. 1. **b**, Scatterplot for summed |estimated coefficient| in cell abundance (*x* axis) and number of DEGs (*y* axis) across seven main cell types in the EN and IN lineage for each TF perturbation. Dot size reflects perturbations with the strongest phenotypes calculated by (summed |estimated coefficient| × total number of DEGs), with embedded pie charts indicating fractions of up- and downregulated DEGs. **c**, Volcano plot of DEGs shared across different numbers of perturbations. DEGs with |log$_2$FC| > 0.2 are shown. DEGs that overlap with seven sets of neuropsychiatric disorder-associated gene sets are coloured red and SCZ-associated genes are labelled. **d**, Dotplot showing gene ontology biological processes enrichment among convergent DEGs that were detected in at least two TF perturbations, using non-convergent DEGs as background.

Dot size denotes *Z* score; colour denotes −log$_{10}$[adjusted *P*]. **e**, Dotplot showing enrichment of seven disease-associated gene sets (Methods) among convergent DEGs versus non-convergent DEGs. One-sided Fisher's exact test; dot size denotes odds ratio; colour denotes −log$_{10}$[*P*]. **f**, Line plot showing odds ratio of disease-associated gene enrichment in DEGs shared across the minimum number of perturbations indicated. **g**, Regulatory network plot showing eight prioritized TFs (*NR2E1*, *ARX*, *SOX2*, *ZNF219*, *NEUROD2*, *PHF21A*, *SOX9* and *CTCF*) and connected convergent DEGs. Edge colour indicates direction of regulation (pink, positive regulation (where DEGs were downregulated upon TF perturbation)). Pink nodes or labels denote disease-associated genes or TFs. AD, Alzheimer's disease; ADHD, attention-deficit/hyperactivity disorder; BP, bipolar disorder; NDD, neurodevelopmental disorder. Panel **a** created in BioRender. Nowakowski, T. (https://BioRender.com/7cr6o12).

maturation phenotypes not captured in this assay due to incomplete KD, redundancy or stage-specific roles, these results highlight the functional importance and disease relevance of TFs prioritized by screening.

At the gene expression level, the predominance of upregulated differentially expressed genes (DEGs) following *NR2E1* (88%) and *ARX* (85%) KD was consistent with their role as transcriptional repressors[71,72] (Fig. 2b). Approximately 25% of DEGs were affected by the perturbation of more than one TF (Fig. 2c). These convergent DEGs showed enrichment for cell adhesion- and synaptic development-related terms, suggesting that effector genes regulating neurogenesis, migration and maturation are highly interconnected across TF regulatory networks (Fig. 2d). Moreover, intersecting these convergent DEGs with seven sets of neurodevelopmental and neuropsychiatric disorder-related genes revealed significant overlaps with SCZ and major depressive disorder (MDD)-associated genes, in comparison with non-convergent DEGs (one-sided Fisher's exact test, *P* < 0.05) (Fig. 2e,f). This overlap includes *PTPRD*, modulated by *ARX*, *SOX2* and *NEUROD2*; and *IL1RAPL1*, modulated by *SOX2*, *ZNF219*, *CTCF* and *TBR1* (Fig. 2c,g). Both *PTPRD* and *IL1RAPL1* are associated with SCZ and depletion of both genes have been shown to induce aberrant neurogenesis, maturation and behaviours in mouse models[73–75]. This finding highlights strong connections among disease-related genes in TF-driven regulatory networks, indicating their potential roles as hub effector genes downstream of developmental TFs.

## Distinct fate outcomes of RG upon TF perturbations

We further investigated the impact of TF perturbations on cell-type composition using cluster-free approaches[76] to map cell fate changes with higher resolution along developmental trajectories (Fig. 3a). These approaches yielded consistent results with cluster-aware methods at the gene and individual sgRNA level and were robust to downsampling (Extended Data Figs. 3e,f and 4).

Different perturbations elicited a range of composition consequences in differentiated cell types (Fig. 3a and Extended Data Fig. 3e,f). These phenotypes included examples consistent with mechanistic studies. For example, repression of SOX2 resulted in accumulation of RG[77], whereas repression of the proneural factor *NEUROD2* enriched for progenitors and immature ENs at the expense of mature ENs (Fig. 3a and Extended Data Fig. 3e). However, screening also revealed additional roles for previously characterized genes implicated in disease. For example, *NR2E1* knockout was previously shown to drive premature neural differentiation in mouse models[78,79], but repression in human RG specifically enriched for post-mitotic INs, in addition to depleting RG (Fig. 3a and Extended Data Fig. 3e). Similarly, *ARX* knockout was previously shown to affect IN differentiation and migration in mouse models[80–82], but repression in human RG specifically enriched for the EN lineage at the expense of INs.

Our screen also uncovered disease-linked genes with composition phenotypes, including *ZNF219* and *PHF21A*, that to our knowledge have yet to be examined functionally during cortical neurogenesis. Repression of *ZNF219* mirrored the effects of repressing *SOX9*—a known ZNF219 interaction partner in chondrocytes[83]—by increasing the proportion of ENs, but showed different effects from *SOX9* in dividing RG and INs (Fig. 3a and Extended Data Fig. 3e). Repression of *PHF21A*—a member of the BRAF/HDAC complex implicated in synapse formation[84]—led to an overall trend towards EN over IN differentiation (Fig. 3a and Extended Data Fig. 3e,f), matching the effects of targeting other ASD causal genes *ARX* and *CTCF*[85,86].

Flow cytometry analysis of the top TF candidates, *ARX*, *NR2E1* and *ZNF219*, with finer temporal resolution validated the cellular phenotypes observed in our Perturb-seq (Extended Data Fig. 5 and Supplementary Table 6), and further revealed early manifestations of compositional changes in KI67⁺ and EOMES⁺ progenitors on differentiation day 4, confirming that these perturbations affect self-renewal and cell fate decisions at the stage of neurogenesis instead of neuronal maturation (Extended Data Fig. 5d).

## TF perturbations alter EN to IN output of individual RG

The influence of multiple TFs on the relative abundance of ENs and INs nominated RG lineage plasticity as a candidate developmental mechanism for observed composition differences. Composition differences could arise from perturbations of RG that preferentially affect EN-biased clones, mixed clones containing both ENs and INs, or IN-biased clones (Fig. 3b). To distinguish between these possibilities, we combined Perturb-seq with lineage tracing using STICR—a GFP-expressing lineage tracing library with static barcodes that can be measured by scRNA-seq[1]. We constructed an mCherry-expressing sub-library targeting *NR2E1*, *ARX* and *ZNF219*, and also included *SOX2* and *NEUROD2* as controls, the former known to impact both lineages and the latter known to promote the EN lineage (Fig. 3b).

As RG fate plasticity changes with maturation, we extended the experiments using this targeted dual library to primary human RG culture isolated from nine human individuals from GW16 to GW22, spanning the peak of excitatory and inhibitory neurogenesis[12,27] to early stages of gliogenesis. Cells co-expressing mCherry and GFP were isolated on day 7 for scRNA-seq, and transcriptomes were reference mapped to the initial Perturb-seq data for cell-type annotation. This analysis yielded 129,003 cells with assigned sgRNAs and lineage barcodes (Fig. 3c and Extended Data Fig. 6a–j; Methods) and recapitulated the gene expression and cell-type composition consequences observed in the initial screen (Extended Data Fig. 6k,l).

We next investigated the landscape of clonal lineage relationships. We considered multicellular clones containing at least three cells with consistent sgRNA assignment, recovering 8,937 clones (Extended Data Fig. 7a; Methods). One person with low overall cellular coverage was dropped from this analysis. Remaining clones contained a mean of five cells in NT conditions (Extended Data Fig. 7b). The most abundant multi-class clones contained RG and ENs (35%) and RG and INs (20%) but, consistent with recent studies, we also observed around 15% clones containing both ENs and INs (Fig. 3d). Lineage coupling, defined as the normalized barcode covariance between cell types[87,88], further supported the presence of mixed clones with a comparable linkage of dividing RG between both excitatory and inhibitory intermediate progenitor cells (IPCs) and immature neurons (Fig. 3e). Unsupervised clustering of clones based on cell-type composition using scLiTr[89] (Methods) revealed five categories of fate biases, comprising RG, EN, IN, oligodendrocytes and EN/IN dual fate biased clonal clusters (Fig. 3f and Extended Data Fig. 7c). A small percentage of INs were observed in oligodendrocyte-biased clones and vice versa, suggesting their close lineage relationship[12,27,90].

Intersecting perturbation and lineage tracing information revealed fate alterations by different TF perturbations. *NR2E1* KD significantly enriched for IN-biased clones at the expense of RG- and EN- biased clones, while *ARX* KD strongly enriched for EN-biased clones at the expense of EN/IN dual fate biased clones (Fig. 3g and Extended Data Fig. 7d). A significant decrease in the fraction of EN/IN dual fate biased clones was also observed upon *ZNF219* KD, suggesting that it shapes the balance of EN/IN output by increasing the diversity of lineage output from individual RG. In each case, the alterations in RG lineage biases were consistent with the observed cell-type composition changes (Fig. 3a and Extended Data Fig. 6k), supporting lineage plasticity as the developmental mechanism underlying the effects of these TFs on EN and IN fates.

## TF perturbations alter RG lineage progression

RG undergo temporal fate restriction sequentially, producing distinct subtypes of neuron as well as glial cell types. Along the developmental axis, we observed a stage-dependent shift in RG fate biases marked by a

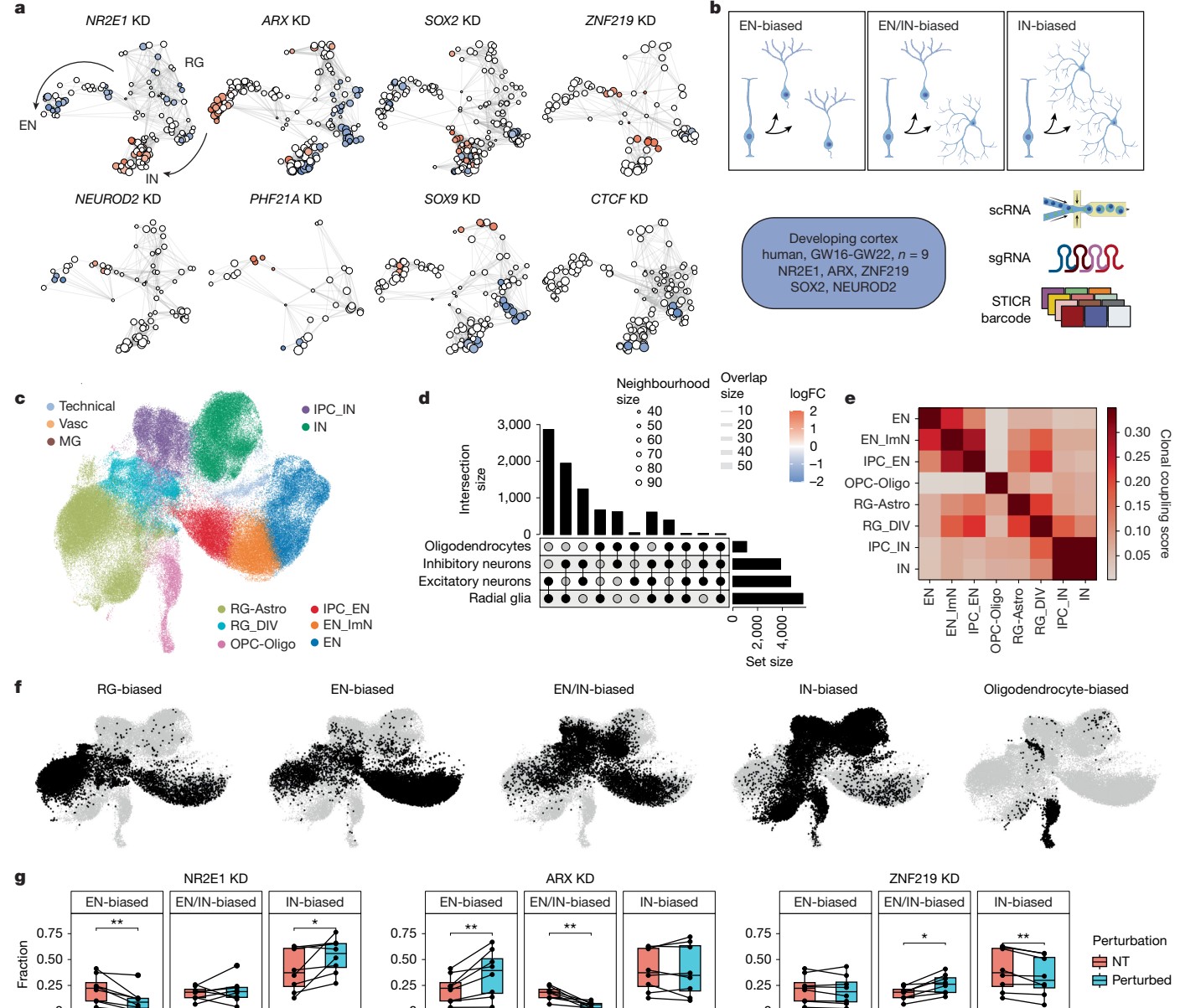

**Fig. 3 | TF perturbations alter EN/IN output of human cortical RG.**
**a**, Neighbourhood graphs based on the UMAP in Fig. 1d showing results of differential abundance testing using Milo under perturbation of eight prioritized TFs at the gene level. NT cells were downsampled to match perturbed cell numbers. Node sizes denote cell numbers in each neighbourhood; edges depict the number of cells shared between neighbourhoods; node colour denotes $\log_2$FC (false discovery rate (FDR) = 0.1) when perturbed. **b**, Top, schematics illustrating RG of distinct fate biases; bottom, experimental design for combined TF perturbation and lineage tracing in cortical RG of nine human individuals. **c**, UMAP of lineage-resolved 129,003 human cells collected on differentiation day 7, coloured by supervised cell type. A median of 15,469 cells per TF perturbation were recovered after filtering. **d**, Upset plot of cell class

compositions in clones containing cells from more than one class (multi-class clones). **e**, Heatmap for lineage coupling score matrix in main cell types. **f**, UMAPs showing cell fate distributions in clonal clusters identified using scLiTr[89]: RG-, EN-, EN/IN-, IN- and oligodendrocyte-biased clusters. **g**, Box plots showing distributions of clonal cluster fractions in eight human individuals in *NR2E1, ARX* and *ZNF219* KD versus NT. Two-sided paired Wilcoxon tests; $P$ = 0.0078, 0.64, 0.016 (*NR2E1*); 0.0078, 0.0078, 1 (*ARX*) and 0.38, 0.023, 0.0078 (*ZNF219*). Centre line, median; box limits, upper and lower quartiles; whiskers, 1.5× interquartile range; points, outliers. \*$P$ < 0.05; \*\*$P$ < 0.01; \*\*\*$P$ < 0.001. Panel **b** created in BioRender. Nowakowski, T. (https://BioRender.com/7cr6o12).

decrease of EN-biased clones and increase of IN-clones around GW18–19 (refs. 12,27) (Fig. 4a), supported by increased lineage coupling between dividing RG and IN after GW19 (Extended Data Fig. 7e). At GW19, *ARX* KD delayed the timing of IN-biased cluster expansion, whereas *NR2E1* KD promoted this transition (Fig. 4a), suggesting stage-dependent effects of TF perturbations on lineage decisions. Furthermore, there seemed to be a critical time window for altering EN/IN fate before GW21–22 (Extended Data Fig. 7d). Pseudotemporal ordering based on clonal types recapitulated the sequence of temporal cell fate choices

beyond the EN/IN transition (Fig. 4b–e). In addition to increasing IN-biased clones at early stages, we found that *NR2E1* KD also increased oligodendrocyte-biased clones at later stages, nominating this TF as a regulator of developmental tempo (Fig. 4a,f and Extended Data Fig. 7d). Conversely, *ARX* KD shifted the clonal pseudotime distribution significantly to more immature stages. Together, analysing lineage progression across differentiation stages, revealed opposing effects of *NR2E1* and *ARX* on delaying and accelerating RG lineage progression, respectively (Fig. 4f).

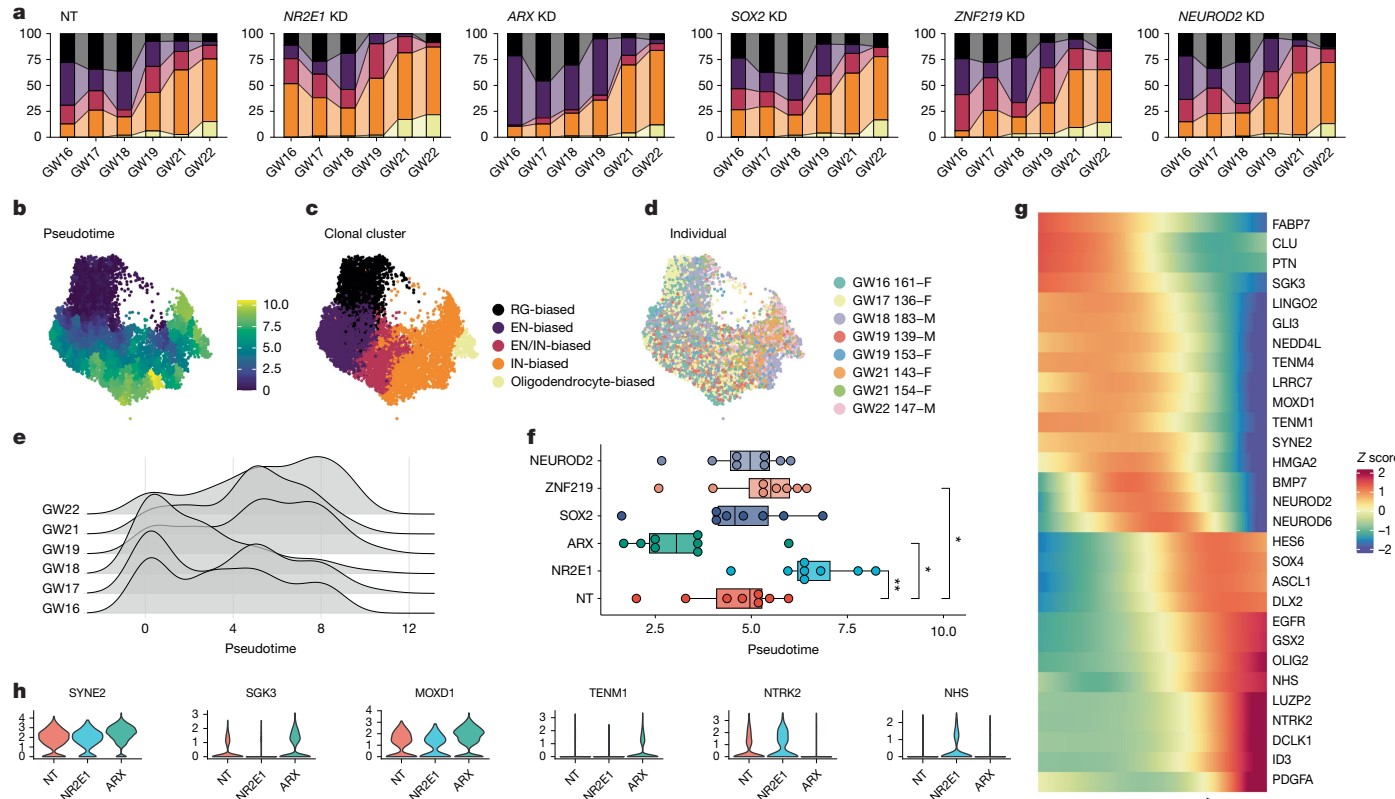

**Fig. 4 | Heterochronic RG lineage progression under TF perturbations.**
**a**, Stacked barplot showing distributions of clonal cluster fractions along human developmental stages in NT, *NR2E1*, *ARX*, *SOX2*, *ZNF219* and *NEUROD2* KD. Legend is shown in **c**. **b**–**d**, UMAPs of human clones coloured by pseudotime (**b**), fate biases (**c**) and individuals (**d**). **e**, Ridge plots of pseudotime distribution in clones grouped by stages. **f**, Boxplot showing median clonal pseudotime in each individual under TF perturbation, highlighting opposing roles of *ARX* and *NR2E1* in regulating maturation of RG fate potential. Two-sided paired Wilcoxon tests; *P* = 0.0078 (*NR2E1*), 0.016 (*ARX*), 0.84 (*SOX2*), 0.0016 (*ZNF219*), 0.25 (*NEUROD2*). Centre line, median; box limits, upper and lower quartiles; whiskers, 1.5× interquartile range; points, outliers. **g**, Heatmap showing changes in gene expression in RG along clonal pseudotime. **h**, Violin plots showing gene expression changes in RG in *ARX* and *NR2E1* KD. **P* < 0.05; ***P* < 0.01; ****P* < 0.001.

To identify candidate effector genes involved in RG lineage progression, we compared gene expression in RG with different lineage histories along clonal pseudotime. Lineage tracing studies in rodents and gene expression data in humans have suggested that the fate transition of RG from EN to olfactory bulb IN neurogenesis involves a shift from GLI3 to EGFR signalling[12,90–92]. Similarly, RG in EN-biased clones preferentially expressed *GLI3*, whereas those involved in IN production (EN/IN- and IN-biased clones) expressed *EGFR* and *PDGFRA* highly (Fig. 4g and Extended Data Fig. 7f). As the INs in these clones expressed mainly cortical-like markers, including *SCGN* and *PAX6* (Extended Data Fig. 6d,g), this observation suggests reuse of a conserved signalling pathway transition for olfactory bulb and local IN production by cortical RG. We further identified downstream genes under *NR2E1* and *ARX* KD with lineage-dependent expression (Fig. 4h), nominating shared effector gene programs contributing to RG lineage progression.

To investigate conservation of RG lineage plasticity and underlying mechanisms among primates, we extended our screening approach to rhesus macaque RG (Extended Data Fig. 6a). The production of local INs by cortical RG has been suggested in rhesus macaque[93–95], but not yet demonstrated by lineage tracing. We observed conservation in EN/IN lineage plasticity (Extended Data Fig. 7g–i), including the production of cortical-like INs expressing *SCGN* and *PAX6*, and conserved changes in cell-type compositions (Extended Data Fig. 6g,k,m), gene expression (Extended Data Fig. 6n,o and Supplementary Table 7) with a minority of divergent changes reflecting gene network evolution (Extended Data Fig. 6p) and conserved effects of *ARX* and *NR2E1* on heterochrony of RG clonal output (Extended Data Fig. 7j). Collectively, these results

suggest conserved lineage plasticity of cortical RG in cortical-like IN neurogenesis across primates and conserved roles of *ARX* and *NR2E1* in modulating cortical developmental tempo.

## *ARX* safeguards IN identity by repressing *LMO1*

Beyond lineage plasticity and progression, we also investigated the roles of TFs in subtype specification of cortical neurons. Both *ARX* and *SOX2* KD drove local composition changes among INs (Fig. 3a), motivating us to further examine their effects on subtype abundance. Iterative clustering of INs revealed populations expressing subtype markers including *ST18* (immature), *PAX6* (lateral ganglionic eminence-like) and *CALB2* (CGE-like), as well as a small population expressing *PBX3* (olfactory bulb-like) (Fig. 5a,b and Extended Data Fig. 8a,b). We observed two distinct subclusters defined exclusively by perturbations ('ectopic'): *ARX* KD created a cluster distinguished by *LMO1* (a previously identified target gene repressed by *ARX* in ventral telencephalon[72,80]) and *RIC3* (a chaperone for nicotinic acetylcholine receptors), whereas *SOX2* KD created a subcluster highly expressing *SPOCK1* (a neuronally produced and secreted proteoglycan) (Fig. 5b–d and Extended Data Fig. 8c). Consistent results were observed across several sgRNAs sharing the same target, between individuals, batches and species (Extended Data Fig. 8d–g), highlighting conserved roles of *ARX* and *SOX2* in not only regulating neurogenesis but also specifying normal IN subtype identity, as well as the distinct ectopic transcriptional states formed by their loss.

To further test the relevance of these findings to in vivo differentiation trajectories, we employed organotypic cortical slice culture,

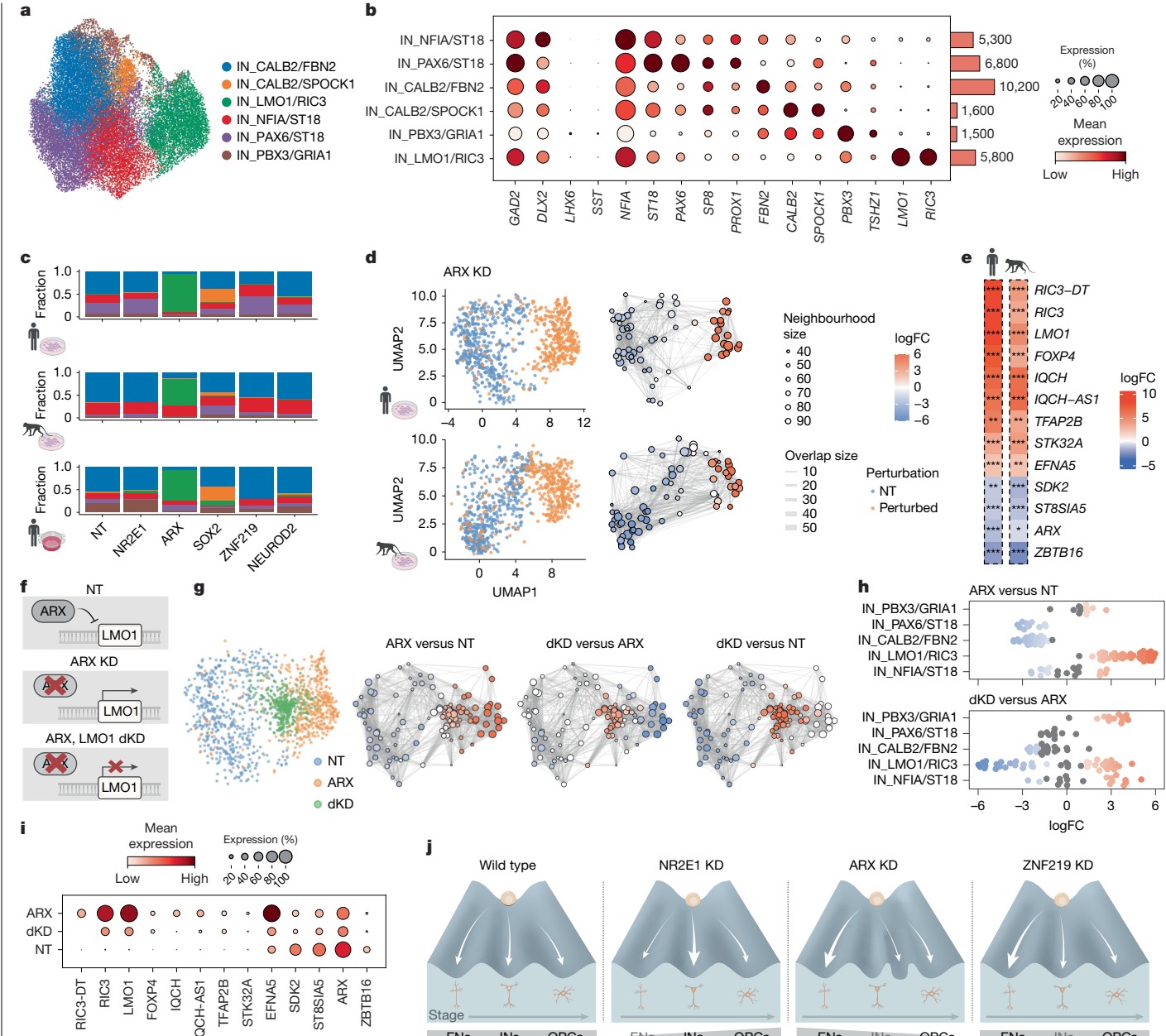

**Fig. 5 | *ARX* safeguards IN identity by repressing *LMO1*. a**, UMAP of integrated human and rhesus macaque IN subtypes identified in the lineage-resolved 2D screen. **b**, Left, dotplot showing marker gene expression in each IN subtype; right, barplot of cell numbers. Dot size denotes the expressing-cell fraction and colour denotes mean expression. **c**, Stacked barplots showing distributions of IN subtypes across TF perturbations in human (top) and macaque (middle) 2D culture and in human slice culture (bottom). **d**, UMAPs showing results of differential abundance testing in IN subtypes in *ARX* KD in human (top) and macaque (bottom). Left, UMAP of downsampled NT and perturbed cells from day 7, coloured by perturbation condition; right, neighbourhood graph of differential abundance testing coloured by log₂FC (FDR = 0.05). **e**, Heatmaps showing log₂FC in top DEGs in the IN_*LMO1/RIC3* cluster versus other IN subtypes in human (left) and macaque (right). Asterisks indicate Benjamini–Hochberg (BH)-adjusted *P* values identified from DEseq2. **f**, Schematics illustrating gene expression regulation under NT, *ARX* KD and *ARX*, *LMO1* dKD.

**g**, UMAPs showing results of differential abundance testing in IN subtypes in NT, *ARX* KD and dKD. Left, UMAP of downsampled NT and perturbed cells coloured by conditions. Right, neighbourhood graph of differential abundance testing coloured by log₂FC (FDR = 0.05) for each contrast: *ARX* KD versus NT, dKD versus *ARX* KD and dKD versus NT. **h**, Beeswarm plots showing differential abundance in neighbourhoods identified in **g** across IN subtypes, with significant changes coloured by log₂FC. **i**, Dotplot showing the IN_*LMO1/RIC3* marker expression in *ARX*, *LMO1* dKD. **j**, Graphical summary of findings: human cortical RG gives rise sequentially to EN, IN and oligodendrocytes along developmental stages. NR2E1 KD promotes the RG lineage progression, whereas ARX KD delays the transition and impairs normal IN subtype specification. ZNF219 KD promotes differentiation of both EN and IN with an overall preference to EN. *\*P < 0.05; \*\*P < 0.01; \*\*\*P < 0.001.* Illustrations in **c**–**e** and schematic in **f** created in BioRender. Nowakowski, T. (https://BioRender.com/7cr6o12). Illustration in **j** created by H. Pinheiro (www.hpinheiro.com).

which preserved the three-dimensional organization of cortical tissue. To preferentially label progenitor populations we locally transduced germinal zone, where progenitors and immature cell types reside, with the targeted CRISPRi and lineage tracing libraries (Extended Data

Fig. 9a–c). Although the sensitivity of this model for detecting cell fate consequences is limited by heterogeneous starter cells and spontaneous differentiation in advance of gene repression[39], the slice culture experiment recapitulated the transcriptomic changes observed in

2D culture (Extended Data Fig. 9d,e) and enabled studies of subtype specification. We observed immature cortically derived INs displaying low expression for migration markers *ERBB4* and *CXCR4* and lineage coupling to ENs (IN_local), along with mature CGE- and medial ganglionic eminence (MGE)-derived IN clusters that were probably labelled post-mitotically and not coupled to ENs (Extended Data Fig. 9f,g). Among INs, ARX repression again induced the *LMO1/RIC3* subtype, while SOX2 repression induced the *CALB1/SPOCK1* subtype, as observed in the 2D model (Fig. 5c and Extended Data Fig. 9h).

Differential gene expression analysis focusing on the *LMO1/RIC3* subtype revealed dysregulated activin and TGFβ signalling pathways driven by *ACVR1* and *SMAD2/3*, as well as impaired cell migration (*SMAD3, FOXP4, CDH20*) and synaptic membrane adhesion (*PTPRD, EFNA5*) (Fig. 5e, Extended Data Fig. 8h,i and Supplementary Table 8). We observed consistent patterns of dysregulated genes in organotypic slice culture and conservation in rhesus macaque, consistent with previous reports of migration defects of INs in *ARX* mutant mice[57] (Fig. 5e and Extended Data Fig. 8j). Despite dysregulation of genes in these pathways, we did not observe an elevated cellular stress response (Extended Data Fig. 8k).

To examine the developmental origin of these ectopic clusters, we analysed lineage coupling and infection during differentiation. We observed similar lineage coupling across cell types in the RG lineage following *ARX* and *SOX2* perturbations in 2D culture (Extended Data Fig. 8l), indicating that effects on subtype identity probably emerge post-mitotically rather than by altering initial progenitor fate specification. Further supporting a post-mitotic effect, the ectopic state emerged in cortical born INs in slice culture under conditions in which RG were not expanded and were permitted to differentiate immediately (Fig. 5c and Extended Data Fig. 9f,h). Furthermore, we observed a similar ectopic cluster characterized by *LMO1* induction in CGE and MGE-derived INs that were infected post-mitotically (Extended Data Fig. 9j–l). Consistent with this observation, ARX expression is retained in post-mitotic INs (Extended Data Fig. 9m–o). Reference mapping to an in vivo atlas of cortical INs[96] showed modest similarity between *LMO1/RIC3* subtypes and MGE-derived *LHX6*⁺ INs, but missing canonical MGE marker expression suggests this subtype is unlikely to have a physiological counterpart (Fig. 5b and Extended Data Fig. 9p).

We next examined the molecular mechanisms underlying the emergence of the *LMO1/RIC3* cluster. As *LMO1* is a conserved downstream target of ARX across species whose expression is dysregulated in *ARX* mutant mice[72,97,98] and acts as a transcriptional coactivator, we reasoned that *LMO1* may mediate the transcriptional programs driving the ectopic cell state. We performed double KD (dKD) with *ARX* and *LMO1* (Fig. 5f) by combining CRISPRi vectors expressing GFP and mCherry and integrated the resulting cell populations with previous data (Extended Data Fig. 10a,b). Reference mapping of IN subtypes revealed enrichment of an intermediate cell state between the *LMO1/RIC3* subtype and other subtypes, suggesting a partial rescue of ectopic cell state driven by *ARX* KD (Fig. 5g,h). Consistent with this model, *LMO1* dKD rescued the expression of many conserved marker genes identified in the *LMO1/RIC3* subtype, including *RIC3* and *EFNA5* (Fig. 5i and Extended Data Fig. 10c–e). These results indicate that *ARX* safeguards normal post-mitotic IN identity, in part by repressing *LMO1*, which is necessary for inducing additional genes characterizing the ectopic cluster. Together, our study introduces a primary cell model of cortical neurogenesis with improved fidelity to normal development and reveals a broadly conserved landscape of responses to TF perturbations in gene regulatory networks, cell fate decisions and neuronal subtype specification (Fig. 5j).

## Discussion

Systematic profiling of single-cell gene expression and chromatin accessibility during human cortical development has supported construction of human cell atlases, inference of developmental trajectories and prioritization of gene regulatory networks[9,10,99,100]. However, functionally examining candidate regulators requires scalable approaches for perturbing gene expression in physiological models of human cortical neurogenesis[101]. We established a primary culture system with improved fidelity and decreased cellular stress response, and applied single-cell Perturb-seq in primary human and rhesus macaque RG undergoing differentiation to examine the role of 44 TFs in cortical development at the level of gene expression, gene network interactions, cell fate biases and subtype specification. Convergence of effector genes downstream of several TFs and their enrichment in neuropsychiatric and neurodevelopmental processes and disorders highlighted the connectedness of TF networks during neurogenesis. Different perturbations elicited distinct RG fate outcomes. By coupling TF perturbation and barcoded lineage tracing, we showed the lineage plasticity of individual human and macaque RG. We identified *ZNF219* as a new regulator in human cortical neurogenesis, and *NR2E1* and *ARX* as known regulators with new functions in regulating inhibitory neurogenesis and guiding temporal lineage progression of individual cortical RG, with post-mitotic roles for *ARX* in IN subtype specification, partially through repressing *LMO1*.

RG undergo waves of neurogenesis and gliogenesis, sequentially generating distinct subtypes of neuron, oligodendrocyte and astrocyte. We derived a culture system that sensitively captured the fate transition of EN to IN production and showed the lineage plasticity of individual RG in response to TF perturbations. *ARX* repression extended the window of EN production, while *NR2E1* repression accelerated the transition to IN and oligodendrocyte production. Mapping clonal clusters along the developmental axis revealed variation in effect sizes of the fate switch potential in RG at different maturation stages, suggesting a restricted time window (GW18–19) for lineage plasticity, corresponding to the timing of endogenous EN/IN transition[12,27]. Combining perturbations with multi-modal profiling including chromatin accessibility of RG at different stages could reveal the underlying *cis*-regulatory changes that account for different cell fate and developmental tempo responses downstream of TF depletion, and extending to genome scales could reveal additional regulators of protracted maturation of human RG.

The dorsal origin of a subset of human cortical INs has been supported through lineage tracing using both lentiviral-based static barcodes and somatic mosaicism[1,12,25–27], but underlying mechanisms regulating this process remained elusive. Previous studies highlighted RG lineage plasticity as a potentially human-specific process in comparison to mouse[1,24], but our lineage tracing experiments in rhesus macaque revealed conservation of lineage plasticity and responses to TF perturbations among primates. Further studies of IN migration and molecular identity will help illuminate possible quantitative differences between species and to distinguish between cortical and olfactory bulb-bound INs generated by cortical RG.

Beyond cell fate commitment and maturation, our screening platform robustly detected changes in IN subtype specification, exemplified by conserved phenotypes observed in *ARX* and *SOX2* KD across primates. Pathogenic ARX mutations associated with neurologic disorders show altered DNA binding preferences and loss of *LMO1* repression[97,98]. Dual repression of *ARX* and *LMO1* partially reverted transcriptional programs driving the *ARX*-mediated ectopic IN state, highlighting *LMO1* repression as crucial for normal IN identity. Increased *LMO1* expression has been associated with tumour invasion and metastasis[102,103], highlighting its potential role in ectopic IN migration. Although *Ebf3*—another *Arx* target—has previously been implicated in Arx-mediated migration deficits[104], future studies combining time lapse imaging and human cellular models can examine the role of *LMO1* as a master regulator mediating aberrant cell behaviours. More generally, the observation that the TFs with the strongest transcriptional and cell composition consequences have also been implicated in neurodevelopmental disorders highlights the correspondence of cellular phenotypes in a simplified

culture system to organismal consequences. Collectively, our study provides a framework for functional dissection of gene regulatory networks in human cortical neurogenesis.

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

## Methods

### Tissue processing and cell culture

**Tissue samples.** De-identified human tissue samples were collected with previous patient consent in strict observance of the legal and institutional ethical regulations. Protocols were approved by the Human Gamete, Embryo and Stem Cell Research Committee (Institutional Review Board) at the University of California, San Francisco.

The Primate Center at the University of California, Davis, provided four specimens of cortical tissue from PCD60 ($n = 1$), PCD75 ($n = 1$) and PCD80 ($n = 2$) rhesus macaques. All animal procedures conformed to the requirements of the Animal Welfare Act, and protocols were approved before implementation by the Institutional Animal Care and Use Committee at the University of California, Davis.

**Cell culture and lentiviral transduction.** For cryopreservation, tissue samples were cut into small pieces, placed in Bambanker (NIPPON Genetics, BB02), frozen in CoolCell at −80 and transferred to liquid nitrogen within 2 weeks. For single-cell dissociation, each cryovial was thawed in 37 °C warm water and placed in a vial containing a pre-warmed solution of Papain (Worthington Biochemical Corporation, catalogue no. LK003153) solution supplemented with 5% trehalose (Fisher Scientific, catalogue no. BP268710) that was prepared according to the manufacturer protocol for 10 min at 37 °C. After approximately 30 min incubation, tissue was triturated following the manufacturer protocol. Cells were plated on a 24-well tissue culture dish coated with 0.1% PEI (Sigma, catalogue no. P3143), 5 µg ml⁻¹ Biolamina LN521 (Invitrogen, catalogue no. 23017-015) and at a density of 500,000 cells cm⁻². Expansion medium contained insulin (Thermo Fisher, catalogue no. A1138IJ), transferrin (Invitria, catalogue no. 777TRF029-10G), selenium (Sigma, catalogue no. S5261-10G), 1.23 mM ascorbic acid (Fujifilm/Wako, catalogue no. 321-44823), 1% polyvinyl alcohol (PVA) (Sigma, catalogue no. P8136-1KG), 100 µg ml⁻¹ primocin (Invivogen, catalogue no. ant-pm-05), 20 ng ml⁻¹ fibroblast growth factor 2 (Preprotech, catalogue no. 100-18B) and 20 ng ml⁻¹ EGF (Preprotech, catalogue no. AF100-15) in DMEM-F12 (Corning, catalogue no. MT10092CM), supplemented with ROCK inhibitor CEPT cocktail[105]. For lentiviral transduction, CRISPRi and STICR lentivirus was added to culture media on day 5 at roughly 1:500 and 1:5,000 dilution, respectively. After 6 h, the virus-containing medium was removed and replaced with fresh medium. Seven days after infection, expansion medium was removed and replaced with differentiation medium containing insulin-transferrin-selenium, 1.23 mM ascorbic acid, 1% PVA, 100 µg ml⁻¹ primocin, 20 ng ml⁻¹ BDNF (Alomone Labs, catalogue no. B-250) in DMEM-F12. At 7 days after differentiation, cultures were dissociated using papain supplemented with 5% trehalose and GFP-positive or GFP and mCherry co-positive cells were isolated by fluorescence-activated cell sorting (FACS) on BD Aria Fusion, resuspended in 0.2% bovine serum albumin (BSA) in PBS and captured with 10x Chromium v.3.1 HT kit (catalogue no. PN-1000348) or Illumina Single Cell CRISPR Library kit (Illumina Single Cell CRISPR Prep, T10, catalogue no. 20132435).

**Organotypic slice culture and lentiviral transduction.** For organotypic slice culture experiments, samples were embedded in 3% low-melting-point agarose (Fisher, catalogue no. BP165-25) and then cut into 300-µm sections perpendicular to the ventricle on a Leica VT1200S vibrating blade microtome in oxygenated artificial cerebrospinal fluid containing 125 mM NaCl, 2.5 mM KCl, 1 mM MgCl₂, 1 mM CaCl₂ and 1.25 mM NaH₂PO₄. Slices were cultured slice medium containing in insulin-transferrin-selenium, 1.23 mM ascorbic acid, 1% PVA, 100 µg ml⁻¹ primocin, Glutamax (Invitrogen, catalogue no. 35050061), 1 mg ml⁻¹ BSA, 15 µM uridine (Sigma, catalogue no. U3003-5G), 1 µg ml⁻¹ reduced glutathione (Sigma), 1 µg ml⁻¹ (+)-α-tocopherol acetate (Sigma, catalogue no. t3001-10g), 0.12 µg ml⁻¹ linoleic (Sigma, catalogue no. L1012) and linolenic acid (Sigma, catalogue no. L2376),

10 mg ml⁻¹ docosahexaenoic acid (Cayman, catalogue no. 10006865), 5 mg ml⁻¹ arachidonic acid (Cayman, catalogue no. 90010.1), 20 ng ml⁻¹ BDNF in DMEM-F12. CEPT cocktail was added on the first day. Lentiviral transduction was performed on the following day locally at the germinal zone to preferentially label neural progenitor cells. CRISPRi and STICR lentivirus was added at 1:20 and 1:100 dilution, respectively. At 24 h after transduction, virus-containing medium was replaced with fresh medium and daily half-medium replacement was performed. At 12–14 days after transduction, cultures were dissociated using papain supplemented with 5% trehalose, and GFP- and mCherry co-positive cells were isolated by FACS and captured with 10x Chromium v.3.1 HT kit (catalogue no. PN-1000348).

**Immunocytochemistry.** Cells were fixed in 2% paraformaldehyde for 15 min at room temperature and then in ice cold 90% methanol for 10 min. After a 1-h incubation in blocking solution (5% BSA, 0.3% Triton-X in PBS), cells were incubated with primary antibodies with the following dilution in blocking solution at room temperature for 1 h: mouse-EOMES (Thermo Fisher, catalogue no. 14-4877-82) 1:500, rabbit-NEUROD2 (Abcam, catalogue no. ab104430) 1:500, goat-SOX9 (R&D, catalogue no. AF3075) 1:300, mouse-KI67 (BD, catalogue no. 550609) 1:500, mouse-DLX2 (Santa Cruz Biotechnology, catalogue no. sc-393879) 1:200. The cells were then washed with PBS three times and incubated with secondary antibodies in blocking solution for 1 h at room temperature, and counterstained with 4′,6-diamidino-2-phenylindole. Finally, after PBS washes, digital image acquisition was performed using an Evos M7000 microscope and Evos M7000 Software.

**Flow cytometry.** Intracellular staining was performed using Foxp3 transcription factor staining kit (Invitrogen, catalogue no. 00-5523-00) according to the manufacturer's protocol. Briefly, cell culture was dissociated with Accutase (STEMCELL Technologies, catalogue no. 07920) supplemented with 5% trehalose (Fisher Scientific, catalogue no. BP268710) and fixed in Foxp3 fixation buffer for 30 min at room temperature. Cells were then washed twice with Foxp3 permeabilization buffer and then stained with primary antibodies mouse-DLX2 (Santa Cruz Biotechnology, catalogue no. sc-393879) and rabbit-NEUROD2 (Abcam, catalogue no. ab104430) at 1:100 dilution. After washing with Foxp3 permeabilization buffer, cells were stained with secondary antibodies donkey anti-mouse-488 and donkey anti-rabbit-647 (Invitrogen) at 1:200. Finally, cells were stained with conjugated antibodies KI67-421 (BD, catalogue no. 565929), SOX2-PerCP-Cy5.5 (BD, catalogue no. 561506), EOMES-PE-Cy7 (Invitrogen, catalogue no. 25-4877-42) at 1:100, washed twice with Foxp3 permeabilization buffer and resuspended in 0.2% BSA in PBS. Cells were then analysed directly by flow cytometry. Data were acquired through BD FACSDiva software and analysed with FlowJo. Fractions of marker positive populations were normalized to the mean value of two NT sgRNAs per individual sample at each timepoint. Two-sided Wilcoxon tests were performed on replicate mean against NT after collapsing two NT sgRNAs to identify significant changes in cell-type composition.

**Immunohistochemistry.** Primary cortical tissue were fixed with 4% paraformaldehyde in PBS overnight, washed three times with PBS, then placed in 10% and 30% sucrose in PBS overnight, sequentially, and embedded in optimal cutting temperature compound for sectioning to 20 µm.

All samples were blocked in blocking solution (5% BSA, 0.3% Triton-X in PBS) for 1 h. Primary and secondary antibodies were diluted in blocking solution. Samples were incubated in primary antibody solution overnight at 4 °C, then washed three times with PBS at room temperature. Samples were then incubated in a secondary antibody solution with 4′,6-diamidino-2-phenylindole for 1 h at room temperature and then washed three times with PBS before being mounted with Fluoromount (Invitrogen, catalogue no. 0100-20). Primary antibodies used include

mouse-DLX2 (Santa Cruz Biotechnology, catalogue no. sc-393879) 1:50; rabbit-SCGN (Millipore-Sigma, catalogue no. HPA006641) 1:500; rabbit-KI67 (Vector, catalogue no. VP-K451) 1:500; mouse-HOPX (Santa Cruz Biotechnology, catalogue no. sc-398703) 1:50; sheep-ARX (R&D, catalogue no. AF7068SP) 1:500. Secondary antibodies in this study include: donkey anti-sheep 647 (Thermo Fisher, catalogue no. A21448) 1:1,000, donkey anti-mouse 488 (Thermo Fisher, catalogue no. A32766) 1:1,000, donkey anti-rabbit 555 (Thermo Fisher, catalogue no. A32794) 1:1,000. Images were collected using ×20 air objectives on an Evos M7000 microscope, and processed using ImageJ/Fiji.

## Target TF selection through enhancer-driven gene regulatory network inference

Two public datasets of single-cell multi omic developing human cortex[10,12] were used for enhancer-driven gene regulatory network (eGRN) inference. The dataset from ref. 12 was subset randomly to 50,000 cells. Cells were grouped into meta cells using SEACells[106] to overcome data sparsity and noise. On average, one meta cell contains 75 single cells and meta cells representing all the cell types originally presented in the dataset were input to the SCENIC+ workflow. Peak calling was performed using MACS2 in each cell type and a consensus peak set was generated using the TGCA iterative peak filtering approach following the pycisTopic workflow. The resulting consensus peaks were then summarized into a peak-by-nuclei matrix and used as input for topic modelling using default parameters in pycisTopic. The optimal number of topics (15) was determined based on four different quality metrics provided by SCENIC+. We applied three different methods in parallel to identify candidate enhancer regions by selecting regions of interest through (1) binarizing the topics using the Otsu method; (2) taking the top 3,000 regions per topic; (3) calling differentially accessible peaks on the imputed matrix using a Wilcoxon rank sum test ($\log_2$FC > 1.5 and Benjamini–Hochberg-adjusted $P$ < 0.05). Pycistarget and discrete-element-method-based motif enrichment analysis were used to link candidate enhancers to TFs. Next, eGRNs, defined as TF-region-gene network consisting of (1) a specific TF, (2) all regions that are enriched for the TF-annotated motif and (3) all genes linked to these regions, were determined by a wrapper function provided by SCENIC+ using the default parameters and subsequently filtered according to the following criteria: (1) only eGRNs with more than ten target genes and positive region-to-gene relationships were retained; (2) eGRNs with an extended annotation was kept only if no direct annotation is available and (3) only genes with top TF-to-gene importance scores (rho > 0.05) were selected as the target genes for each eGRN. Specificity scores were calculated using the RSS algorithm based on region-or gene-based eGRN enrichment scores (area under the curve scores). To infer potential effects of TF repression on RG differentiation, in silico KD simulation was applied following the SCENIC+ workflow to a subset of TFs that showed high correlation between TF expression and target region enrichment scores. Briefly, a simulated gene expression matrix was generated by predicting the expression of each gene using the expression of the predictor TFs, while setting the expression of the TF of interest to zero. The simulation was repeated over five iterations to predict indirect effects. Cells were then projected onto an eGRN target gene-based PCA embedding and the shift of the cells in the original embedding was estimated based on eGRN gene-based area under the curve values calculated using the simulated gene expression matrix. Metrics including (1) normalized TF expression in RG, (2) scaled TF motif accessibility and target gene expression in RG, (3) predicted eGRN sizes, (4) cell-type specificity (RSS) of eGRNs and (5) predicted transcriptomic consequences in the RG lineage were used to select TF targets for this study (Supplementary Table 1).

## Plasmids and lentivirus production

**Plasmids.** The all-in-one CRISPRi plasmid encoding dCas9-KRAB and sgRNA were obtained from Addgene (catalogue no. 71237). Capture sequence 1 was cloned into the vector by Vectorbuilder to enable sgRNA capture with 10x Genomics single-cell capture[50]. mCherry-H2B was cloned into the vector to substitute GFP for lineage tracing and flow cytometry experiments. For single sgRNA cloning, the oligonucleotides containing 20 bp protospacer and overhang were obtained from Integrated DNA Technologies and cloned into the *Bsm*BI-v2 (New England Biolabs, catalogue no. R0739)-digested backbone through T4 ligation (T4 DNA ligase, New England Biolabs, catalogue no. M0202). Protospacers were obtained from dual sgRNA CRISPRi libraries[107] or Dolcetto[52]. For experiments in rhesus macaques, protospacers that were mapped uniquely to the rheMac10 genome were selected. STICR plasmids (Addgene catalogue numbers 180483, 186334, 186335) used for lineage tracing were obtained from the Nowakowski laboratory.

**sgRNA library construction.** Oligonucleotide pools were synthesized by Twist Bioscience. *Bsm*BI recognition sites were appended to each sgRNA sequence along with the appropriate overhang sequences for cloning into the sgRNA expression plasmids, as well as primer sites to allow differential amplification of subsets from the same synthesis pool. The final oligonucleotide sequence was: 5′-(forward primer) CGTCTCA*CACCG*(sgRNA, 20 nt)*GTTT*CGAGACG(reverse primer).

Primers were used to amplify individual subpools using 50 µl 2× NEBNext Ultra II Q5 Master Mix (New England Biolabs, catalogue no. M0544S), 20 µl of oligonucleotide pool (approximately 20 ng), 0.5 µl of primer mix at a final concentration of 0.5 µM, and 29 µl nuclease-free water. PCR cycling conditions: (1) 98 °C for 30 s; (2) 98 °C for 10 s; (3) 68 °C for 30 s; (4) 72 °C for 30 s; (5) go to (2), eight times; (6) 72 °C for 2 min.

The resulting amplicons were PCR-purified (Zymo, catalogue no. D4060) and cloned into the library vector using Golden Gate cloning with Esp3I (Thermo Fisher Scientific, catalogue no. ER0451) and T7 ligase (New England Biolabs, catalogue no. M0318); the library vector was pre-digested with *Bsm*BI-v2 (New England Biolabs, catalogue no. R0739). The ligation product was electroporated into MegaX DH10B T1R Electrocomp cells (Invitrogen, catalogue no. C640003) and grown at 30 °C for 24 h in 200 ml LB broth with 100 µg ml⁻¹ carbenicillin. Library diversity and sgRNA representation were assessed through PCR amplicon sequencing.

**Lentivirus production.** Lentivirus was produced in HEK293T cells (Takaro Bio). HEK293Ts were seeded at a density of 80,000 cells cm⁻² 24 h before transfection. Transfection was performed using Lipofectamine 3000 (Invitrogen, catalogue no. L3000001) transfection reagent according to the manufacturer's protocol. After 6 h, the medium was removed and replaced with fresh medium supplemented with 1× ViralBoost (Alstem, catalogue no. VB100). Supernatant was collected at 48 h after transfection and concentrated roughly at 1:100 with lentivirus precipitation solution (Alstem, catalogue no. VC100).

## Generation and analysis of scRNA-seq libraries

**scRNA-seq library generation.** The manufacturer-provided protocol (CG000421 Rev D) was used to generate 10x single-cell gene expression and sgRNA libraries. Samples from several individuals were pooled (Supplementary Table 3) and each 10x lane was loaded with ~100,000 cells in total. sgRNA libraries were amplified separately from each 10x cDNA library using the 10x 3′ Feature Barcode Kit (catalogue no. PN-1000262). To generate STICR barcode libraries, 10 µl of 10x cDNA library was used as template in a 50 µl PCR reaction containing 25 µl Q5 Hot Start High Fidelity 2× master mix (NEB, catalogue no. M0494) and STICR barcode read 1 and 2 primers (0.5 µM, each) described in Delgado et al.[1] using the following program: 1, 98 °C, 30 s; 2, 98 °C, 10 s; 3, 62 °C, 20 s; 4, 72 °C, 10 s; 5, repeat steps 2–4 15 times; 6, 72 °C, 2 min; 7, 4 °C, hold. A 0.8–0.6 dual-sided size selection was performed using SPRIselect Bead (Beckman Coulter, catalogue no. B23318) after PCR amplification.

For the *ARX*, *LMO1* dKD experiment, cells are captured with Illumina Single Cell CRISPR Prep (T10, catalogue no. 20132435) where direct sgRNA capture was enabled. Gene expression and sgRNA libraries were prepared following the manufacturer-provided protocol (FB0004762; FB0002130).

**Alignments and quality control.** Libraries were sequenced on Illumina NovaSeq platforms to the depth of roughly 20,000–30,000 reads per cell for gene expression and 5,000 reads per cell for sgRNA and STICR libraries. Data were acquired using Illumina sequencer control software and bcl2fastq software.

The 10x gene expression libraries, together with sgRNA and STICR libraries, were aligned to the hg38 genome obtained from CellRanger (refdata-gex-GRCh38-2024-A) with feature barcode reference (Supplementary Table 2) using CellRanger v.7.2.0. Aligned counts were then processed with Cellbender[108] for ambient removal. The resulting counts were processed by Scanpy[109] to remove low quality cells containing fewer than 1,000 genes, a high abundance of mitochondrial reads (greater than 15% of total transcripts) or a high abundance of ribosomal reads (greater than 40% of total transcripts). Illumina single-cell CRISPR libraries were aligned using DRAGEN single-cell RNA v.4.4.5 with additional arguments --single-cell-barcode-tag and --single-cell-umi-tag to support downstream individual demultiplexing. Low quality cells that had fewer than 500 genes or high abundance of mitochondrial or ribosomal reads were removed.

To align 10x runs with pooled human and rhesus macaque individual samples (Supplementary Table 3), we made a composite genome using CellRanger mkref function with (1) hg38 (CellRanger refdata-gex-GRCh38-2024-A) and (2) a version of rheMac10 included in a hierarchical alignment and annotated with the Comparative Annotation Toolkit[110] as part of ref. 111, then filtered with litterbox (https://github.com/nkschaefer/litterbox). Cells identified as cross-species multiplets were removed from downstream analyses.

**Demultiplexing of individuals from pooled sequencing and doublet removal.** CellSNP-lite followed by Vireo[112,113] were used to identify different individuals based on reference-free genotyping using candidate SNPs identified from 1000 Genome Project. Sex information was acquired through PCR-based genotyping using genomic DNA extracted from each person. Each person was assigned based on sex and pooling information (Supplementary Table 3). The same individuals from different 10x lanes were merged based on identical SNP genotypes. Inter-individual doublets and unassigned cells were removed from downstream analyses.

**sgRNA and lineage barcode assignments.** Cellbouncer[114] was used for sgRNA assignment using the sgRNA count matrix obtained from CellRanger. 'effective_sgRNA' was defined for each cell based on assigned sgRNAs after collapsing NT sgRNA. For example, cells assigned with NT sgRNA and SOX2-targeting sgRNA are classified as 'effective_sgRNA'=SOX2. 'Target_gene' was defined for each cell based after collapsing NT sgRNA and sgRNAs sharing the same target genes. For example, cells infected with two different SOX2-targeting sgRNAs are classified as 'Target_gene'=SOX2 and included for analysing SOX2 KD phenotypes at the gene level. Multi-infected cells with sgRNAs targeting different genes were assigned to have more than one 'Target Gene' and removed from downstream analyses.

KD efficiency in the initial screen was calculated in each cell class following the DEseq2 (ref. 115) workflow, where single-cell data were pseudobulked by cell class per sgRNA per individual. We reasoned that KD efficiency can vary between cell types and classes depending on the baseline expression level of the target gene, and that KD effects may be difficult to detect in cell types with low baseline expression. $\log_2$FC output by the DEseq2 Wald test was obtained in each cell class and the lowest $\log_2$FC among cell classes was used for visualization

and downstream filtering (Supplementary Table 4). sgRNAs that had KD efficiency less than 25% ($\log_2$FC > −0.4, 18 sgRNAs) were considered inactive and excluded from downstream analyses. KD efficiency at the gene level was calculated pseudobulking each supervised cell type per TF per gene per individual after collapsing all active sgRNAs targeting the same gene. Pearson correlation coefficients comparing $\log_2$FC of DEGs (DEseq2 Wald test, Benjamini–Hochberg-adjusted $P < 0.05$) at the sgRNA level and at the gene level were calculated to evaluate variabilities between sgRNAs.

STICR barcodes for lineage tracing experiments were aligned and assigned using a modified NextClone[116] workflow that allows for whitelisting. The pipeline is available at https://github.com/cnk113/NextClone. Individual barcodes were filtered by at least three reads supporting a single unique molecular identifier and at least two unique molecular identifiers to call cells with a barcode. Clone calling was done using CloneDetective[116].

**Comparison with public datasets and cell-type annotation.** The processed developing human cortex multiomic dataset, including metadata and aligned CellRanger output, was obtained from ref. 12 and used for reference mapping. The reference model was built with scvi-tools[117] using the top 2,500 variable genes, and label transfer to query datasets generated in this study was performed for data integration and to examine correspondence of cell-type assignment to the reference dataset. Cell-type annotation in the initial human screen generated in this study was performed based on marker expression as well as scANVI predictions. The initial human screen dataset was then used as a reference dataset to map other 2D populations collected from lineage-resolved targeted screen in human and rhesus macaque, and the *ARX*, *LMO1* dKD experiment to minimize impacts of batch effects and species differences on cell-type annotation.

Public datasets of processed single-cell RNA-seq from iNeuron[53], FeBO[54] and iPS-cell-derived organoid[55] studies were used to compare fidelity of in vitro specified cell types across different systems. Reference mapping and label transfer to the ref. 12 in vivo dataset were performed to examine correspondence to the in vivo cell types. Pearson correlation coefficients per cell type between datasets were calculated using expression of the top 25 markers identified from the in vivo dataset to compare the fidelity of cell identities. To assess the level of cellular stress, gene scores for glycolysis, endoplasmic reticulum stress, reactive oxygen species (ROS), apoptosis and senescence were calculated using the score_genes function in Scanpy with gene sets obtained from the MSigDB database[118,119]. NT control cells of all cell types were subset from data obtained in this study for this comparison. Each dataset was then downsampled randomly to 5,000 cells to ensure balanced representation between datasets.

**Trajectory analysis using pseudotime and RNA velocity.** The initial human screen data was subset to NT cells for pseudotime and RNA velocity inference. Pseudotime was calculated using scFates[120] by tree learning with SimplePPT and setting the root node within the RG class on ForceAtlas2 embedding. Excitatory and inhibitory trajectories were defined as main branches of the principal graph that led to distinct sets of clusters. The velocyto[121] pipeline was implemented to quantify spliced and unspliced reads from CellRanger output. Highly variable genes were separately calculated using spliced and unspliced matrices and the top 3,000 genes were used for inferring RNA velocity using scVelo[122].

**Differential composition and gene expression analysis.** Cells with single 'Target_gene' were subset and sgRNAs targeting the same gene were collapsed for downstream analyses at the gene level. Composition changes in each cluster per TF perturbation were quantified using DCATS[123], which detects differential abundance using a beta-binomial generalized linear model model and returns the estimated coefficients

and likelihood ratio test $P$ values. To detect compositional changes in a finer resolution, Milo[76] was used to quantify differential abundance in a label-free manner. Briefly, perturbed and/or NT cells were downsampled randomly to ensure the balance of total cell numbers between the two groups. Neighbourhoods were constructed based on $k$-nearest-neighbour graphs, and differential cell abundance in each neighbourhood was then tested against NT using design = ~stage + sex + batch + perturbation. Robustness to downsampling was tested in four perturbations through downsampling randomly to 200–600 cells.

Cluster-aware differential gene expression analysis was performed following the DEseq2 (ref. 115) pipeline using NT cells as reference and individual as replicates. When pseudobulking within cell types, conditions that have fewer than five cells per cell type or fewer than two biological replicates were removed from downstream analyses. DEGs (DEseq2 Wald test, Benjamini–Hochberg-adjusted $P < 0.05$) identified under more than one perturbation across 44 TFs perturbations at the gene level in the initial human screen were defined as 'convergent DEGs' and gene ontology enrichment analysis was performed using non-convergent DEGs as background with richR (https://github.com/guokai8/richR). One-sided Fisher's exact test was performed to identify significant overlap between convergent DEGs and seven sets of disease-associated genes (see below) using non-convergent DEGs as background.

To prioritize TFs whose repression led to strongest cellular and transcriptional and consequences in the initial human screen, |estimated coefficient| from DCATS and numbers of DEGs from DEseq2 across seven cell types in the EN and IN lineage (RG_DIV, RG-Astro, IPC_EN, EN_ImN, EN, IPC_IN, IN) were summed to quantify accumulated effects of TF repression in composition and gene expression, respectively. To test the robustness of prioritized TFs, scCoda[124] was applied to estimate $\log_2$FC in cell abundance per cell type. Euclidean and energy distance between each TF perturbation were calculated with Pertpy[125] to compare the global transcriptional landscape post perturbation.

The DCATS and DEseq2 pipeline described above was also implemented in the datasets generated in lineage-resolved targeted screens in human and macaque 2D culture and in human organotypic slice culture to compare results between batches, cultural systems and species, and to support robustness and reproducibility of the phenotypes reported.

**Disease-associated genes from previous studies.** Genes significantly associated with neurodevelopmental and neuropsychiatric disorders were obtained from:
- ASD: SFARI gene database[85], score 1
- NDD: Supplementary Table 11 in ref. 86
- MDD: Supplementary Table 9 in ref. 126
- SCZ: Supplementary Table 12 in ref. 127
- BP: Supplementary Table 4 in ref. 128
- ADHD: Supplementary Table 7 in ref. 129
- AD: Supplementary Table 5 in ref. 130

**Lineage coupling and clonal clustering.** Clones with fewer than three cells and clones that have conflicted sgRNA assignments were removed from the clonal analysis to consider only multicellular clones generated post infection. One human individual (GW17 187-M) with overall low cellular coverage was dropped from the subsequent analyses. Cospar[88] was used to calculate the fate coupling scores per perturbation, defined as the normalized barcode covariance between different cell types.

scLiTr[89] was used to identify and cluster fate biased clones by training a neural network to predict clonal labels of nearest neighbours for each clonally labelled cell. To minimize batch effects in cluster identification, we used integrated lineage-resolved data to define five clusters with distinct fate biases. The results obtained were exported and a two-sided paired Wilcoxon test was performed comparing the mean fractions of clonal clusters per TF perturbation against NT. Wilcoxon rank sum test through Scanpy was used to identify marker genes in the RG class

(RG-Astro and RG_DIV) between different clonal clusters in human. Monocle3 (ref. 131) was used to construct pseudotime trajectory at the clonal level per species. The median pseudotime per individual was calculated under each perturbation and used for two-sided paired Wilcoxon tests to evaluate the changes on clonal pseudotime. To fit gene expression along the clonal pseudotime, cells in the RG class from multicellular clones were subset and assigned with pseudotime values based on their clone identities.

**IN subtype identification and gene expression analysis.** To identify IN subtypes, the IN cluster from the initial human screen was subset and re-clustered into six subtypes, each expressing distinct markers identified through Scanpy using Wilcoxon rank sum test. Reference mapping and label transfer were then performed querying INs generated in the lineage-resolved human and macaque datasets and the dKD dataset to identify equivalent clusters. For cell abundance testing in IN subtypes in each perturbation, NT and perturbed INs were downsampled and the Milo pipeline described above was applied to identify differential abundance within the IN cluster. Differential expression analysis was performed using DEseq2 contrasting the IN_LMO1/RIC3 cluster with other physiological subtypes identified in unperturbed INs (IN_NFIA/ST18, IN_PAX6/ST18, IN_CALB2/FBN2, IN_PBX3/GRIA1) in each species, using individuals as replicate. Gene ontology terms enrichment in biological processes in the human IN_LMO1/RIC3 cluster were identified using pathfindR[132] (one-sided hypergeometric test).

To survey physiological counterparts of the IN_LMO1/RIC3 cluster, an in vivo atlas of the developing macaque telencephalon[96] was used as reference and label transfer using scvi-tools was performed to map INs generated in the targeted 2D screen. Gene scores for glycolysis, endoplasmic reticulum stress, ROS, apoptosis and senescence were calculated in INs per perturbation as described above to assess changes in cellular stress induced by TF perturbation.

**Comparison with organotypic slice culture dataset.** The lineage-resolved targeted screen dataset in human organotypic slice culture was preprocessed as described above. Cell-type annotation was done based on marker gene expression and reference mapping to the ref. 12 atlas. Only cells derived from multicellular clones were included for visualizing cell-type compositions per TF perturbation to consider the effects on dividing progenitors.

Four IN clusters, including those derived from RG in vitro (IN_local) and those labelled post-mitotically (IN_MGE, IN_CGE, IN_OB), were used to compare the pre- and post-mitotic effects of *ARX* KD on IN gene expression and subtype specification using Milo and DEseq2 as described above. IN_local was used for iterative clustering to identify IN subtypes described in 2D and compare between perturbations.

***ARX* and *LMO1* dKD.** A GFP expressing CRISPRi library targeting *LMO1* and NT controls, together with an mCherry-expressing vector targeting *ARX*, were used to achieve dKD and double infected cells expressing both colours were sorted and captured with Illumina Single Cell CRISPR Library Kit. The resulting library was aligned with DRAGEN Single Cell RNA v.4.4.5, reference mapped to the initial human screen data and integrated with previous data from the targeted human screen. sgRNA assignment was done using cellbouncer and double infected INs (*ARX*, *LMO1* dKD or *ARX*, NT KD) were used for downstream analysis. Differential gene expression in INs was performed using the DEseq2 Wald test by contrasting *ARX*, *LMO1* dKD against *ARX* KD. Cluster-free differential abundance testing was performed using Milo after integrating with previous batches to compare differential abundance of IN subtypes in NT, *ARX* KD and dKD.

## Ethics and inclusion statement
All studies were approved by UCSF GESCR (Gamete, Embryo, and Stem Cell Research) Committee.

## Reporting summary

Further information on research design is available in the Nature Portfolio Reporting Summary linked to this article.

## Data availability

Raw sequencing data generated in this study are publicly available through dbGaP under accession number phs002624 as of the date of publication. Processed data for both species and raw data for macaque specimens are deposited on GEO under accession number GSE284197. Processed human data can be visualized at cortical-lineage-perturb-44tf.cells.ucsc.edu. Processed single-cell datasets from previous publications that are used in this study include: (1) Wang et al. developing human cortex[12]: https://cellxgene.cziscience.com/collections/ad2149fc-19c5-41de-8cfe-44710fbada73; (2) Trevino et al. developing human cortex[10]: GEO GSE162170; (3) iNeurons[53]: ArrayExpress E-MTAB-10632; (4) primary brain organoids FeBO[54]: GEO GSE248481; (5) iPS-cell-derived organoids[55]: https://cellxgene.cziscience.com/collections/de379e5f-52d0-498c-9801-0f850823c847; and (6) developing macaque telencephalon[96]: GEO GSE226451.

## Code availability

Codes used for data analysis and visualization in this study are available at https://github.com/Pollen-lab/perturbTF.

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

**Acknowledgements** We thank M. Paredes, D. Shin, M. Steyert, L. El Hayek, N. Sanchez-Luege and other Pollen laboratory members for comments, H. Pinheiro (www.hpinheiro.com) for the illustration in Fig. 5j, J. Srivastava and V. Saware for their assistance performing cell sorting at the Gladstone Flow Cytometry Core, supported by National Institutes of Health (NIH) grant S10 RR028962, L. Wang for access to the multiome dataset in the developing human brain, S. Wang for transferring samples and A. Tarantal for providing samples. Sequencing was performed at the UCSF CAT, supported by UCSF PBBR, RRP IMIA and NIH 1S10OD028511-01 grants. This work was supported by the following funding sources: CIRM fellowship (J.W.D.), NIH R01MH134981-01, DP2MH122400-01 (A.A.P.), UM1MH130991 (A.A.P. and T.J.N.), R01NS123263 (T.J.N.), Schmidt Futures Foundation (A.A.P. and T.J.N.), William K. Bowes Jr. Foundation (A.A.P. and T.J.N.), Shurl and Kay Curci Foundation (A.A.P. and T.J.N.), the Arc Ignite Award (A.A.P.), the California Institute for Regenerative Medicine (CIRM) DISCO-14429 (T.J.N.) DISC4-16285 (A.A.P.) as well as gifts from the Esther A. & Joseph Klingenstein Fund. T.J.N. is a New York Stem Cell Foundation Robertson Neuroscience Investigator. A.A.P. is a New York Stem Cell Foundation Robertson Investigator.

**Author contributions** J.W.D. and A.A.P. conceived the project and experimental design. A.A.P. supervised the study and secured the funding. J.W.D. performed all the experiments and analysed the data with the help of C.N.K. under the supervision of T.J.N. and A.A.P. C.N.K. performed STICR barcode assignment. M.S.O. helped on the validation with flow cytometry. J.L.W. and B.J.P. performed initial testing of the primary culture system. Y.A. prepared and tested the CRISPRi vector. N.K.S. supported sgRNA assignments and prepared the rhesus macaque reference genome. J.W.D. and A.A.P. prepared the manuscript with input from all authors.

**Competing interests** The authors declare no competing interests.

**Additional information**
**Correspondence and requests for materials** should be addressed to Alex A. Pollen.

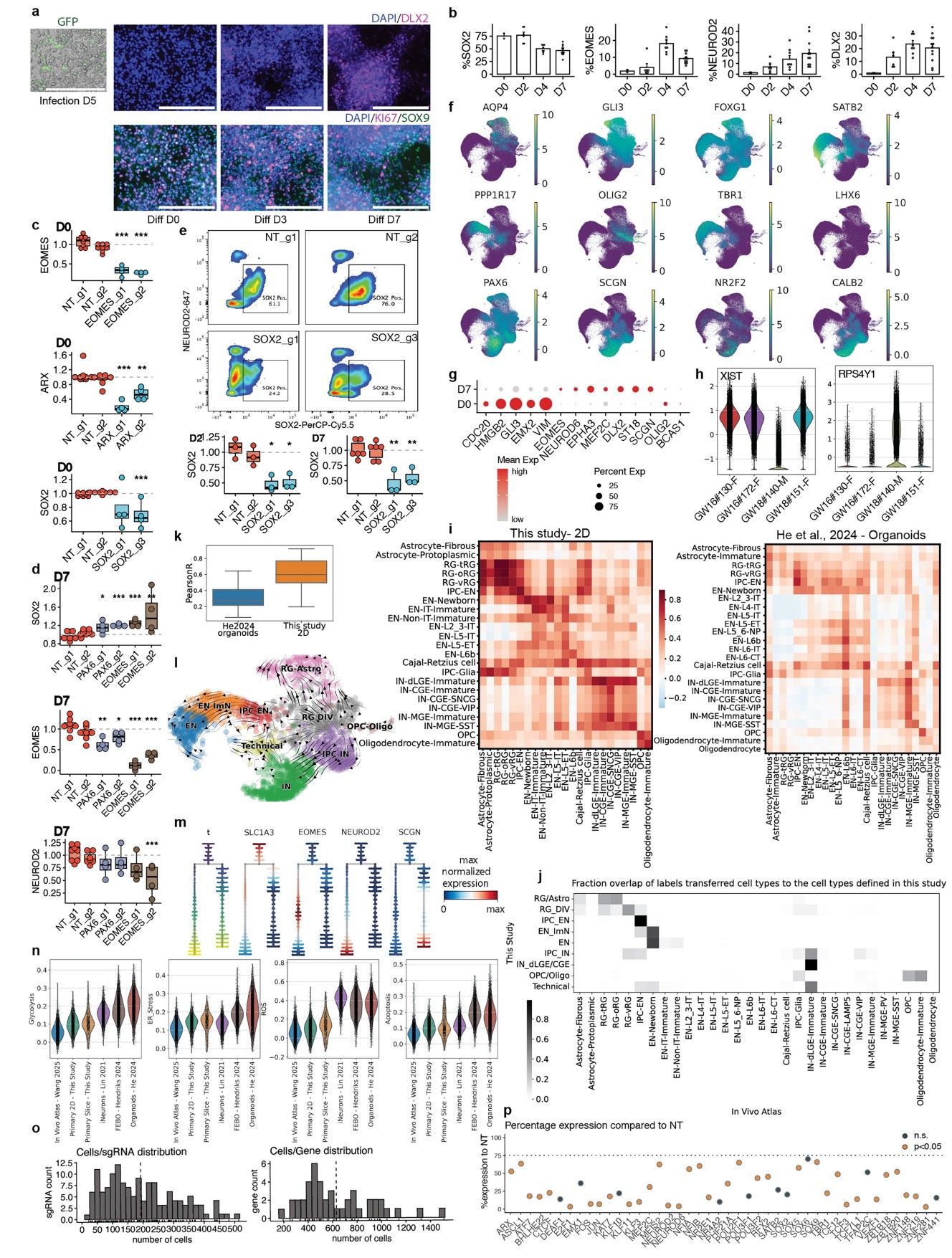

**Extended Data Fig. 1** | See next page for caption.

**Extended Data Fig. 1 | Supplemental analysis of the primary model, CRISPRi, and Perturb-seq dataset. a**. (Left) 2D culture of primary human cortical RG showing GFP expression on infection D5. (Right) Immunocytochemical labelling of SOX9, MKI67 and DLX2 before (D0) and after induced differentiation (D3, D7). Representative images from four individuals are shown. **b**. Barplots showing fractions of SOX2, KI67, EOMES and DLX2 positive populations across the differentiation timepoints (D0, D2, D4, D7) in cells expressing all-in-one CRISPRi vectors with NT sgRNAs. Quantification was done through flow cytometry. Data are presented as mean values +/− SEM. Dots represent independent biological replicates and sgRNAs. D0, n = 2; D2, n = 3 × 2 sgRNAs; D4, n = 3 × 2 sgRNAs; D7, n = 6 × 2 sgRNAs. **c**. Barplots showing fold changes in fractions of EOMES, ARX and SOX2 positive populations relative to NT in respective KDs to confirm KD efficiency before differentiation (D0), measured by flow cytometry (n = 4 individuals). Dots represent biological replicates from independent individuals and sgRNAs. Two-sided Wilcoxon test on replicate means after collapsing 2 NT sgRNAs; p = 0.00041, 0.00041 (EOMES); 0.00041, 0.0017 (ARX); 0.15, 0.00083 (SOX2). Centre line, median; box limits, upper and lower quartiles; whiskers, 1.5X interquartile range; points, outliers. **d**. Barplot showing fold changes in fractions of SOX2, EOMES and NEUROD2 positive populations measured by flow cytometry in *PAX6* and *EOMES* KD versus NT on D7. Dots represent biological replicates from independent individuals and sgRNAs (n = 4 individuals x 2 sgRNAs). Increase in SOX2 and decrease in EOMES and NEUROD2 populations were detected in both perturbations. Two-sided Wilcoxon test; p values in significant conditions: 0.049, 0.00052, 0.00052, 0.0062 (SOX2); 0.0062, 0.037, 0.00052, 0.00052 (EOMES); 0.00052 (NEUROD2). Centre line, median; box limits, upper and lower quartiles; whiskers, 1.5X interquartile range; points, outliers. **e**. Flow cytometry quantification of fractions of SOX2 positive populations versus NT to confirm efficient KD throughout differentiation (D2, D7). Two-sided Wilcoxon test; p = 0.024, 0.024 (D2); 0.0044, 0.0044 (D7). **f**. UMAP of cells collected on D0 and D7, coloured by marker gene expression. **g**. Dotplot of marker gene expression, grouped by differentiation timepoints. **h**. Violin plot of *XIST* and *RPS4Y1* expression in each individual, highlighting inferred sex genotype. **i**. Heatmap showing Pearson correlation coefficients of top 25 cell marker gene expression defined in Wang et al.[12] in vivo atlas to datasets from this study (left) and from He et al.[55] organoid atlas (right). Cell type annotation was done using reference mapping of both datasets to the Wang et al. dataset. **j**. Heatmap for fraction overlap of labels transferred from reference cell types[12] to the cell types defined in this study. **k**. Boxplot showing Pearson correlation coefficients in N = 25 (He et al.) and 23 (this study) predicted cell types (diagonal values from (**i**)). Centre line, median; box limits, upper and lower quartiles; whiskers, 1.5X interquartile range. **l**. UMAP of RNA velocity on D7 NT cells, coloured by supervised cell types. **m**. Pseudotime tree plots coloured by pseudotime and marker gene expression. **n**. Violin plots showing gene scores for glycolysis, ER stress, ROS, apoptosis and senescence across six datasets including in vivo human atlas[12], 3D organotypic slice culture, 2D primary RG culture, iNeurons[53], primary brain organoids FeBO[54] and iPSC-derived organoids[55]. Organotypic slice culture data presented at Extended Data Fig. 9 is used here for system benchmarking. **o**. Histogram showing distributions of singly infected cell counts for each TF-targeting sgRNA (top) and TF (bottom) on D7. **p**. Dotplot showing target gene KD efficiency compared to NT cells on D7 at the gene level after collapsing active sgRNAs targeting the same gene, colored by significance. $\log_2$FC were calculated with DEseq2 in each cell class in n = 4 individuals. *, p < 0.05; **, p < 0.01; ***, p < 0.001. ER, endoplasmic reticulum; ROS, reactive oxygen species. Scale bars: 200 μm.

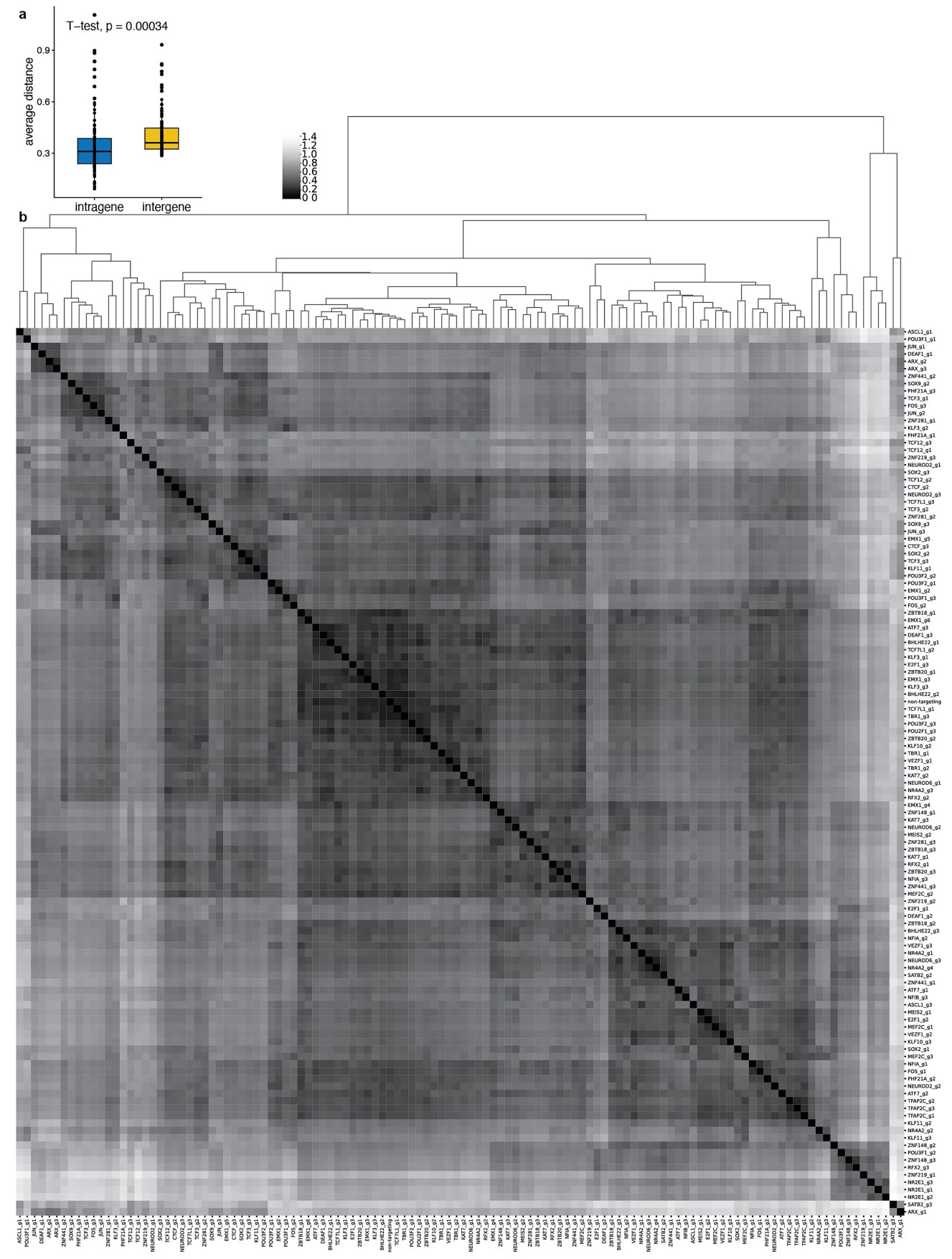

**Extended Data Fig. 2 | Euclidean distance between sgRNAs on differentiation D7. a.** Boxplot showing mean Euclidean transcriptional distance between sgRNAs targeting the same gene (intragene, blue) or different genes (intergene, yellow). Two-sided paired t-test across n = 144 TF-targeting sgRNAs, p = 0.00034. Centre line, median; box limits, upper and lower quartiles; whiskers, 1.5X interquartile range; points, outliers. **b.** Heatmap showing pairwise Euclidean distances between sgRNAs.

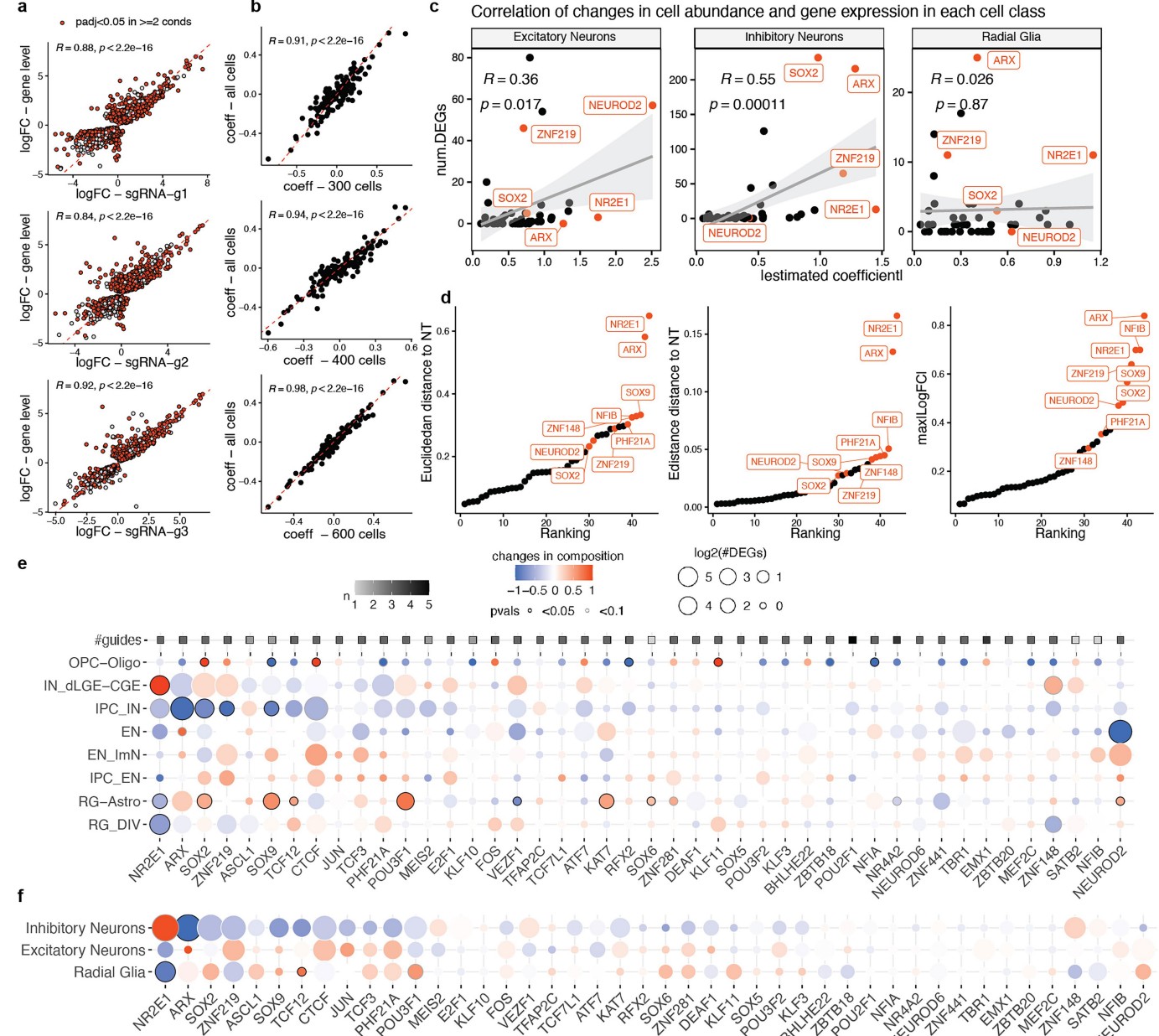

**Extended Data Fig. 3 | Supplemental analysis for responses to TF perturbation and target prioritization. a**. Scatterplots of logFC of DEGs from DEseq2 Wald test in each class at the sgRNA (X axis) and gene (Y axis) levels. Significant DEGs (BH-adjusted p < 0.05) at the gene level are shown. Red dots denote DEGs detected in ≥2 conditions; Pearson correlation coefficients and two-sided p values are shown; red dashed line represents Y = X. **b**. Scatterplots of estimated coefficients for differential abundance from DCATS in each class in downsampled (X axis) versus full populations (Y axis); each condition was randomly downsampled to 300, 400 and 600 cells. Pearson correlation coefficients and two-sided p values are shown. Red dashed line represents Y = X. **c**. Scatterplots showing correlation between summed |estimated coefficient| in cell abundance (X axis) and number of DEGs (Y axis) per cell type within 3 classes

(Excitatory Neurons, Inhibitory Neurons, Radial Glia). Pearson correlation coefficients and two-sided p values are shown with regression lines (95% CI). **d**. Plots showing (left) Euclidean distance to NT, (middle) energy distance[56] to NT, (right) maximum |log₂FC| of cell abundance at the gene level among eight cell types listed in (**e**). **e**. Dotplot showing global changes in cell abundance (color, DCATS coefficient; border, p value) and gene expression (size, number of DEGs, BH-adjusted p < 0.05) across eight cell types. Top row, number of active sgRNAs per TF. **f**. Dotplot showing global changes in cell abundance and gene expression across three major classes. Note that the overall trend toward depletion of IN_IPCs across perturbations drives broader depletion of the IN lineage at the level of class, even in cases like *SOX2, ZNF219* which showed postmitotic IN enrichments.

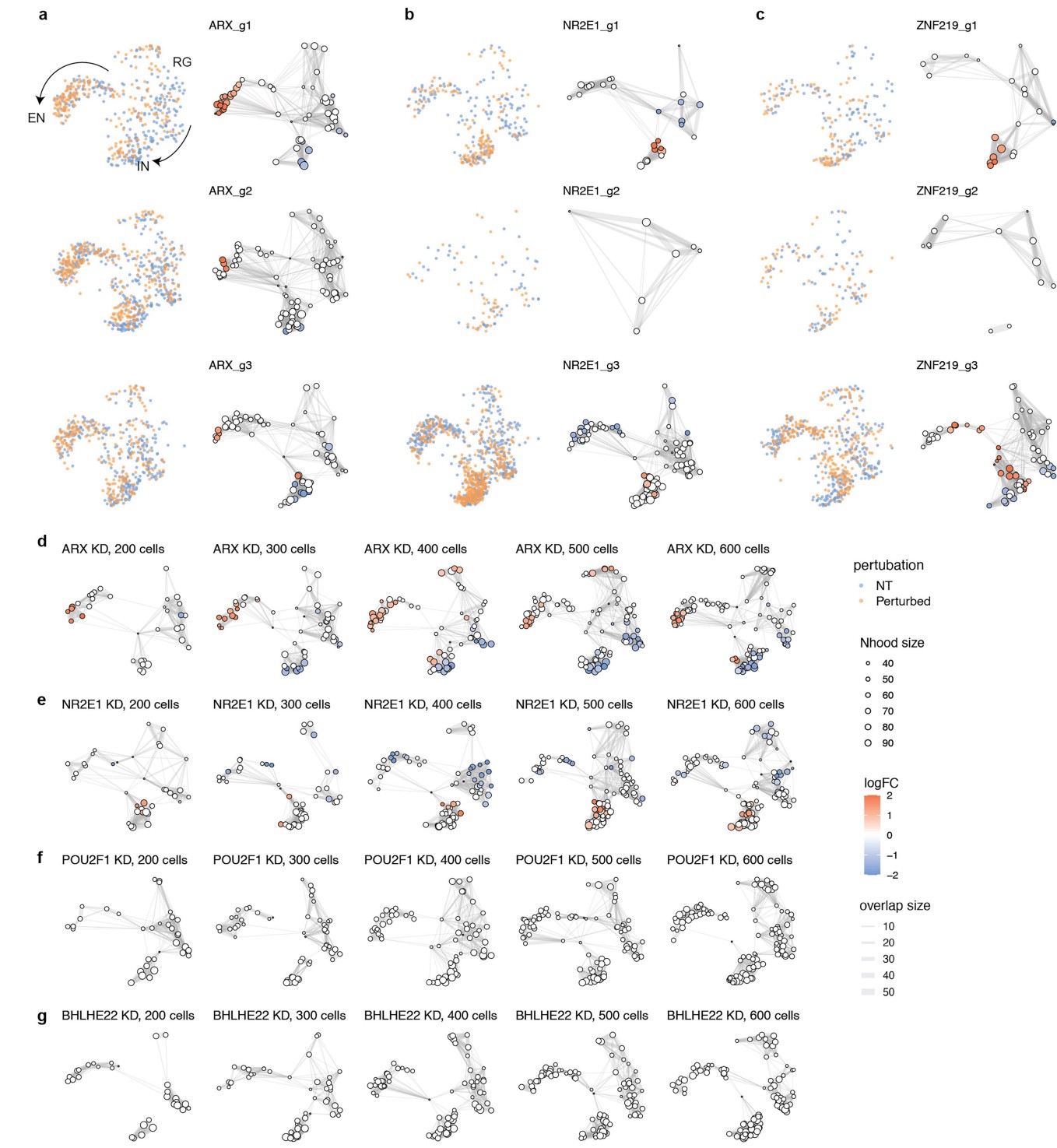

**Extended Data Fig. 4 | Supplementary analysis for cluster-free differential abundance testing using Milo. a-c**. UMAPs showing results for differential abundance testing using Milo at the sgRNA level under *ARX*(**a**), *NR2E1*(**b**) and *ZNF219*(**c**) KD. (left) UMAPs of downsampled NT and perturbed cells on D7, colored by condition. (right) Neighborhood graphs. Node size denotes cell numbers in each neighborhood; edge depicts the number of cells shared between neighborhoods; node color denotes log₂FC (FDR = 0.1) when perturbed. **d-g**. UMAPs showing results at the gene level under *ARX*(**d**), *NR2E1*(**e**), *POU2F1*(**h**), and *BHLHE22*(**g**) KD after downsampling to 200–600 cells.

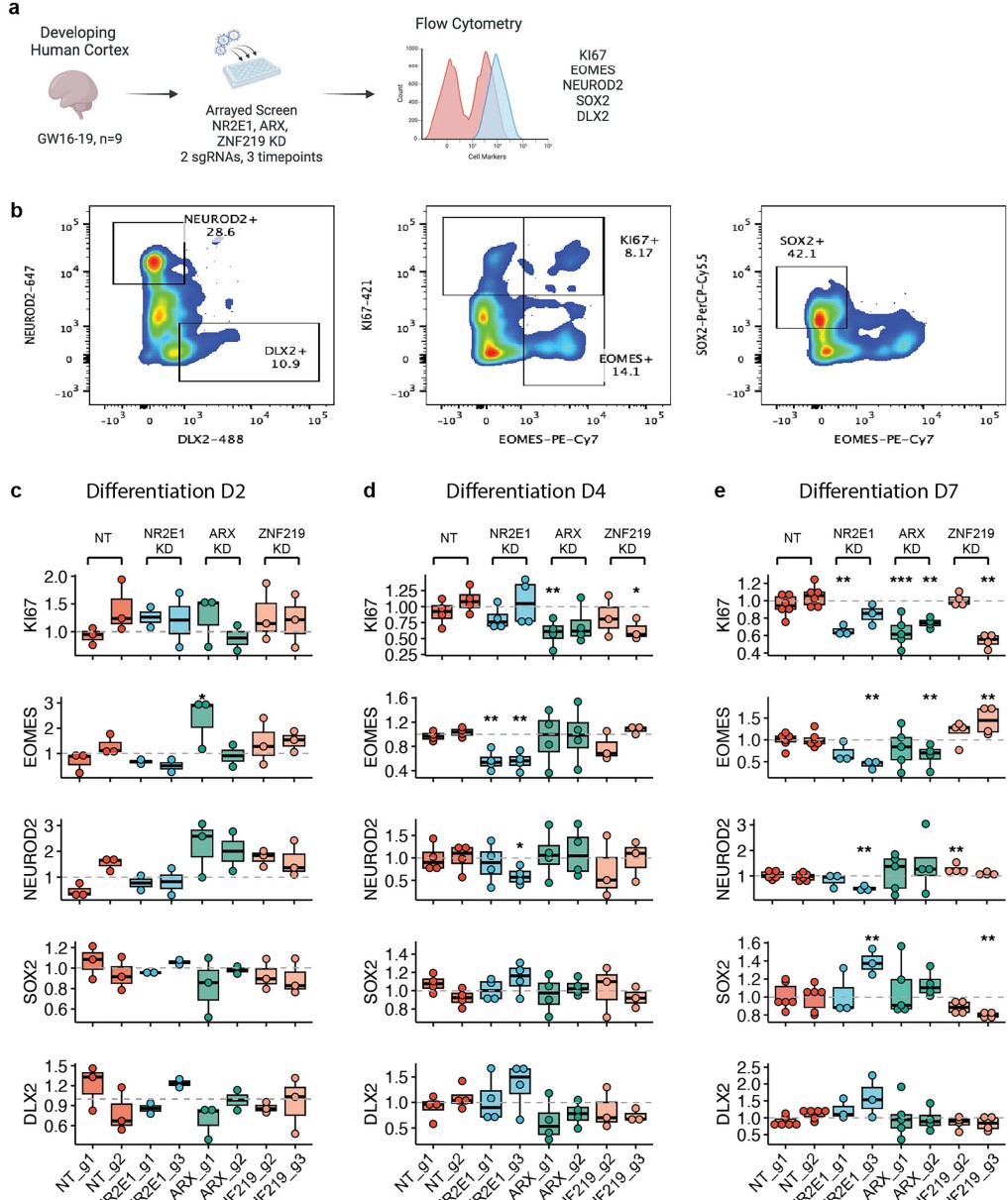

**Extended Data Fig. 5 | Flow cytometry analysis of cell type composition changes with temporal resolution during differentiation. a.** Experimental design of flow cytometry validation of prioritized targets from the Perturb-seq. The workflow described in Fig. 1a was repeated on human cortical RG derived from n = 9 individuals (GW16-19) and fractions of KI67, EOMES, NEUROD2, SOX2 and DLX2 positive populations were quantified in NT, *NR2E1*, *ARX* and *ZNF219* KD populations. **b.** Representative gating strategy for KI67, EOMES, NEUROD2, SOX2 and DLX2 positive populations on NT cells on D7. **c-e.** Barplot showing fold changes in fractions of KI67, EOMES, NEUROD2, SOX2 and DLX2 positive

populations measured flow cytometry in *NR2E1, ARX* and *ZNF219* KD at the sgRNA level relative to NT on differentiation D2 (c, across n = 3 individuals), D4 (d, across n = 4 individuals) and D7 (e, across n = 6 individuals). Two sgRNAs were tested per TF. Dots denote biological replicates; colors indicate target genes. Two-sided Wilcoxon tests on replicate mean against NT after collapsing 2 NT sgRNAs. Centre line, median; box limits, upper and lower quartiles; whiskers, 1.5X interquartile range; dots, biological replicates. *, p < 0.05; **, p < 0.01; ***, p < 0.001. Panel **a** created in BioRender. Nowakowski, T. (https://BioRender.com/7cr6o12).

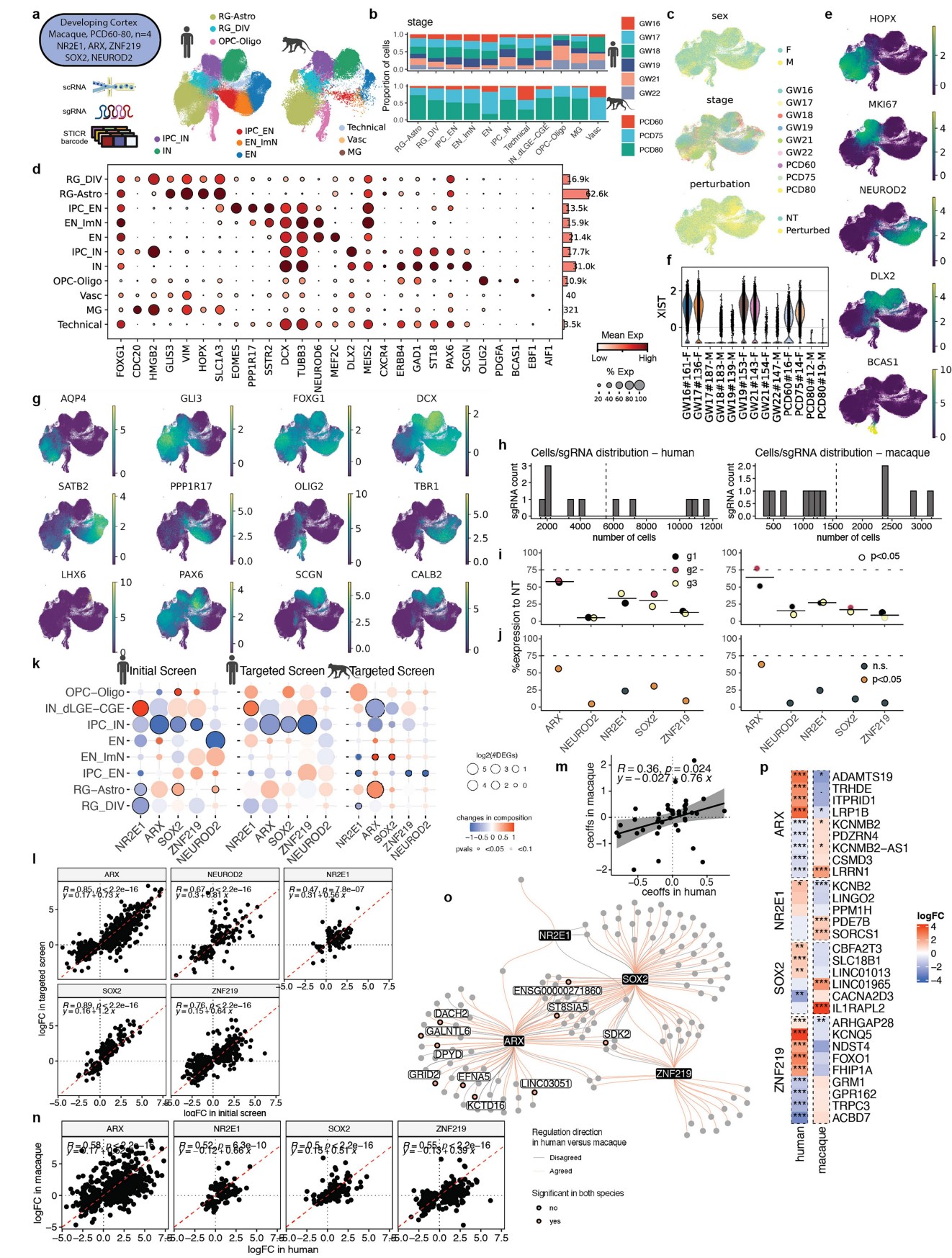

**Extended Data Fig. 6** | See next page for caption.

**Extended Data Fig. 6 | Lineage-resolved targeted Perturb-seq in human and rhesus macaque. a.** Experimental design for combined TF perturbation and lineage tracing in four macaque individuals (left), with UMAPs of lineage-resolved 129,003 human (middle) and 24,381 macaque (right) cells, colored by supervised cell types. A median of 15,469 and 3,013 cells per perturbation were recovered after filtering in human and macaque, respectively. **b.** Stacked barplot showing developmental stage distributions across cell types in the human (top) and macaque (bottom) lineage-resolved dataset. **c.** UMAP of integrated cells from both species, colored by sex, developmental stages and perturbation. **d.** Dotplot showing marker gene expression across annotated clusters (left), with barplots showing cell numbers (right) in both species. Dot size denotes the expressing-cell fraction and color denotes mean expression. **e.** UMAPs showing expression of lineage markers. **f.** Violin plot of $XIST$ expression per individual, supporting inferred sex genotype. **g.** UMAPs showing expression of cell type markers. **h.** Histograms showing distributions of cell numbers per sgRNA in human (left) and macaque (right) targeted libraries. **i.** Dotplot for percentage target gene expression in each sgRNA compared to NT cells grouped by TF target, colored by sgRNA bordered based on significance. $\log_2 FC$ were calculated within each cell type and the lowest value for each sgRNA was used for visualization. **j.** Dotplot showing target gene KD efficiency compared to NT cells in human (left, n = 9) and macaque (right, n = 4) on D7, filled by individual sgRNA with borders highlighting significance. $\log_2 FC$ were calculated with DEseq2 in each cell type and the lowest value for each sgRNA was used for visualization and filtering. **k.** Dotplot showing global changes in cell abundance (color, DCATS coefficient; border, p value) and gene expression (size, number of DEGs, BH-adjusted p < 0.05) across eight cell types in the initial human screen (left), lineage-resolved targeted human (middle) and macaque (right) screen. **l.** Scatterplot of $\log_2 FC$ for DEGs in the initial (X axis) and targeted (Y axis) human screens identified in $ARX$, $NEUROD2$, $NR2E1$, $SOX2$ and $ZNF219$ KD. Pearson correlation coefficients and two-sided p values are shown with regression line equations. Red dashed line represents Y = X. **m.** Scatterplot of estimated coefficients from DCATS in human (X axis) and macaque (Y axis). Pearson correlation coefficients and two-sided p values are shown with regression lines (95% CI) and equations. **n.** Scatterplots of $\log_2 FC$ for DEGs in human and macaque identified in $ARX$, $NR2E1$, $SOX2$ and $ZNF219$ KD. Pearson correlation coefficients and two-sided p values shown with regression line equations. Red dashed line represents Y = X. **o.** Regulatory network plot shows convergent DEGs (defined in Fig. 2) connected to $NR2E1$, $ARX$, $SOX2$, $ZNF219$ in humans compared to macaques. Edge colors denote the conservation of regulation (pink = consistent direction in both species). **p.** Heatmaps showing $\log_2 FC$ in top divergent DEGs in response to $ARX$, $NR2E1$, $SOX2$ and $ZNF219$ KD in human and macaque. Dots or asterisks indicate p values from DEseq2. PCD, post conceptional day. *, p < 0.05; **, p < 0.01; ***, p < 0.001. Panels **a**, **b** and **k** created in BioRender. Nowakowski, T. (https://BioRender.com/7cr6o12).

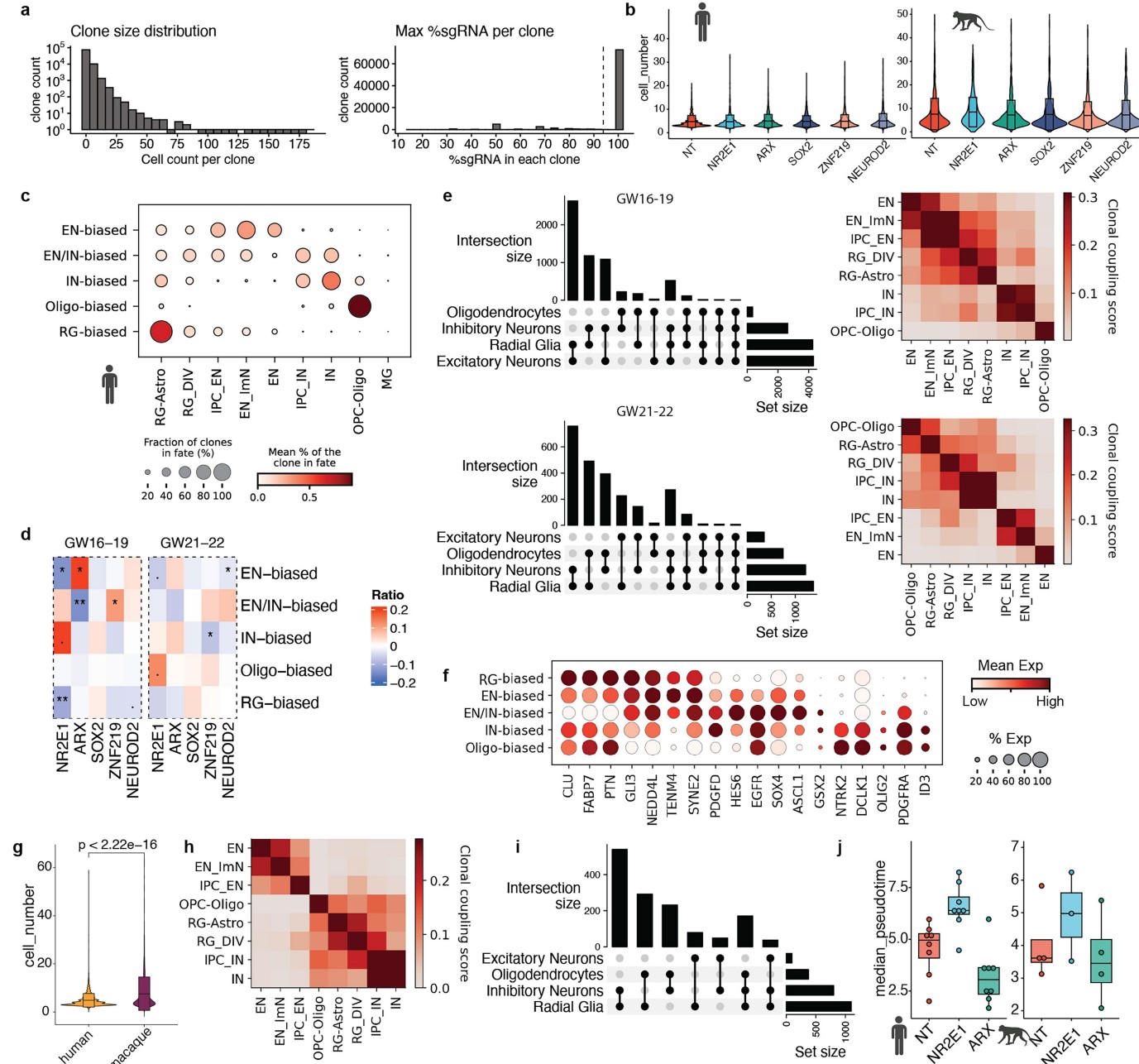

**Extended Data Fig. 7 | Supplemental analysis for RG lineage plasticity and progression. a**. Histograms showing distributions of clone size (top) and maximum percentage of detected sgRNA per clone in cells with assigned sgRNA (bottom) before filtering. Clones with <3 cells or conflicted sgRNA assignments (maximum sgRNA percentage <100%) were excluded from downstream analyses. **b**. Violin plots for clone size distribution in 8937 human clones (left) and 1391 macaque clones (right). Centre line, median; box limits, upper and lower quartiles; whiskers, 1.5X interquartile range. **c**. Dotplot showing distributions of supervised cell types across clonal clusters in human. Dot size denotes the fraction of clones in which a cell type is detected; dot color denotes the mean percentage of cell type in clones. **d**. Heatmap showing changes in clonal cluster fractions under TF perturbations in early (left) and late (right) midgestation. Color indicates fraction changes relative to NT; dots or asterisks denote significance (two-sided paired Wilcoxin test). **e**. Upset plots (left) showing cell class compositions in multi-class clones and heatmaps

(right) of lineage coupling score matrix in human in early (top, GW16-19) and late (bottom, GW21-22) midgestation. **f**. Dotplots of marker gene expression in the human radial glia class (RG-Astro & RG_DIV) grouped by clonal clusters. Dot size denotes the expressing-cell fraction and color denotes mean expression. **g**. Violin plot for clone size distribution in both species. Two-tailed t-test on mean clone sizes (human, n = 8937; macaque, n = 1391 clones). Centre line, median; box limits, upper and lower quartiles; whiskers, 1.5X interquartile range. **h**. Heatmap for lineage coupling score matrix in major cell types identified in macaque. **i**. Upset plot showing cell class compositions in multi-class clones in macaque. **j**. Boxplot showing median pseudotime of all clones in NT, *NR2E1* and *ARX* KD in human (left, n = 8 individuals) and macaque (right, n = 4 individuals). Dots represent biological replicates. Centre line, median; box limits, upper and lower quartiles; whiskers, 1.5X interquartile range. Ctx, cortex; PFC, prefrontal cortex. •, p < 0.1; *, p < 0.05; **, p < 0.01; ***, p < 0.001. Panels **b**, **c** and **j** created in BioRender. Nowakowski, T. (https://BioRender.com/7cr6o12).

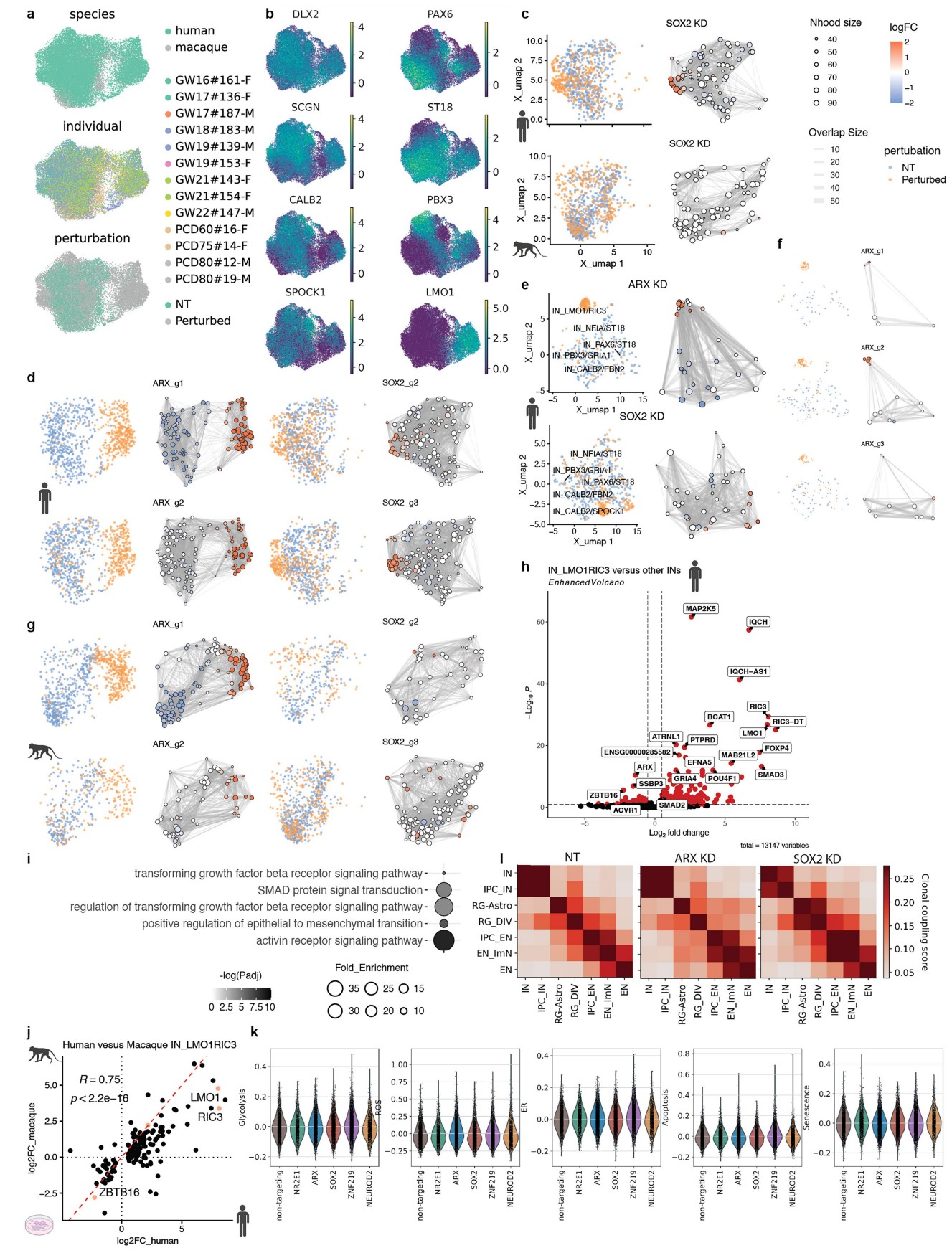

**Extended Data Fig. 8** | See next page for caption.

**Extended Data Fig. 8 | Ectopic IN subtypes in human and rhesus macaque.**
**a**. Integrated UMAPs of lineage-resolved human and macaque INs, colored by species, individual and perturbation. **b**. UMAPs of marker gene expression in INs. **c**. UMAPs showing results of differential abundance testing in INs using Milo in *SOX2* KD at the gene level in human (top) and macaque (bottom). (left) UMAP of downsampled NT and perturbed cells, colored by perturbation condition; (right) Neighborhood graph of differential abundance testing colored by $\log_2$FC. **d,g**. UMAPs showing results of differential abundance testing for *ARX* (left) and *SOX2* (right) KD at the sgRNA level in human (**d**) and macaque (**g**). **e,f**. UMAPs showing results of differential abundance testing for *ARX* (top) and *SOX2* (bottom) KD at the gene level (**e**); and *ARX* KD at the sgRNA level (**f**) in the initial human screen. **h**. Volcano plot for DEGs in human IN_LMO1/RIC3 versus other IN subtypes detected in NT. Genes are colored if BH-adjusted p value < 0.01 and $|\log_2$FC$| > 0.5$. **i**. Dotplot showing GO biological processes enrichment in DEGs identified in human IN_LMO1/RIC3 cluster using pathfindR (one-sided hypergeometric test). Dot color denotes -$\log_{10}$(adjusted p value); dot size denotes fold enrichment per category. **j**. Scatterplot comparing $\log_2$FC of DEGs identified in IN_LMO1/RIC3 between human and macaque. Pearson correlation coefficient and p value are labeled. Red dashed line represents Y = X. **k**. Violin plots of gene scores for glycolysis, ER stress, ROS, apoptosis and senescence in INs under TF perturbations; ARX KD does not increase stress gene scores. **l**. Heatmap showing lineage coupling score matrices in major human cell types in NT (left), *ARX* (middle) and *SOX2* (right) KD. Panels **c**–**e**, **g**, **h** and **j** created in BioRender. Nowakowski, T. (https://BioRender.com/7cr6o12).

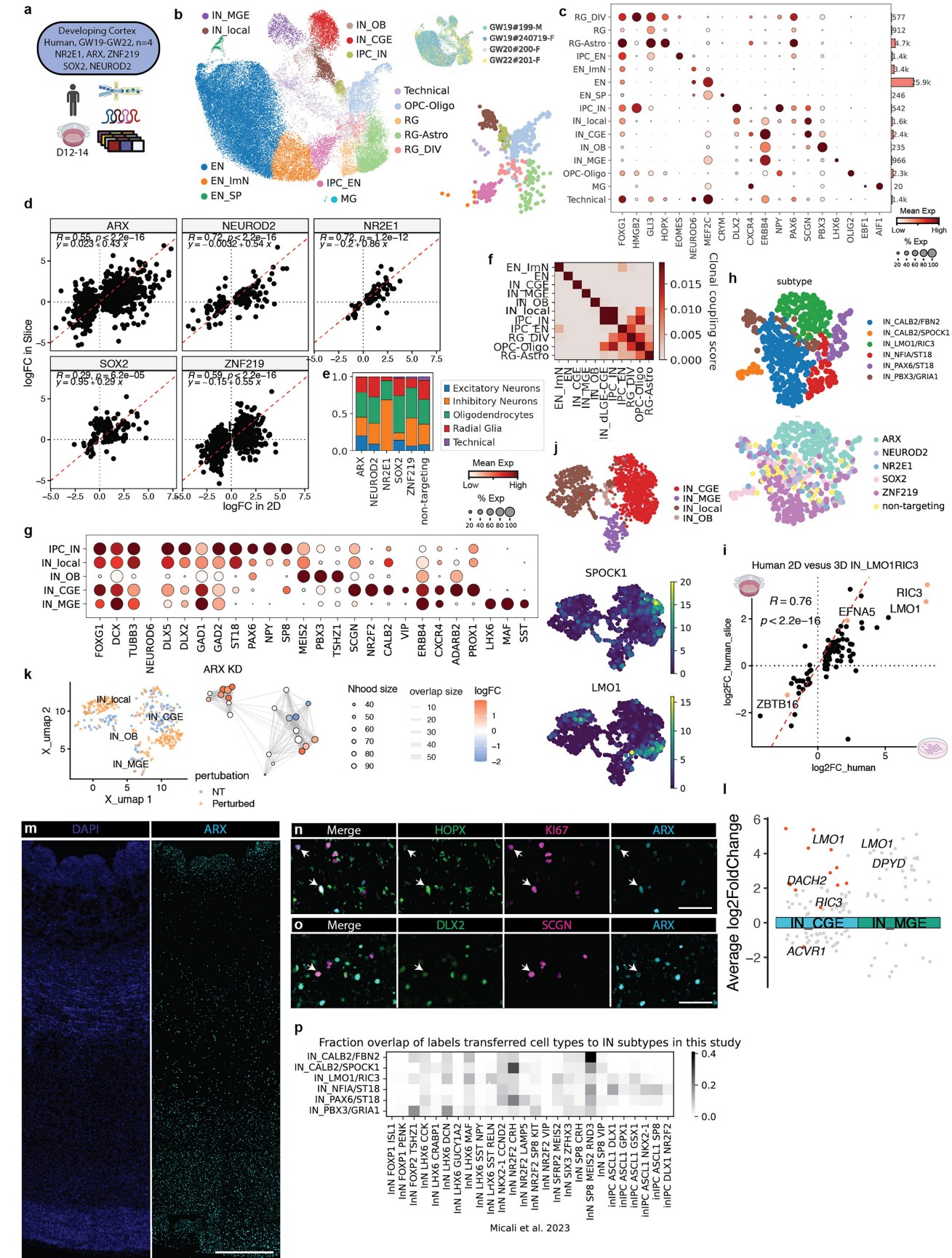

**Extended Data Fig. 9** | See next page for caption.

**Extended Data Fig. 9 | Ectopic IN subtypes in human organotypic slice culture. a**. Experimental design for targeted TF perturbation and lineage tracing in organotypic slice culture in four human individuals. CRISPRi and STICR lentivirus were locally co-injected to the germinal zone to target RG populations. **b**. UMAPs showing cell type annotations and individuals (top right), with a UMAP of newborn populations after filtering for cells from multicellular clones (bottom right). Note that large fractions of postmitotic ENs were labeled but not clonally linked to other cell types. **c**. Dotplot of marker gene expression across annotated clusters (left), with barplot of cell numbers (right). Dot size denotes the expressing-cell fraction and color denotes mean expression. **d**. Scatterplots comparing $\log_2$FC of DEGs identified in *ARX, NR2E1, SOX2* and *ZNF219* KD between human 2D and slice culture. Pearson correlation coefficients and two-sided p values are shown together with regression line equations; red dashed line denotes Y = X; average R = 0.6 across perturbations. **e**. Barplot showing cell class compositions in multicellular clones per perturbation. **f**. Heatmap showing lineage coupling score matrix in major cell types in the RG lineage. IN_local represents locally derived INs clonally linked to RG and ENs. **g**. Dotplot showing marker expression across IN subtypes. **h**. UMAPs of cortical born IN subtypes in IN_local, colored by subtypes and perturbations. **i**. Scatterplot comparing $\log_2$FC of DEGs identified in IN_LMO1/RIC3 between human 2D and slice culture. Pearson correlation coefficient and p value are shown; red dashed line denotes Y = X. **j**. UMAPs of all INs identified in slice culture, including postmitotically labeled GE-derived clusters; colored by supervised clusters (left) and *LMO1* and *SPOCK1* expression (right). **k**. UMAPs showing results of differential abundance testing in INs in *ARX* KD. (left) UMAP of NT and perturbed cells, colored by perturbation condition. (right) Neighborhood graph colored by $\log_2$FC. **l**. Volcano plot of DEGs in *ARX* KD versus NT in GE-derived IN clusters in slice culture from DEseq2. DEGs are shown if $|\log_2$FC $| > 0.2$, colored red if adjusted p < 0.01. **m-o**. Immunohistochemistry in midgestation cortex showing ARX expression (**m**, scale bar 500 μm); colabeling with HOPX and KI67 in subventricular zone (**n**, 50 μm); colabeling with DLX2 and SCGN in cortical plate (**o**, 50 μm); representative images from multiple fields in two individuals. **p**. Heatmap of fraction overlap between IN subtypes identified in the human targeted 2D screen and labels transferred from an in vivo atlas of the developing macaque telencephalon[96]. Panel **a** created in BioRender. Nowakowski, T. (https://BioRender.com/7cr6o12).

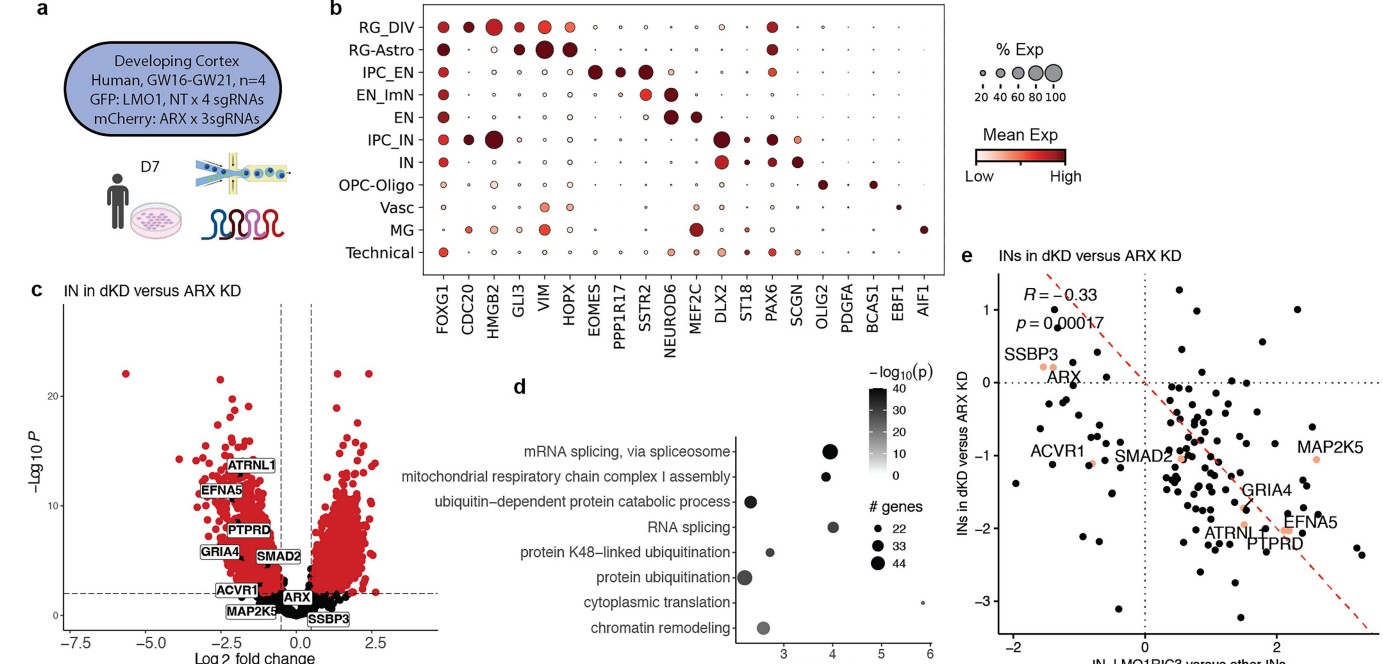

**Extended Data Fig. 10 | Supplemental analysis for *ARX* and *LMO1* double KD.** **a**. Experimental design for *ARX*, *LMO1* dKD in cortical RG of four human individuals. Cells were transduced with a GFP expressing library that includes eight sgRNAs targeting *LMO1* and NT, and an mCherry expressing library with three sgRNAs targeting *ARX*. GFP and mCherry double positive cells were sorted and captured for scRNA-seq. **b**. Dotplot of cell marker expression across annotated clusters. **c**. Volcano plot of DEGs in *ARX, LMO1* dKD versus ARX KD in INs; genes are colored if adjusted p < 0.01 and |log$_2$FC| > 0.5. **d**. Dotplot showing GO biological processes enrichment in DEGs from (**c**) using pathfindR (one-sided hypergeometric test). Color denotes -log$_{10}$(adjusted p value); dot size denotes number of DEGs per category. Enrichment of chromatin related terms show downstream effects of *LMO1* perturbation. **e**. Scatterplot comparing log$_2$FC of DEGs identified in the IN_*LMO1/RIC3* subtype (X axis) and log$_2$FC in INs of dKD versus *ARX* KD (Y axis). Significant DEGs (adjusted p value < 0.05 and |log$_2$FC| > 0.5) in the IN_*LMO1/RIC3* subtype from the targeted human screen are shown. Pearson correlation coefficient and two-sided p value were labeled. Red dashed line denotes Y = -X. An overall negative correlation suggests partial rescue of the ARX-driven transcriptional program in dKD. Panel **a** created in BioRender. Nowakowski, T. (https://BioRender.com/7cr6o12).

# Reporting Summary

## Statistics

For all statistical analyses, confirm that the following items are present in the figure legend, table legend, main text, or Methods section.

| n/a | Confirmed | |
|---|---|---|
| ☐ | ☒ | The exact sample size (*n*) for each experimental group/condition, given as a discrete number and unit of measurement |
| ☐ | ☒ | A statement on whether measurements were taken from distinct samples or whether the same sample was measured repeatedly |
| ☐ | ☒ | The statistical test(s) used AND whether they are one- or two-sided *Only common tests should be described solely by name; describe more complex techniques in the Methods section.* |
| ☐ | ☒ | A description of all covariates tested |
| ☐ | ☒ | A description of any assumptions or corrections, such as tests of normality and adjustment for multiple comparisons |
| ☐ | ☒ | A full description of the statistical parameters including central tendency (e.g. means) or other basic estimates (e.g. regression coefficient) AND variation (e.g. standard deviation) or associated estimates of uncertainty (e.g. confidence intervals) |
| ☐ | ☒ | For null hypothesis testing, the test statistic (e.g. *F*, *t*, *r*) with confidence intervals, effect sizes, degrees of freedom and *P* value noted *Give P values as exact values whenever suitable.* |
| ☒ | ☐ | For Bayesian analysis, information on the choice of priors and Markov chain Monte Carlo settings |
| ☒ | ☐ | For hierarchical and complex designs, identification of the appropriate level for tests and full reporting of outcomes |
| ☐ | ☒ | Estimates of effect sizes (e.g. Cohen's *d*, Pearson's *r*), indicating how they were calculated |

*Our web collection on statistics for biologists contains articles on many of the points above.*

## Software and code

Policy information about availability of computer code

| Data collection | Software used included Thermo Fisher EVOS M7000 Software v2.2.804.158, BD FACSDiva software V9.0, Illumina sequencer control software (NovaSeq) and bcl2fastq (bcl2fastq2) software. |
|---|---|
| Data analysis | Code used for data analysis is available at https://github.com/jding5066/perturbTF and https://github.com/cnk113/NextClone. Analytical details can be found in the Methods section in the manuscript. Packages used to analyze scRNA-seq data include: Cellranger v7.0.2, DRAGEN Single Cell RNA v4.4.5, seacells v0.3.3, SCENIC+ v1.0.1.dev4 +ge4bdd9f, Vireo v0.5.8, Scanpy v1.9.6, SCVI-tools v0.20.0, Pertpy v0.5.0, pyDEseq2 v0.4.10, Milo v2.0.0, scLiTr v0.1.4, Monocle3 v1.3.7, CoSpar v0.3.3, scFates v1.0.9, pathFindR v2.4.1.9001, scVelo v0.3.1, velocyto v0.17.17, DCATS v1.2.0. Image processing was performed using Fiji/ImageJ v2.14.0 and flow data analysis was performed using Flowjo v10.10.0. |

For manuscripts utilizing custom algorithms or software that are central to the research but not yet described in published literature, software must be made available to editors and reviewers. We strongly encourage code deposition in a community repository (e.g. GitHub). See the Nature Portfolio guidelines for submitting code & software for further information.

## Data

Policy information about availability of data

All manuscripts must include a data availability statement. This statement should provide the following information, where applicable:

- Accession codes, unique identifiers, or web links for publicly available datasets
- A description of any restrictions on data availability
- For clinical datasets or third party data, please ensure that the statement adheres to our policy

Raw sequencing data for macaque and processed lineage tracing data for both species are deposited on GEO accession number: GSE284197.
Raw sequencing and processed data for human specimens are available through dbGaP under accession number phs002624.v5.p1.

## Research involving human participants, their data, or biological material

Policy information about studies with human participants or human data. See also policy information about sex, gender (identity/presentation), and sexual orientation and race, ethnicity and racism.

| Reporting on sex and gender | Sex and Gender were not used as selection criteria for sample collection. Sex of de-identified samples were determined based on PCR-based on genotyping and sex-specific gene expression. Male and female samples were treated equally. |
|---|---|
| Reporting on race, ethnicity, or other socially relevant groupings | No race, ethnicity, or other socially relevant groupings were performed in this study. |
| Population characteristics | De-identified human brain samples from gestational week 16 to 23 were used for this study and listed in Supplementary Table 3 and 6. No population characteristics other than age were used in the data analysis. |
| Recruitment | No recruitment criteria were used. De-identified tissue samples were collected from previous patient consent in strict observance of the legal and institutional ethical regulations, which was performed by the clinic. |
| Ethics oversight | Human Gamete, Embryo, and Stem Cell Research Committee (institutional review board) at the University of California, San Francisco. |

Note that full information on the approval of the study protocol must also be provided in the manuscript.

# Field-specific reporting

Please select the one below that is the best fit for your research. If you are not sure, read the appropriate sections before making your selection.

☒ Life sciences      ☐ Behavioural & social sciences      ☐ Ecological, evolutionary & environmental sciences

For a reference copy of the document with all sections, see nature.com/documents/nr-reporting-summary-flat.pdf

# Life sciences study design

All studies must disclose on these points even when the disclosure is negative.

| Sample size | 4 individuals were collected for the initial Perturb-seq. Validation experiments focusing on top candidates were carried out using additional 13 individuals (2D single cell lineage tracing) in both human and macaque, 4 human individuals in organotypic slice culture and 9 individuals (flow cytometry) in human. Data were collected from as many individuals as were available. Downsampling was performed and showed in extended data fig. 3 and 4 to show sufficiency of sample sizes. |
|---|---|
| Data exclusions | Cells assigned to sgRNAs with knockdown efficiency lower than 75% was excluded from this study to ensure recovery of knockdown phenotypes. For lineage tracing analysis, clones lower than 3 cells and clones with conflicted sgRNA assignments were excluded from downstream analysis to consider multicellular clones labeled at the progenitor stage and expanded during differentiation to study effects of TF perturbations on progenitor lineage. |
| Replication | In order to improve replicability, samples for initial Perturb-seq were collected from 4 different individuals spanning from gestational week (GW)16-18. For lineage resolved targeted screen, 13 human individuals(GW17-22) and 4 macaque (PCD60-80) individuals and organotypic slice culture were used to ensure reproducibility of findings from the initial screen. Furthermore, flow cytometry was used to corroborate findings from different differentiation timepoints across 9 human individuals. Phenotypes of ARX, NR2E1 and ZNF219 KD reported in the study were consistently observed across batches, species and modalities. |
| Randomization | Randomization was not relevant to this study. Samples were pooled and treated equally between perturbation conditions. |
| Blinding | Blinding was not relevant to this study. Data derived from pooled screens were analyzed based on  on unbiased computational pipeline. |

# Reporting for specific materials, systems and methods

We require information from authors about some types of materials, experimental systems and methods used in many studies. Here, indicate whether each material, system or method listed is relevant to your study. If you are not sure if a list item applies to your research, read the appropriate section before selecting a response.

## Materials & experimental systems

| n/a | Involved in the study |
|---|---|
| ☐ | ☒ Antibodies |
| ☐ | ☒ Eukaryotic cell lines |
| ☒ | ☐ Palaeontology and archaeology |
| ☒ | ☐ Animals and other organisms |
| ☒ | ☐ Clinical data |
| ☒ | ☐ Dual use research of concern |
| ☒ | ☐ Plants |

## Methods

| n/a | Involved in the study |
|---|---|
| ☒ | ☐ ChIP-seq |
| ☐ | ☒ Flow cytometry |
| ☒ | ☐ MRI-based neuroimaging |

## Antibodies

| Antibodies used | mouse-EOMES (Thermofisher, 14-4877-82, 2288573)<br>rabbit-NEUROD2 (Abcam, ab104430, GR3414328-2)<br>goat-SOX9 (R&D, AF3075, WIL0421041)<br>mouse-DLX2 (Santa Cruz, sc-393879, C0424)<br>mouse-HOPX (Santa Cruz, sc-398703, C0823)<br>mouse-KI67 (BD, 550609, 2052205)<br>sheep-ARX (R&D, AF7068SP, CFOM0225031)<br>rabbit-SCGN (Millipore-sigma, HPA006641)<br>rabbit-KI67 (Vector, VP-K451)<br>KI67-421 (BD, 565929, 4282091)<br>SOX2-PerCP-Cy5.5 (BD, 561506, 3313075)<br>EOMES-PE-Cy7 (Invitrogen, 25-4877-42, 2510765)<br>donkey anti-mouse-488(Thermo, A32766, WF319853)<br>donkey anti-rabbit-647 (Thermo, A32795, WA308388)<br>donkey anti-sheep-647 (Thermo, A21448, 2045339)<br>donkey anti rabbit 555 (Thermo, A32794, WG322207) |
|---|---|
| Validation | All primary antibodies used in this study have been validated by the manufacturer to be suitable for the respective application (ICC, IHC and Flow) against human protein. Antibodies used for ICC and IHC were tested in fixed human tissue and showed expected anatomical distribution and subcellular localization. No off-target staining was observed.<br>All secondary antibodies were pre-adsorbed to minimize cross-reactivity and validated by the manufacturer. No reaction was observed against serum proteins of other species. |

## Eukaryotic cell lines

Policy information about cell lines and Sex and Gender in Research

| Cell line source(s) | Lenti-X HEK293T (Takara Bio, 632180) were used for lentiviral production.<br>Primary cell culture from a total of 18 humans (8 females, 5 males, 5 unknown) and 4 macaques (2 female and 2 males) were derived from cryopreserved tissue chunks from the developing cortex were used in this study. Organogtypic slice culture were derived from 4 humans (1 female and 3 males). De-identified human tissue samples were collected with previous patient consent in strict observance of the legal and institutional ethical regulations. Protocols were approved by the Human Gamete, Embryo, and Stem Cell Research Committee (institutional review board) at the University of California, San Francisco. The Primate Center at the University of California, Davis, provided 4 specimens of cortical tissue from PCD60 (n=1), PCD75 (n=1) and PCD80 (n=2) macaques. All animal procedures conformed to the requirements of the Animal Welfare Act, and protocols were approved before implementation by the Institutional Animal Care and Use Committee at the University of California, Davis. |
|---|---|
| Authentication | HEK293T cells were authenticated by the vendor (Takara Bio) and morphology. |
| Mycoplasma contamination | HEK293T and primary cell culture was tested and found negative for mycoplasma infection. |
| Commonly misidentified lines<br>(See ICLAC register) | No commonly misidentified cell lines were used in the study. |

# Plants

| | |
|---|---|
| Seed stocks | *Report on the source of all seed stocks or other plant material used. If applicable, state the seed stock centre and catalogue number. If plant specimens were collected from the field, describe the collection location, date and sampling procedures.* |
| Novel plant genotypes | *Describe the methods by which all novel plant genotypes were produced. This includes those generated by transgenic approaches, gene editing, chemical/radiation-based mutagenesis and hybridization. For transgenic lines, describe the transformation method, the number of independent lines analyzed and the generation upon which experiments were performed. For gene-edited lines, describe the editor used, the endogenous sequence targeted for editing, the targeting guide RNA sequence (if applicable) and how the editor was applied.* |
| Authentication | *Describe any authentication procedures for each seed stock used or novel genotype generated. Describe any experiments used to assess the effect of a mutation and, where applicable, how potential secondary effects (e.g. second site T-DNA insertions, mosiacism, off-target gene editing) were examined.* |

# Flow Cytometry

## Plots

Confirm that:

☒ The axis labels state the marker and fluorochrome used (e.g. CD4-FITC).

☒ The axis scales are clearly visible. Include numbers along axes only for bottom left plot of group (a 'group' is an analysis of identical markers).

☐ All plots are contour plots with outliers or pseudocolor plots.

☒ A numerical value for number of cells or percentage (with statistics) is provided.

## Methodology

| | |
|---|---|
| Sample preparation | Cryopreserved tissue chunks of developing human cortex were dissociated with papain (Worthington Biochemical Corporation, LK003153) and plated for culture at a density of 500,000 cells/cm2. Cells were expanded for 12 days, transduced with all-in-one CRISPRi vector on day 5 and differentiated for 7 days.Cell culture was dissociated with Accutase (STEMCELL Technologies, 07920) supplemented with 5% Trehalose (Fisher Scientific, BP268710) and fixed in Foxp3 fixation buffer for 30 min at room temperature. Cells were then washed twice with Foxp3 permeabilization buffer and then stained with primary antibodies mouse-DLX2 (Santa Cruz Biotechnology, sc-393879) and rabbit-NEUROD2 (Abcam, ab104430) at 1:100 dilution. After washing with Foxp3 permeabilization buffer, cells were stained with secondary antibodies donkey anti-mouse-488 and donkey anti-rabbit-647 (Invitrogen) at 1:200. Finally, cells were stained with conjugated antibodies KI67-421 (BD, 562899), SOX2-PerCP-Cy5.5 (BD, 561506), EOMES-PE-Cy7 (Invitrogen, 25-4877-42) at 1:100, washed twice with Foxp3 permeabilization buffer and resuspended in 0.2% BSA in PBS. |
| Instrument | BD LSRFortessa |
| Software | BD FACSDiva was used for data collection and Flowjo was used for analysis. |
| Cell population abundance | Cells were stained for nuclear markers KI67-421, SOX2-PerCP-Cy5.5, DLX2-488, NEUROD2-647 and EOMES-PE-Cy7. Populations include heterogeneous progenies derived from cortical radial glia which varies between individual, stage, differentiation timepoints and knockdown conditions. |
| Gating strategy | FSC/SSC values were used to filter debris and doublets, resulted singlets were used for downstream gating. Positive and negative populations for each marker were identified based on the bimodal distribution of marker expression. |

☒ Tick this box to confirm that a figure exemplifying the gating strategy is provided in the Supplementary Information.

