## [Peer Review File · Nature]

Dissecting Gene Regulatory Networks Governing Human Cortical Cell Fate

Corresponding Author: Dr Alex Pollen

Version 0:

Reviewer comments:

Referee #1

(Remarks to the Author)

Ding et al. constructed TF-enhancer-gene eGRNs to identify 44 TFs predicted to have strong impacts on transcriptomes when perturbed. Using CRISPRi knockdown in human primary radial glia (RG) cultures, they studied the transcriptional and cell fate consequences of these TFs in cortical neurogenesis. Their findings reveal multiple TFs that bias differentiation between excitatory neuron and interneuron lineages. Additionally, they identified effector genes that were convergently perturbed across multiple TF knockdowns, highlighting potential roles in neurodevelopmental disorders. While this work provides insights and novelties of a human RG culture system, concerns remain regarding the model validity and power sufficiency. The 7-day RG culture generates small clones (~4.6 cells), considerably lower than the estimated naïve clone size (~15 cells). This discrepancy raises questions about whether the culture accurately models in vivo conditions and perturbation relevance.

Additionally, weak statistical significance in several analyses warrants further examinations (e.g., DEGs are not significant using traditional tools like DEseq2; the tool they used, TRADE, does not account for statistical dispersion; TRADE specifies that null hypothesis testing which the authors omitted and may have inflated the pvalues; several analyses used a lenient FDR < 0.1 cutoff).

Major:

1. One of the major novelty is the use of primary RG cell culture – but the validity remains additional evidence. Similar screens (at much higher scale) has been done by iPSC-derived NPCs and neurons (Tian Nat Neuro 2021): what are the unique advantages of this RG system?
 - o The cell types in the RG culture appear significantly different from naïve human cortex. Pearson correlations in Fig. S1e are weak ($r \approx 0.2-0.4$), suggesting substantial divergence from naïve, in-vivo states. A differential expression analysis comparing this culture with naïve cortical tissue would help clarify these differences? How different are these cells from naïve states, and are they too different to be useful for perturbation studies?
 - o The clone size in STICR experiments (~4.6 cells) is much smaller than expected for naïve RG, which is concerning. How might this affect differentiation potential and phenotypic robustness?
 - o The dominance of interneuron populations also raise concerns about potential subpallial contamination during dissection or artifacts from RG dissociation. How do the dorsal INs compare with other IN populations (LGE/CGE-derived INs) in gene expression and relative composition? How do they align with existing data (e.g., Delgado et al.)?
 - o How does this culture compare to iPSC-derived NPCs or neurons, where Perturb-seq has been widely applied? Does it reveal phenotypes that iNs cannot?
 - o Additional validations, such as tissue-level imaging, donor-to-donor reproducibility, and assessments of culture health, would bolster confidence in this system.
2. Statistical power and sampling size in the study is another concern. The study's design (400~500 cells/gene; Fig. S1i) translates to ~56 cells/perturbation/cell type across eight cell types—considerably lower than recent screens (e.g., Replogle et al., Cell, 2022; ~200 cells in a genome-wide screen; Li et al., Nature, 2023; ~80 cells per perturbation per cell type in organoids). Given this low sample size, can the authors justify the screen's statistical power? How are the effect observed across batches/replicates? Perturb-seq downsampling analysis is a routine – is the current cell number well powered to

reach conclusions? Additional replicates or cell numbers may improve reproducibility.

3. I am concerned about the DEG statistical tests for several reasons, as the authors used a somewhat unusual pipeline:
 - o For DESeq, the authors only check that more than 1 cell is has the perturbed condition within the cell type they are looking for – this is a very low cell number threshold..
 - o The authors then send a few outputs to TRADE: only taking in the un-adjusted P-value and does its own FDR and Bonferroni correction, but this does not account for statistical dispersion? It also cannot account for the low statistical power that has been already introduced.
 - o All the figure panels are made with the TRADE output, and not the DESeq output. I wonder how many statistically relevant genes that DESeq identifies? If at all, do they overlap with the TRADE analysis?
 - o TRADE itself specifies that null hypothesis testing should be done with a jackknife sterr method, which involves rerunning the pipeline many times with 1 batch leave-out. The authors don't seem to do that anywhere in their code though, and this may lead to inflated statistical values.
4. Flow cytometry validation (Fig. S6) seems contradictory to the Perturb-seq results. Several single-cell phenotypes fail to replicate in flow assays. Notably, DLX2 perturbation does not alter interneuron composition, and Neurod2 excitatory neuron phenotype is inconsistent with single-cell data. Can the authors address these discrepancies?
5. The experimentally measured TF impacts vs. eGRN predictions are inconsistent: the 44 TFs were selected for their predicted strong influence on transcription, yet the observed effects (Fig. 2b) appear weaker than expected. Could this be due to insufficient statistical power or the culture system failing to replicate naïve tissue states (where the eGRNs were derived)?
6. The study reports divergent phenotypic effects between human and macaque cultures. Could these differences stem from technical factors (e.g., RG survival biases due to culture conditions or cryopreservation conditions, or general age differences in donor tissues)? Testing key hits (e.g., NR2E1, ARX) across multiple developmental stages may help clarify whether these effects are age-dependent.
7. gRNA Variability is another issue. Given the observed variability in gRNA effects, per-guide analyses would be beneficial. Do different guides targeting the same TF yield consistent phenotypes?

Minor:

- The 44 TFs were selected based on their broad transcriptional impact but are not necessarily causal drivers of differentiation. Pseudotime trajectory analyses could provide insights into their effects on differentiation dynamics beyond discrete cell-type compositions.
- The cross-species comparisons in Fig. 4 are not entirely convincing. Some patterns appear conserved, while others do not. Statistical correlation analyses would help support the conservation hypothesis.
- Lines 933-935: What method was used for in silico perturbation, and was it validated against in-vitro or in-vivo datasets?

Referee #2

(Remarks to the Author)

In the manuscript entitled Dissecting Gene Regulatory Networks Governing Human Cortical Cell Fate, the authors investigate the transcription factor (TF) circuits that regulate human cortical neurogenesis. Using single-cell CRISPRi screening in a human primary culture system, they examine the functional roles of 44 TFs in determining radial glia (RG) lineage progression. Their findings reveal key regulators such as NR2E1, ARX, and ZNF219, which influence the balance between excitatory and inhibitory neuron (EN/IN) differentiation. They also identify effector genes implicated in neurodevelopmental disorders and demonstrate conserved mechanisms of RG lineage plasticity across primates. This study offers a framework for dissecting gene regulatory networks underlying human cortical cell fate decisions.

The manuscript presents a solid study with several strengths, including the use of valuable human fetal cells to investigate authentic gene regulatory networks controlling human cortical specification. The combination of cutting-edge technologies, such as Perturb-seq and lineage barcoding, provides important insights into the developmental origins and lineage plasticity of cortical cell types. Additionally, the cross-species comparison between human and macaque offers an interesting perspective on the conservation of lineage plasticity.

However, the study lacks the level of novelty required for publication in Nature for the following reasons:

1. Limited technological innovation – While the Perturb-seq approach is a powerful tool, its application has already been demonstrated in various biological contexts. In this study, the reviewer does not see significant methodological advancements beyond what has already been published. It should also be noted that number of genes targeted (44) is relatively limited by today's standards. Without major technical improvements or novel adaptations, the study does not provide a sufficient technological breakthrough to justify publication in a high-impact journal.
2. Moderate biological impact – The findings are not sufficiently groundbreaking, as out of the 44 TFs analyzed, only ZNF219 is identified as a novel regulator of human corticogenesis. The roles of NR2E1 and ARX in cortical development have already been established, and while the authors provide additional details on lineage specification and inhibitory neuron subtype regulation, these findings do not represent a major conceptual leap in the field without additional follow up work on the mechanism of action.
3. Lack of internal controls – The study would benefit from including perturbations of well-characterized TFs known to regulate human cortical cell fate. This would serve as an internal benchmark to validate the experimental system and provide a clearer functional framework for interpreting the role and timing of the novel TFs being investigated.

Minor comments include:

- ARX and SOX2 Perturbations (line 183-): The formation of new clusters (“ectopic”) following ARX and SOX2 knockdown is interesting, but the characterization of these clusters is limited. It would strengthen the paper to discuss whether these ectopic clusters have any known in vivo counterparts or if they represent entirely novel cell states. Additional marker

analysis could help clarify this.

- Lineage Tracing Experiment: While the lineage tracing approach is technically impressive, its contribution to the overall conclusions of the paper feels limited. Consider elaborating on how this experiment adds new insights beyond what is already provided by the perturbation and transcriptional analyses.

- Gestational Week (GW) Sample Variation: The inclusion of samples across a broad range of gestational weeks provides developmental context, but the potential impact of this variation on the results isn't fully addressed. It would be helpful to discuss whether differences in cell composition or gene expression could be influenced by developmental stage, and how this might affect the interpretation of TF perturbation outcomes.

Overall, while the study is rigorous and contributes valuable data to the field, its lack of significant novelty limits its suitability for Nature journal.

Referee #3

(Remarks to the Author)

In humans, most cortical interneurons (INs) originate from the ganglionic eminences in the ventral forebrain and migrate to the cortex. However, recent work has identified a small but significant population of locally generated interneurons that arise in the dorsal telencephalon alongside excitatory neurons (ENs). In this study, Ding and colleagues show that these interneurons are generated from the same progenitors that give rise to excitatory neurons in the developing cortex. This suggests a broader neurogenic potential of radial glial cells (RGs) than previously thought. The authors here screened 44 transcription factors to determine the molecular program to generate ENs or INs and found that NR2E1, ARX, SOX2, and ZNF219 play an important role. The presence of cortical-generated INs has been demonstrated in humans, but here they show that this can be extended to other primates. Overall, this study improves our understanding of the regulatory networks that control cell fate decisions during human neurogenesis and suggests that this may also play an important role in neurodevelopmental processes and disorders.

Here are my suggestions to strengthen this study:

Selection of transcription factors: I understand how the TFs were initially selected, but I'm a bit puzzled because some very well-known, important, TFs are missing, e.g. PAX6, GLI3, HOPX, EOMES. On the contrary, some only neuronal TFs (TBR1, SATB2) are included while the fate of RGs is investigated.

Model system: I'm slightly surprised by the 2D approach when we now have many ways to look at these events in 3D. For example, why not use human fetal slices directly? Alternatively, the authors could use the fetal brain organoid model (FeBO) recently developed by the Artegiani lab. From their data, it appears that identity and developmental stages are maintained when FeBOs are grown in 3D. In addition, the authors should provide evidence for the precise identity of these cells in 2D (I find it hard to find these data, day0 cells are very few in the single cell RNA-seq in Fig. 1B). What is the exact identity of these cells? How can they specifically show that they are only cortical and that there are no ventral RGs?

Timing: It would be nice to have a rationale for selecting the fetal time period that the authors decided to start with. Is this the time when the INs are generated? Why are so many INs generated at the end? Shouldn't they be a tiny fraction of the general neuronal population?

Virus selection: Lentiviruses infect all cells, including differentiated cells. The authors use a protocol enriching for progenitors, but this does not exclude the presence of neurons. Therefore, it would be important to show the % of cells in the progenitor state, and the % of differentiated cells (ENs and INs) with some IHC. In addition, the use of a retrovirus could be a better choice, because it excludes infection of postmitotic cells since it only infects dividing cells.

At D0, the authors use HOPX and KI67 to define cycling RGs. why HOPX? why not PAX6? PAX6 should define RGs and HOPX bRGs according to previous papers by the same authors. Some cell clusters are not clear to me. Why do IPCs express TBR1? Why is PAX6 in INs? Some cells are not FOXP1, what are they?

The authors nicely link the TFs to neurodevelopmental disorders but have not explored this more in-depth. For example, recent work has focused attention on ASD and hypothesized that E/I imbalance is critical. It would be nice to test this hypothesis in a disease context using dorsally patterned cerebral organoids derived from ASD patients.

The authors show that more than 25% DEGs are common for more than 1 TFs. This is very intriguing and could be extended a bit with GO and validated in some way, e.g. IHC...

The perturbation results are interesting as well as unexpected as it seems that so many of these TFs regulate the fate of INs. Interesting because in the cortex these are only a very tiny fraction and therefore it is perplexing that this population is so highly orchestrated by so many TFs. However, since the authors focus on ARX, NR2E1, and ZNF219, they could expand and show the expression of these TFs at the protein level. IHC and double staining with some of the known TFs (PAX6, HOPX, DLX...) would strengthen these findings.

KD of ARX results in IN with GE identity. The authors may want to follow up on this point and compare the ARX INs with LMO GE-derived INs. Perhaps KD of these (LMO, RIC3) genes would allow to define their function in this subtype.

Lineage: the same progenitors can give rise to excitatory and inhibitory neurons in the developing cortex. This is very intriguing and the authors should confirm these results with live imaging. Ideally, this should be performed in a 3D human slice by infecting a RG and following up divisions and fate.

How can the authors be sure they are infecting a single cell and not multiple cells with the same virus? how many cells are infected with more than 1 viral particle (different virus)? can they check after a few hours?

Clonal size is also an issue without live imaging: how can the authors control for clone size? Some cell types could be more or less sensitive to isolation and the clones could be larger.

Evolution: considering the proportion of EN/IN cortex-derived neurons, isn't 47% unexpectedly high as a common origin of ENs and INs in macaques?

The human-macaque-specific responses to TFs (ARX and ZNF219) are very interesting. It would be nice to functionally prove the function of these TFs in evolution.

Version 2:

Reviewer comments:

Referee #1

(Remarks to the Author)

I appreciate the authors for their sincere responses to our questions. In the revised manuscript, the authors compared their system with other human models, strengthening their claim of higher level of fidelity and cell health in their system. Additionally, the authors performed further statistical analysis on the data, including the comparisons to DESeq2 results, phenotypic reproducibility across replicates and downsampling experiments that demonstrated conserved patterns in the data.

The novelty of their culture system and the biological discovery, yet, remain limited. It is already well established that neural progenitor cells (NPCs) can be derived from ES or iPSCs, a routine practice to form 3D organoids (Li et al., *Development*, 2009; Duan et al., *Stem Cells Translational Medicine*, 2015; Gunhanlar et al., *Molecular Psychiatry*, 2017; Scholz et al., *Frontiers in Molecular Biosciences*, 2022). NPCs derived from different ESC lines are also known to exhibit radial glia signatures as well as the heterogeneity of radial glia – so the novelty point of this culture system is likely overstated (Duan et al., *Stem Cells Translational Medicine*, 2015; Luciani et al., *Nature Communications*, 2024). By inducing the CRISPR editing system at either the ESC/iPSC/NPCs, it is already possible to investigate gene functions during early developmental stages, such as the radial glia.

Furthermore, the roles of NR2E1 and ARX in neurodevelopment have been extensively characterized in the last two decades (Roy et al. *The Journal of Neuroscience*, 2004; Friocourt et al., *The Journal of Neuroscience*, 2008; Colasante et al., *Cerebral Cortex*, 2013; Joseph et al., *iScience*, 2021; Kandel et al., *PNAS*, 2022); and LMO1 has been shown to be directly repressed by ARX in interneuron development (Fulp et al., *Human Molecular Genetics*, 2008).

Referee #2

(Remarks to the Author)

The authors have addressed my comments and concerns. In addition to reinforcing the technological novelty of their CRISPRi screening platform, they have now significantly enriched the manuscript with compelling biological findings. These additions greatly enhance the depth and relevance of the study, transitioning it from a primarily methodological contribution to one that also provides meaningful insights into human neurogenesis.

The integration of functional perturbations, lineage tracing, and cross-species validation adds considerable strength to the biological conclusions. I find that the manuscript now offers a well-rounded and impactful contribution, combining technical innovation with mechanistic discovery.

Referee #3

(Remarks to the Author)

The authors carefully addressed all the points I raised in my initial assessment.

We are grateful for constructive feedback from reviewers and provide a possible revision plan below supported by new data generated and analyzed during review. To summarize key aspects of novelty ahead of the point-by-point response:

The **technological novelty** of our system enables dissection of mechanisms underlying cell cycle dynamics and cell fate choice not possible with existing human cortical CRISPR systems by:

- **Increasing the fidelity** to normal cortical development, with more precise cell type homologies than cortical organoids (**Review Figure 1A**), reduced metabolic stress and ER stress (**Review Figure 1B**).
- **Improving the reproducibility of cortical patterning**, with the samples we use having been patterned 16-22 weeks in vivo, compared with organoids enduring long-term stress and showing variable patterning and less than 50% cortical patterning (Fleck et al. 2022) and induced neurons showing less than 50% CNS patterning with cells best resembling PNS neurons (**Review Figure 1C-D**).
- **Improving the scalability across individuals**, with 21 samples across 2 species and 3D organotypic slice culture model now included with our new experiments (**Review Figure 1E**)
- **Enabling sensitive detection of cell fate consequences**, with a week-long RG expansion phase allowing for CRISPR activity at the progenitor stage rather than manifesting in cells differentiated to various stages (**Review Figure 1F**) We now show our 2D system further recapitulates gene expression consequences of TF knockdown in gold standard human organotypic slice (**Figure 1G-H**), while also uniquely providing the RG expansion point for localizing gene repression to RG.
- **Increasing temporal precisions**, with organoids radial glia (RG) showing variable maturation stages in the same organoid (Pollen et al., 2019; Bhaduri et al., 2020), while induced neurons short circuit physiological RG (and normal cortical cell types) altogether. In contrast, our system allows us to survey defined in vivo maturation stages and recapitulates a known sequence of temporal cell fate decisions at the individual clone level (**Review Figure 1I-J**)

These technological advances enable several **biologically novel findings**. In addition to our findings about the increase connectivity of neurodevelopmental disease-linked genes (hub effector concept) and of a critical window for engineering EN/IN plasticity, relevant for disorders driven by E/I imbalance, we plan to expand on two concepts, supported by new data from the review.

- **TFs regulating heterochrony:** We surveyed additional stages of maturation (9 total samples spanning 6 weeks) combining perturb-seq with lineage tracing. Joint profiling of gene expression and lineage allowed us to reconstruct sequential stages of RG maturation and cell fate choice recapitulating a known sequence of development events. Combining with perturbation data revealed that ARX and NR2E1 have opposing roles in regulating the developmental tempo and cell fate choice in the human cortex. In both human and rhesus, ARX knockdown consistently transitioned RG to more immature lineage potentials while NR2E1 consistently transitioned NR2E1 to more mature stages (**Figure 1I-J**). We plan to add an updated Figure 4 focused on the biological theme of conserved TF mechanisms regulating heterochrony of cell fate choice revealed by the study.
- **TFs safeguarding IN Identity:** Within the IN lineage, we discovered a conserved role of ARX and SOX2 in safeguarding normal post-mitotic IN identity. Our findings add a novel cellular mechanism to ARX-linked lissencephaly and epilepsy by revealing that in addition to impaired migration, inhibitory neurons with reduced ARX adopt an ectopic cell state – conserved across species – and by providing a set of the earliest dysregulated genes in human primary cells rather than mouse models.
- We have now confirmed this ectopic state arises in organotypic slice culture (**Figure 1H**). To buttress the finding that these stable and conserved ectopic states exist outside of normal morphospace, we will: 1) mine existing mammalian brain atlases for any signs of these dysregulated states in normal cell types, and 2) examine using dual gene repression, whether this ectopic cell state is mediated by LMO1, a TF co-factor displaying strong and conserved upregulation. We plan to add an updated Figure 5 focusing on the biological theme of TFs safeguarding post-mitotic inhibitory neuron identity.

E. Scale across individuals

H. Conserved ectopic state

I. Sequential lineage changes with age

J. Opposing Effects on maturation by ARX and NR2E1

Review Figure 1: Technological and biological novelty of this study.

A. Pearson correlation of top 25 cell marker gene expression defined in Wang 2025 in vivo atlas using data in this study (left) and in He 2024 organoid atlas (right), with a boxplot showing quantitative differences between the two systems.

B. Violin plots of scores of glycolysis and response to ER stress in in vivo human atlas, 3D organotypic slice culture, 2D primary RG culture, iNeurons and organoids obtained from published studies.

C. (reproduced from Fleck et al., 2022, Fig. 1d) with example patterning variation.

D. Left (reproduced from Lin et al., 2021, Fig. 3C), highlighting that most iNeurons across individuals best correspond to the peripheral nervous system and retina with partial brain correspondence; Right (Lin et al., Fig. 3E) show that induced neurons and organoids show a substantial fraction of non CNS cells.

E. Biological replicates included in this study. New samples added during review are highlighted in blue, bringing the total from 12 to 21 individuals (and 2 species) across experiments coupling CRISPRi and scRNA-seq, compared with typically one individual in organoid and induced neuron studies.

F. Schematics showing timeline of cell differentiation and CRISPRi in 2D and 3D organotypic slice culture.

G-H. Scatterplot for correlation of log fold change of differentially expressed genes under perturbations (H) and in ectopic IN subcluster in ARX knockdown (I) identified in 2D and 3D organotypic slice culture.

I. UMAPs of clones colored by pseudotime and fate biases,

J. Top: Heatmaps showing changes in clonal compositions under perturbations in early and late mid-gestation, highlighting critical window for EN/IN fate plasticity before GW19; Bottom: Ridge plots of pseudotime distribution in clones grouped by stages and perturbation, highlight opposing roles of ARX and NR2E1 in regulating maturation of cell fate potential, with conserved properties in rhesus macaque.

Point-by-point:

Referee #1 (Remarks to the Author):

Referee 1's concerns are mainly due to misunderstandings about experimental clone size, statistics, and fidelity that are addressed below.

Ding et al. constructed TF-enhancer-gene eGRNs to identify 44 TFs predicted to have strong impacts on transcriptomes when perturbed. Using CRISPRi knockdown in human primary radial glia (RG) cultures, they studied the transcriptional and cell fate consequences of these TFs in cortical neurogenesis. Their findings reveal multiple TFs that bias differentiation between excitatory neuron and interneuron lineages. Additionally, they identified effector genes that were convergently perturbed across multiple TF knockdowns, highlighting potential roles in neurodevelopmental disorders.

While this work provides insights and novelties of a human RG culture system, concerns remain regarding the model validity and power sufficiency. The 7-day RG culture generates small clones (~4.6 cells), considerably lower than the estimated naïve clone size (~15 cells). This discrepancy raises questions about whether the culture accurately models in vivo conditions and perturbation relevance.

We agree with the reviewer that further addressing the fidelity of our model to normal development is an important component of presenting our model, and we discuss evidence for fidelity and additional steps we propose below. However, this comment reflects a misunderstanding about the interpretation of clone size detected by sampling through scRNAseq rather than the accuracy of our model.

To clarify the timing: our culture system involves both a 7 day RG expansion culture after viral infection to ensure that CRISPRi knockdown is achieved prior to differentiation and cell fate choice, followed by an additional 7 days in conditions permissive of both self-renewal and differentiation to study how GRNs influence these dynamics. Our system further enables survey across multiple stages of maturation that recapitulate known temporal changes in fate potential.

Importantly, the comparison to naïve clone size suggested by the reviewer as flagging the accuracy of our model is not an apples to apples comparison because we are sampling lineages rather than comprehensively measuring them. This sampling requirement results from at least 50% GFP cell loss during FACS, scRNA-seq by 10x Genomics, joint amplification and conservative assignment of sgRNA and cell barcode. The clone size we detect here is akin to genes detected in scRNAseq as a technical metric and not a reflection of the total number of events.

Therefore, the clone size in a 14 day post infection experiment followed by FACS and droplet-based scRNA-seq is not a relevant comparison to naive clone size, which itself would depend on stage and duration examined.

Nevertheless, we agree with the sentiment to more rigorously evaluate the extent to which our system improves on existing models in terms of physiological relevance and we do so in several ways below.

Additionally, weak statistical significance in several analyses warrants further examinations (e.g., DEGs are not significant using traditional tools like DEseq2; the tool they used, TRADE, does not account for statistical dispersion; TRADE specifies that null hypothesis testing which the authors omitted and may have inflated the p-values; several analyses used a lenient FDR < 0.1 cutoff).

We agree with the reviewer that it is important to ensure statistical power in analyses used in this study, however this comment reflects a misunderstanding from the reviewer of the TRADE workflow.

In fact, all the DEG results (logFC and FDR corrected p values) and related panels used in this study are identical from DEseq2 output and account for statistical dispersion (Review Figure 2). While TRADE estimates an overall transcriptome-wide impact based on logFC and unadjusted p value from DEseq2, it does not modify p-values for individual genes. The TRADE output for significant DEGs are therefore identical with the initial DEseq2 results without inflation.

Review Figure 2: Identical p values obtained from TRADE and DEseq2 output. Red dotted line denotes $y=x$.

Major:

1. One of the major novelty is the use of primary RG cell culture – but the validity remains additional evidence. Similar screens (at much higher scale) has been done by iPSC-derived NPCs and neurons (Tian Nat Neuro 2021): what are the unique advantages of this RG system?

We agree with the reviewer on the importance of emphasizing the unique features of the screening system.

Tian Nat Neuro 2021 performed a neuronal survival screen on induced neurons from a single individual with no phenotypes measured for cell type composition or gene expression. On the other hand, Zheng et al., 2024

introduced a new perturb-seq system in vivo in mouse and screened only 4 genes in a single genetic background, but provided more physiological relevance and richer scRNAseq data.

The unique feature of our screen – that we will communicate more clearly – is that we can inactivate genes specifically in radial glia and then to allow differentiation dynamics and cell fate to occur normally, and we can do this at scale across precise maturation stages with distinct fate potentials that we now better illustrate.

No other CRISPR system offers the feature of CRISPR-based repression restricted to progenitors. Induced neuron systems drive artificial cell cycle exit and glutamatergic neuron cell fate making it impossible to study how perturbations affect either of these cell behaviors. Induced neurons also represent an artificial low-fidelity patterning system that produces a combination of peripheral and central nervous system neurons, with mixed identities, limiting the physiological relevance, while organoid models also show inconsistent patterning, lower fidelity, and increased metabolic stress (e.g., PMID: 34358451, 30735633; 31996853; 39567792; **Review Figures 3-6**). In vivo studies in mouse models represent a high level of fidelity to organismal development, but cannot address human-specific phenotypes. Importantly, introducing AAVs in vivo also requires that Cas9 be expressed and translated, and for DNA to be cut, which can permit normal differentiation to occur prior to CRISPR activity, reducing the sensitivity of the system for composition phenotypes on RG lineages.

In contrast, we optimized our system to detect composition effects on differentiation dynamics and cell fate choices known to occur in vivo. Our innovation to expand radial glia at the progenitor stage for one week in 2D culture allows us not just to localize infection to radial glia, but also to ensure that knockdown occurs at the radial glia stage (which takes ~1 week to achieve in human cells with current CRISPR tools, (e.g., Gilbert al., 2014; PMID: 25307932). This approach allowed us to identify sgRNAs affecting cell cycle, differentiation dynamics, and excitatory vs. inhibitory neuron cell fate choice – all known to occur in vivo – that would be difficult to detect if CRISPR machinery was induced at later and variable stages of these dynamic processes. Together with the permissive differentiation conditions that we optimized here, the combination of innovations in the 2D system allows us to detect cell fate consequences not possible in any existing human neural perturb-seq system (**Review Table 1**).

We have further performed perturb-seq experiments in organotypic 3D slice culture during review to compare with the 2D system. The 3D system confirms the physiological relevance of the cell type-specific gene expression consequences we observe in 2D. Both the 2D and 3D primary models show reduced metabolic stress compared to organoid models (see below), further highlighting the physiological relevance of our system. The strong correlation of gene expression consequences between the 2D and 3D organotypic models described below further adds confidence to the 2D studies that maximize sensitivity for RG lineage phenotypes.

Review Table 1. Different systems for Perturb-seq

Systems	Induced Neurons	Organoids	Primary RG 2D	Organotypic slice	in vivo Mouse
Fidelity.to.in.vivo	-	+	++	+++	++++
Human.relevance	+	+	+	+	-
RG.maturation	-	+	+++	+++	+++
Recapitulation.of.cell.fate.choice	-	+	+++	++	++
Scalability.across.genetic.background	++	++	+++	+	-
Culture.health	++	+	+++	+++	++++
Genetic.epigenetic.stability	+	+	++++	++++	++++

Table 1. Comparison of different CRISPR systems used to study cortical development.

Review Figure 3: Comparison of cell type fidelity in primary 2D RG system developed in this study and iPSC-derived organoids. Pearson correlation of top 25 cell marker gene expression defined in Wang 2025 in vivo atlas using data in this study (left) and in He 2024 organoid atlas (right). Cell type annotation was done using reference mapping of both datasets to in vivo atlas.

o The cell types in the RG culture appear significantly different from naïve human cortex. Pearson correlations in Fig. S1e are weak ($r \approx 0.2-0.4$), suggesting substantial divergence from naïve, in-vivo states. A differential expression analysis comparing this culture with naïve cortical tissue would help clarify these differences? How different are these cells from naïve states, and are they too different to be useful for perturbation studies?

We agree with the reviewer on the importance of measuring fidelity. However, the reviewer comment misreads the heatmap in Fig. S1e. The r values for the maximum correlations in major cell types are actually 0.72 (RG/Astro), 0.68 (EN), 0.50 (IN) with an average value of 0.60 across all cell types. These correlation values also reflect comparing whole cell scRNAseq data from our analysis correlated to single nucleus multiome data collected in another lab, and should be considered conservative given differences in cellular compartment, chemistry, and laboratory.

Based on the reviewer's comment, we have more directly compared the fidelity of our system to a comprehensive atlas of organoid datasets. When using predicted cell type annotation through reference mapping from Wang 2025 Nature, we observe an average Pearson correlation R of 0.62 to corresponding cell populations in vivo, representing a near 2-fold increase in fidelity compared to 0.34 in organoid datasets from He 2024 Nature (Review Figure 4), which typically fail to consistently pattern to telencephalon (Review Figure 5). Similarly, induced neurons, frequently used to study cortex, best resemble the peripheral nervous

system, with only a low fraction patterning to any central nervous system regions (Review Figure 5). Additionally, the primary cell culture model we introduce substantially reduces endoplasmic reticulum stress and ectopic glycolysis that characterize organoid models (Review Figure 6).

Thus, to the reviewer's point, our system offers substantial improvements in fidelity and reproducibility of human cortical models, while also enabling gene inactivation in radial glia themselves to sensitively measure cell fate consequences.

Review Figure 4: Pearson correlation to predicted cell types in this study compared to organoid models. Boxplot to show distribution of Pearson correlation to predicted cell types (diagonal values from Review Figure 3).

[REDACTION]

Review Figure 5: Presence of mispatterned non-telencephalic cells in organoid models and induced neurons. Left (reproduced from Fleck et al., 2022, Fig. 1d) with example patterning variation. Middle (reproduced from Lin et al., 2021, Fig. 3C), highlighting that most iNeurons across individuals best correspond to the peripheral nervous system and retina with partial brain correspondence, Right (Lin et al., Fig. 3E) show that induced neurons and organoids show a substantial fraction of non CNS cells.

Review Figure 6: Comparison of metabolic stress in human model systems. Violin plots of scores of glycolysis and response to ER stress in in vivo human atlas, 3D organotypic slice culture, 2D primary RG culture, iNeurons and organoids obtained from published studies.

o The clone size in STICR experiments (~4.6 cells) is much smaller than expected for naïve RG, which is concerning. How might this affect differentiation potential and phenotypic robustness?

Delgado et al. observed a median of 23 cells per clone after 6 weeks of culture. It is unsurprising that the clone size is lower in our experiment, sampling after 2 weeks in culture.

We further note that the clone size detected depends on the fraction of all GFP cells collected during FACS, which can differ between experiments based on the sorting efficiency and the capacity to load sorted cells into the 10x instrument.

Our study does something that has never been done before in human cortical neurogenesis - combine perturb-seq and lineage tracing. This requires sorting for dual infected cells, and maintaining the low multiplicity of infection to ensure a single perturb-seq lentiviral particle per cell means that we are sorting for ~1-2% of co-infected cells in the system. As such, this reduces the efficiency of capturing the entire lineage tree for each clone. In addition, conservative assignments of both sgRNAs and lineage barcodes to cells can further reduce the detected clone size.

Instead of estimating overall in vivo clone size, we use the topologies of observed lineage trees to identify how perturbations affect clonal fate biases. As a testament to the fidelity of our system to in vivo development and as an example of biological findings from lineage trees, we have been able to capture the hypothesized transcriptional profile of human “tri-IPCs” - radial glia in mixed inhibitory and oligo lineages as molecularly distinct from radial glia in excitatory only lineages, recapitulating the proposed markers from mouse experiments PAX6, GLI3, EGFR, and OLIG2 as the top distinguishing features between these clone types (PMID: 32234482), and supporting a novel homology between human and rodent development.

Further, we have been able to identify perturbations whose clonal fate phenotypes provide answers to developmental mechanisms underlying the cell composition effects that we observe in the initial screen.

o The dominance of interneuron populations also raise concerns about potential subpallial contamination during dissection or artifacts from RG dissociation. How do the dorsal INs compare with other IN populations (LGE/CGE-derived INs) in gene expression and relative composition? How do they align with existing data (e.g., Delgado et al.)?

Contamination of mis-identified tissue is unlikely to account for our results for several reasons:

-First, contamination cannot account for the presence of EN clones and EN/IN mixed clones, which are only produced in cortex.

-Second, contamination cannot account for the reproducible temporal gradient we observe that the fraction of mixed clones and IN biased clones increases with sample age. Collection of an additional 9 individuals further supports this age-dependent pattern, highlighting a biologically relevant pattern that may correspond to the shift late in mouse cortical neurogenesis to produce inhibitory neurons known to migrate to olfactory bulb. Indeed, we find the first clonal lineage evidence of a shared transcriptional state between primate IN-biased clones and the reported tri-IPCs analyzed by lineage tracing in mouse.

-Third, sgRNA composition effects that we detect are compared to a baseline in the same pool and would be robust to possible contamination.

-Fourth, all samples show a high fraction of cells expressing cortical markers (e.g., EMX2, PAX6 etc)

We appreciate the reviewers comment to compare with other CGE/LGE populations. Below, we highlight molecular differences between IN subtypes identified in our 3D slice culture (**Review Figure 7**):

The Immature locally born IN cluster (IN_dLGE-CGE), clonally coupled to cortical RG and immature ENs, is adjacent to 2 subtypes of cortical INs that have OB or CGE like molecular identities, respectively.

IN_dLGE-CGE highly expresses dorsal marker PAX6 together with other LGE/CGE markers MEIS2, CALB2 and PROX1, as well as immature EN markers (ST18, NPY) compared to other IN subclusters, but lowly express migration markers (ERBB4, CXCR4).

In revision, we can integrate with existing data (Delgado 2022, Keefe, Steyert & Nowakowski 2025) to show fidelity and reproducibility of dorsal IN observations.

Review Figure 7: Molecular identities of different IN subtypes. Left, UMAPs showing cell types identified in the slice culture. Middle, heatmap to show lineage coupling of major cell types. The IN_dLGE-CGE cluster is clonally linked to cortical RG. Right, dot plot highlighting marker gene expression in each IN subtypes.

o How does this culture compare to iPSC-derived NPCs or neurons, where Perturb-seq has been widely applied? Does it reveal phenotypes that iNs cannot?

We thank the reviewer for raising this question as this is an important phenotype to better illustrate.

Perturb-seq has been widely used in induced neurons and is possible in early NPCs and organoids, though silencing of CRISPR machinery during longer differentiation experiments has limited its use. However, these iPSC systems have major limitations for interrogating the dynamics of neurogenesis. As mentioned above, induced neurons have limited fidelity to normal development with NGN2 neurons often producing a mix of CNS and PNS neurons (**Review Figure 5**). The process of forced transcription overexpression overrides normal development making it difficult to study cell cycle and cell fate choices in an artificial model. Meanwhile, early NPCs are transcriptomically distinct with maturation times though to reflect in vivo timescales (PMID: 34535062), making it difficult to model mid-fetal cortical neurogenesis, when important neural cell fate events, including the transition from excitatory neurogenesis to inhibitory neurogenesis and gliogenesis take place.

In contrast, our system enables sensitive detection of perturbations that affect cell fate choice at defined stages of radial glia maturation that recapitulate cell fate choice and lineage plasticity that occurs in vivo.

Our system also compares favorably along several dimensions:

iPSC-derived models experience cell culture acquired genetic and epigenetic changes that can confound results (PMID: 28445466; PMID: 28017796; PMID: 35885940; PMID: 34407428) and the engineering of individual cell lines has typically limited screens to a single individual. This is problematic as screens are performed in a single genetic background, limiting generalization.

Even for commonly used iPSC lines, the genetics is often compromised. The KOLF2.1J iPSC line, introduced to standardize iPSC lines and overcome these problems has CNVs causing haploinsufficiency in 3 neurodevelopmental disorder genes (PMID: 37425875). The line CW20107 selected by SSpsygene and multiple CIRM projects for expensive engineering shows 60% aneuploidy of Chr1 duplications plus additional subclonal aberrations. Clones of the Tuba1-GFP and the commonly used WTC-11 iPSC line in circulation for induced neuron experiments also have structural variants and/or aneuploidy that increases with prolonged passaging (PMID: 35885940).

In contrast, our primary cells do not go through clonal bottlenecks or prolonged cell culture (19 days in culture) versus low-efficiency clonal isolation and typically 15 - 50 passages prior to even starting a differentiation experiment.

In a previous attempt to overcome the limitations of screening in a single genetic background, we spent several years engineering identical CRISPRi machinery into multiple safe harbor sites in iPSCs from 6 human and 6 chimpanzee, and one orangutan individual (PMID: 37343560). While this worked for iPSCs, most lines showed variable silencing of CRISPRi machinery after 2-3 weeks of neural differentiation (and some lines showed P53 hypofunction), making it difficult to scale CRISPR-based approaches by engineering iPS lines.

In contrast, our introduction of an all-in-one direct capture vector and the primary cell system allows us to scale our approach to many individuals and multiple species. Here we have performed experiments, including CRISPRi + flow cytometry, in 26 individuals from two species. Our ability to cryopreserve primary samples also lets us compare different age samples in common culture and cell capture batches further mitigating technical batch effects, while surveying a greater and more well-defined landscape of normal human development.

o Additional validations, such as tissue-level imaging, donor-to-donor reproducibility, and assessments of culture health, would bolster confidence in this system.

We appreciate the reviewer's comment to assess reproducibility and culture health. We have calculated glycolysis and ER stress scores and compared them with other model systems including human datasets in vivo, in iNeurons and organoids (**Review Figure 6**). We found that the cultural system presented in this study had lower metabolic stress than iPSC-derived neurons or organoids.

If necessary, we can include time lapse imaging during revision to further address this point, but note here that previous studies have recapitulated relevant cell type behaviors such as outer radial glia mitotic somal translocation (PMID: 25088420; PMID: 26406371).

Review Figure 6 (reproduced from above): Comparison of metabolic stress in human model systems. Violin plots of scores of glycolysis and response to ER stress in in vivo human atlas, 3D organotypic slice culture, 2D primary RG culture, iNeurons and organoids obtained from published studies.

We thank the reviewer for raising the importance of emphasizing donor-to-donor reproducibility in our study. Throughout the study, we included 9 human individuals and 3 macaque individuals in the single cell experiments, and 9 human individuals (6 not included in the single cell workflow) in the flow cytometry validation. During review, we repeated our single cell workflow on another 5 individuals (human, n=4, 3 not included in the flow cytometry; macaque, n=1) in 2D and 4 human individuals in 3D slice culture spanning the stage of excitatory and inhibitory neurogenesis and early gliogenesis (Review Figure 1). Cross validation from a total of 26 individuals (human, n=22, macaque, n=4) confirmed that the phenotypes that we are reporting are robust across genetic backgrounds. Biological replicates are already included as a parameter in our models, and we can add analyses in revision further quantifying the reproducibility of across individuals.

2. Statistical power and sampling size in the study is another concern. The study's design (400~500 cells/gene; Fig. S1i) translates to ~56 cells/perturbation/cell type across eight cell types—considerably lower than recent screens (e.g., Replogle et al., Cell, 2022; ~200 cells in a genome-wide screen; Li et al., Nature, 2023; ~80 cells per perturbation per cell type in organoids). Given this low sample size, can the authors justify the screen's

statistical power? How are the effect observed across batches/replicates? Perturb-seq downsampling analysis is a routine – is the current cell number well powered to reach conclusions? Additional replicates or cell numbers may improve reproducibility.

The initial screen in this study collected an average of 631 cells per perturbation and 79 cells per cell type instead of 56. This number is greater than Li et al., Nature, 2023 which reported 64 cells per cell type across 17 cell types.

Effects of perturbation for both cell type composition and gene expression were calculated accounting for biological replicates. 4 biological replicates were used for the initial screen and 8 more for the targeted lineage tracing study. Throughout the study, consistency was observed between individual replicates (Review Fig8), batches (FigS7k.), sgRNAs (FigS2.4) and species (Fig4c, FigS8j).

We have implemented downsampling in the differential composition analysis in this study and here further show the robust detection of consistent phenotypes across conditions tested (200-600 cells/gene) (Review Figure 9). Therefore, our initial screen is well powered to detect effects of different perturbations.

Review Figure 8: Reproducibility across biological replicates. Cluster-free differential abundance testing using Milo at the individual level using sgRNA as replicates. Nodes are neighborhoods with sizes representing the number of cells in each neighborhood. The edges depict the number of cells shared between neighborhoods and the layout of nodes is determined by the position of the neighborhood index cell in the UMAP. The nodes are colored by their log₂FC (FDR=0.1) when perturbed.

Review Figure 9: Robustness to downsampling. Cluster-free differential abundance testing using Milo after downsampling to 200-600 cells.

3. I am concerned about the DEG statistical tests for several reasons, as the authors used a somewhat unusual pipeline:

- o For DESeq, the authors only check that more than 1 cell is has the perturbed condition within the cell type they are looking for – this is a very low cell number threshold..
- o The authors then send a few outputs to TRADE: only taking in the un-adjusted P-value and does its own FDR and Bonferroni correction, but this does not account for statistical dispersion? It also cannot account for the low statistical power that has been already introduced.
- o All the figure panels are made with the TRADE output, and not the DESeq output. I wonder how many statistically relevant genes that DESeq identifies? If at all, do they overlap with the TRADE analysis?
- o TRADE itself specifies that null hypothesis testing should be done with a jackknife sterr method, which involves rerunning the pipeline many times with 1 batch leave-out. The authors don't seem to do that anywhere in their code though, and this may lead to inflated statistical values.

We agree with the reviewer the importance of ensuring proper statistical testing for our analysis. However, this comment represents a few misunderstandings in our workflow:

1. The cell number threshold before pseudo bulking for DESeq2 workflow was at least 2 cells per individual. After pseudo bulking, we further applied a threshold of a minimum 2 biological replicates in each cell type and condition. Also note that conditions that have 2 or lower cells only apply to rare cell populations that are not the main focus of this study (microglia, vascular, etc) and therefore does not impact our conclusions. We can rerun this analysis with a high cell number threshold in revision.
2. The TRADE output is identical with the DESeq2 output and therefore the p values are not inflated.
3. All the figure panels use p-values that are identical to DESeq2 output.
4. The null hypothesis testing that the reviewers mentioned applies to TRADE estimates of “overall transcriptional impacts” which we did not include in this study.

Review Figure 2 (reproduced from above): Correlation plot to show identical p values obtained from TRADE and DEseq2 output.

4. Flow cytometry validation (Fig. S6) seems contradictory to the Perturb-seq results. Several single-cell phenotypes fail to replicate in flow assays. Notably, DLX2 perturbation does not alter interneuron composition, and Neurod2 excitatory neuron phenotype is inconsistent with single-cell data. Can the authors address these discrepancies?

The reviewer has misread Figure S6. Flow cytometry validation was only performed in perturbation of top TF candidates ZNF219, ARX and NR2E1. DLX2 perturbation was not performed.

DLX2+ populations were stained to quantify the changes in inhibitory populations in each KD condition and consistency was observed in all conditions. For example, decrease in EOMES+ and NEUROD2+, and increase of DLX2+ populations in NR2E1 KD mimics the decreased EN/IN ratio observed in the initial Perturb-seq screen and the follow-up lineage tracing experiments in both species. Additionally, 2 other differentiation timepoints were included to investigate the temporal dynamics of cell fate changes under perturbation, reinforcing consistency along the RG differentiation trajectory.

5. The experimentally measured TF impacts vs. eGRN predictions are inconsistent: the 44 TFs were selected for their predicted strong influence on transcription, yet the observed effects (Fig. 2b) appear weaker than expected. Could this be due to insufficient statistical power or the culture system failing to replicate naïve tissue states (where the eGRNs were derived)?

The reviewer raises an important question raised by our study about why all 44 TFs do not have a strong influence on cell fate. We detect different numbers of altered genes depending on the TF perturbation and we only report on a subset with strong composition consequences.

As shown above, our screen is well powered to detect composition and gene expression changes under TF perturbations. Similarly, our system is closer to in vivo metabolic state than organoid models and better recapitulates the in vivo cell types where the eGRNs were derived.

Although statistical power is not an issue based on downsampling our strong hits, we expect that additional screening would add confidence to smaller effect perturbations.

One possible explanation for why more TFs do not cause strong cell fate consequences is that CRISPRi produces quantitative repression and not complete inactivation. Given the difference in knockdown efficiencies,

we do not make strong claims that the absence of strong phenotypes for some TFs means those genes are unimportant.

However, we must note that the in silico perturbation tools are still in their infancy and may not account for compensation/redundancy, which is why performing the experiments is important

We could re-run the initial 44 TF screen to increase statistical power for smaller effect mutations.

6. The study reports divergent phenotypic effects between human and macaque cultures. Could these differences stem from technical factors (e.g., RG survival biases due to culture conditions or cryopreservation conditions, or general age differences in donor tissues)? Testing key hits (e.g., NR2E1, ARX) across multiple developmental stages may help clarify whether these effects are age-dependent.

We agree with the reviewer that confounding effects from stage differences can lead to phenotypic differences we observe in the system. As mentioned above, during review, we have expanded our observation of single cell lineage tracing study to 9 independent human individuals from broader developmental stages and confirmed the stage-dependent effects of top TF candidates in this study. For example, we identified ARX and NR2E1's capability to alter developmental tempo across multiple cell lineages, while this RG lineage plasticity appears to peak between GW16-19 (**Review Figure 10**).

Review Figure 10: Human data with targeted library and lineage tracing. (left) UMAPs of cell type annotation, stage, sex, individual and marker gene expression. (middle) dotplot showing cell type compositions in each clonal cluster and heatmaps showing compositional changes of clonal clusters under perturbations. (right) Barplots for changes in clonal clusters along developmental stages under perturbations.

When performing comparative studies in macaques, we have minimized the confounding effects through matching sgRNA, cryopreservation and cell culture condition, as well as by matching developmental stages. We reported largely conserved phenotypes in cell type composition and correlated gene expression responses. Against the background of conserved responses, we highlight the subset of divergent gene expression

responses upon TF perturbation as a new approach to functionally dissect effector gene evolution in conserved GRNs. Importantly, the gene divergence was calculated in each cell type under multiple biological replicates (humans, n=5; macaques, n=3) and stages (humans, GW17-22; macaques, PCD75-80) in both species, therefore is not confounded by composition differences across stages.

7. gRNA Variability is another issue. Given the observed variability in gRNA effects, per-guide analyses would be beneficial. Do different guides targeting the same TF yield consistent phenotypes?

Consistent phenotypes between sgRNAs sharing the same gene target were observed at the level of cell type composition (Figure S4) and gene expression (Figure S2, **Review Figure 11**).

Review Figure 11: Consistency of DEG results between sgRNA. Correlation plots of logFC of differentially expressed genes in the sgRNA level and gene level. Red dots denote DEGs that were detected as significant in at least 2 conditions.

Minor:

- The 44 TFs were selected based on their broad transcriptional impact but are not necessarily causal drivers of differentiation. Pseudotime trajectory analyses could provide insights into their effects on differentiation dynamics beyond discrete cell-type compositions.

We agree with the reviewer that finer resolution of changes will help uncover effects of TF perturbations. We applied Milo for cluster free differential abundance testing, which provided subcluster resolution. In revision, we can include comparison of pseudotime distribution during revisions.

- The cross-species comparisons in Fig. 4 are not entirely convincing. Some patterns appear conserved, while others do not. Statistical correlation analyses would help support the conservation hypothesis.

We thank the reviewer for the suggestion. We have included correlation analysis at the differential gene level (FigS8j) and here attach correlation of compositional changes between the two species to support the conservation hypothesis (**Review Figure 12**).

Review Figure 12: Correlation of compositional changes in humans and macaques. Scatter plot of estimated coefficients from DCATS in human (x axis) and macaque (y axis). Pearson correlation coefficients and p values were calculated in each TF perturbation and labeled together with regression line equations.

• Lines 933-935: What method was used for in silico perturbation, and was it validated against in-vitro or in-vivo datasets?

We applied the SCENIC+ workflow for in silico perturbation. In the original publication, the authors used in vivo datasets in human and mouse brains and observed the predicted results consistent with previous results.

Referee #2 (Remarks to the Author):

In the manuscript entitled Dissecting Gene Regulatory Networks Governing Human Cortical Cell Fate, the authors investigate the transcription factor (TF) circuits that regulate human cortical neurogenesis. Using single-cell CRISPRi screening in a human primary culture system, they examine the functional roles of 44 TFs in determining radial glia (RG) lineage progression. Their findings reveal key regulators such as NR2E1, ARX, and ZNF219, which influence the balance between excitatory and inhibitory neuron (EN/IN) differentiation. They also identify effector genes implicated in neurodevelopmental disorders and demonstrate conserved mechanisms of RG lineage plasticity across primates. This study offers a framework for dissecting gene regulatory networks underlying human cortical cell fate decisions.

The manuscript presents a solid study with several strengths, including the use of valuable human fetal cells to investigate authentic gene regulatory networks controlling human cortical specification. The combination of cutting-edge technologies, such as Perturb-seq and lineage barcoding, provides important insights into the developmental origins and lineage plasticity of cortical cell types. Additionally, the cross-species comparison between human and macaque offers an interesting perspective on the conservation of lineage plasticity. However, the study lacks the level of novelty required for publication in Nature for the following reasons:

1. Limited technological innovation –While the Perturb-seq approach is a powerful tool, its application has already been demonstrated in various biological contexts. In this study, the reviewer does not see significant methodological advancements beyond what has already been published. It should also be noted that number of genes targeted (44) is relatively limited by today's standards. Without major technical improvements or novel adaptations, the study does not provide a sufficient technological breakthrough to justify publication in a high-impact journal.

We thank the reviewer for raising the important point that we need to demonstrate the technological innovation of our system, which we believe is qualitatively new, enabling analysis not possible in any existing system.

We did not sufficiently discuss the limitations of existing systems in our initial submission, nor did we discuss the methodological advances we made.

In brief: Organoids and induced neurons show patterning heterogeneity, limited fidelity to primary tissue, ectopic glycolysis pathway activation and endoplasmic reticulum stress pathway upregulation. Induced neurons artificially short circuit normal developmental events, making it impossible to study cell cycle, physiological differentiation dynamics, or cell fate choice (while also variable patterning to peripheral nervous system glutamatergic neurons). Organoids show limited and variable maturation, while exhibiting necrotic cores, and frequently silence CRISPR machinery. No existing system allows restricting CRISPR-based gene restriction to RG to study lineage consequences. Neither induced neurons nor organoids have been scaled across multiple genetic backgrounds or applied to study temporal changes in cell fate potential.

We address all these limitations and now provide analyses to demonstrate the improvements (as well as the novel biological findings they enable). Reproduced from above:

The **technological novelty** of our system enables dissection of mechanisms underlying cell cycle dynamics and cell fate choice not possible with existing human cortical CRISPR systems by:

- **Increasing the fidelity** to normal cortical development, with more precise cell type homologies than cortical organoids (**Review Figure 1A**), reduced metabolic stress and ER stress (**Review Figure 1B**).
- **Improving the reproducibility of cortical patterning**, with the samples we use having been patterned 16-22 weeks in vivo, compared with organoids enduring long-term stress and showing variable patterning and less than 50% cortical patterning (Fleck et al. 2022) and induced neurons showing less than 50% CNS patterning with cells best resembling PNS neurons (**Review Figure 1C-D**).
- **Improving the scalability across individuals**, with 21 perturb-seq samples across 2 species and 3D organotypic slice culture model now included with our new experiments (**Review Figure 1E**)
- **Enabling sensitive detection of cell fate consequences**, with a week-long RG expansion phase allowing for CRISPR activity at the progenitor stage rather than manifesting in cells differentiated to various stages (**Review Figure 1F**) We now show our 2D system further recapitulates gene expression consequences of TF knockdown in gold standard human organotypic slice (**Figure 1G-H**), while also uniquely providing the RG expansion point for localizing gene repression to RG.
- **Increasing temporal precisions**, with organoids radial glia (RG) showing variable maturation stages in the same organoid (Pollen et al., 2019; Bhaduri et al., 2020), while induced neurons short circuit physiological RG (and normal cortical cell types) altogether. In contrast, our system allows us to survey defined in vivo maturation stages and recapitulates a known sequence of temporal cell fate decisions at the individual clone level (**Review Figure 1I-J**)

2. Moderate biological impact – The findings are not sufficiently groundbreaking, as out of the 44 TFs analyzed, only ZNF219 is identified as a novel regulator of human corticogenesis. The roles of NR2E1 and ARX in cortical development have already been established, and while the authors provide additional details on lineage specification and inhibitory neuron subtype regulation, these findings do not represent a major conceptual leap in the field without additional follow up work on the mechanism of action.

We thank the review for raising this foundational point - our initial manuscript was organized more around novelty of the screen design combining perturb-seq and lineage tracing in primary cortical tissue, but with new experiments, we can now conclude the paper with two figures devoted to important biological findings enabled by our system. Figure 4 will address TFs regulating developmental tempo with conserved phenotypes across primates, and Figure 5 will address TFs safeguarding post-mitotic inhibitory neuron maturation.

By performing new experiments during review (9 total samples spanning 6 weeks), we identified the effects of ARX and NR2E1 on modulating other lineages beyond balancing EN/IN neurogenesis, suggesting their opposing roles in regulating the developmental tempo of the human cortex (**Review Figure 13**) addressing a major current question in developmental biology to identify mammalian regulator of developmental tempo. We further identified a critical temporal window for EN/IN plasticity of RG that can be engineered through TF perturbation, providing clinically relevant insights for disorders driven by E/I imbalance. We plan to add an updated Figure 4 focused on the biological theme of conserved TF mechanisms regulating heterochrony of cell fate choice revealed by the study.

Review Figure 13: Human data with targeted library and lineage tracing. (top) UMAPs of clones colored by pseudotime and fate biases. (bottom) Ridge plots of pseudotime distribution in clones grouped by stage and perturbations.

For the new Figure 5, we will focus on **TFs safeguarding IN Identity**. Within the IN lineage, we discovered a conserved role of ARX and SOX2 in safeguarding normal post-mitotic IN identity. Loss of ARX has long been linked to impaired inhibitory neuron migration and epilepsy in ARX-linked lissencephaly. Our findings add a novel cellular mechanism to this literature by revealing that inhibitory neurons with reduced ARX adopt an ectopic cell state – conserved across species – and by providing a set of the earliest human dysregulated genes. During review, we have confirmed that the ectopic state is recapitulated in organotypic slice culture.

To buttress the finding that these stable and conserved ectopic states exist outside of normal morphospace, we will mine existing mammalian brain atlases for any signs of these dysregulated states in normal cell types. To our knowledge, TFs safeguarding normal developmental morphospace in mammals have not been previously identified. We will further test the hypothesis raised by Reviewer 3 that induction of LMO1 mediates this ectopic

state by performing dual repression of ARX and LMO1. We plan to add an updated Figure 5 focusing on the biological theme of TFs safeguarding post-mitotic cell identity.

As additional novelty, we novel genes regulating cortical development beyond ZNF219, including PHF21A, previously implicated in neurodevelopmental disorders. In addition, in Figure 2, we synthesized our results to advance a concept that neurodevelopmental disorder-linked genes show increased connectivity in functionally defined gene regulatory networks.

We agree that the paper will be improved by focusing on these biologically novel findings about heterochrony, cell fate potential, and safeguarding normal developmental morphospace.

3. Lack of internal controls – The study would benefit from including perturbations of well-characterized TFs known to regulate human cortical cell fate. This would serve as an internal benchmark to validate the experimental system and provide a clearer functional framework for interpreting the role and timing of the novel TFs being investigated.

We included NEUROD2 as an internal control, even using the sgRNAs to optimize our silencing and differentiation time course and confirm protein knockdown by flow cytometry prior to perturb-seq experiments to maximize sensitivity for detecting altered RG differentiation sensitivity. However, we did not use the term positive control in the study, because the genes largely have not been studied with this developmental resolution during human cortical neurogenesis.

We agree that other well known genes such as PAX6 and GLI3 could serve as additional benchmarks, and we could include these in an additional replication screen.

Minor comments include:

- ARX and SOX2 Perturbations (line 183-): The formation of new clusters (“ectopic”) following ARX and SOX2 knockdown is interesting, but the characterization of these clusters is limited. It would strengthen the paper to discuss whether these ectopic clusters have any known in vivo counterparts or if they represent entirely novel cell states. Additional marker analysis could help clarify this.

We agree - we investigated the similarity of these clusters to other cell types in the developing human cortex as well as developing macaque telencephalon using public atlas data from Wang 2025 Nature and Micali 2023 Science. Here we attach results from reference mapping to in vivo datasets and marker expression. We found that while the ectopic clusters define unique transcriptomic states that is distinguishable from other IN subtypes observed in NT controls, they still showed no clear correspondence to other in vivo IN subtypes identified in macaque telencephalon at both transcriptomically and marker expression level, instead retaining general CGE/LGE features while occupying distinct transcriptional states (Review Figure 14).

Review Figure 14: Molecular identities of ectopic clusters in ARX and SOX2 KD. Heatmap of results from reference mapping of IN clusters to in vivo atlas of the developing human cortex (Wang, top left) and developing macaque telencephalon (Micali, top right), with dot plot showing marker gene expression of known IN subtypes.

- Lineage Tracing Experiment: While the lineage tracing approach is technically impressive, its contribution to the overall conclusions of the paper feels limited. Consider elaborating on how this experiment adds new insights beyond what is already provided by the perturbation and transcriptional analyses.

We agree with the reviewer and this fits the constructive theme of Reviewer 2's comments that we need to better illustrate the biological novelty of our study rather than just the technical achievements. For this point, we will elaborate more in revision how results from lineage tracing results provide developmental mechanisms support the conclusions beyond the initial screen.

First, lineage tracing in NT controls confirmed shared lineage of ENs and INs and provided evidence for TF perturbation engineering RG plasticity.

Second, lineage tracing now allows us to reconstruct the pseudotime of cell fate choices during cortical development, confirming sequential changes in cell fate potential with increased age and revealing the transcriptomic profile of the "tri-IPC" resembles that of mouse suggesting re-use of mouse cortical olfactory bulg progenitor programs.

Third, coupling lineage tracing with perturbations was required to demonstrate that distinct TF perturbations altered developmental tempo of the human cortex in both directions across multiple lineages spanning excitatory, inhibitory neurogenesis and gliogenesis (**Review Figure 13**).

- Gestational Week (GW) Sample Variation: The inclusion of samples across a broad range of gestational weeks provides developmental context, but the potential impact of this variation on the results isn't fully addressed. It would be helpful to discuss whether differences in cell composition or gene expression could be influenced by developmental stage, and how this might affect the interpretation of TF perturbation outcomes.

We agree with the reviewer that variation along developmental stages can impact effects of TF perturbation. By increasing our sample size during review and harnessing the reproducibility of in vivo maturation, we have now turned this scalability across ages into an advantage to expand our biological findings.

Indeed, when we extended the targeted screen to late mid-gestation (GW22), changes in clone composition and perturbation effects were noted. During review, we have collected 4 more individuals spanning early to late mid-gestation, making a total of 9 individuals (GW16, n=1; GW17, n=2; GW18, n=1; GW19, n=2; GW21, n=2, GW22, n=1) to test the stage-dependent effects of top TF candidates on RG lineage plasticity.

We observe consistent phenotypes with data presented in the manuscript. Moreover, sampling densely from stages spanning excitatory, inhibitory neurogenesis and oligodendrogenesis further revealed opposing roles of NR2E1 and ARX KD on the developmental tempo of cortical RG, impacting gliogenesis beyond EN/IN balance.

Review Figure 10 (Reproduced from above): Human data with targeted library and lineage tracing. (left) UMAPs of cell type annotation, stage, sex, individual and marker gene expression. (middle) dotplot showing cell type compositions in each clonal cluster and heatmaps showing compositional changes of clonal clusters under perturbations. (right) Barplots for changes in clonal clusters along developmental stages under perturbations.

Overall, while the study is rigorous and contributes valuable data to the field, its lack of significant novelty limits its suitability for Nature journal.

We thank the reviewer for these comments and hope that by extending the biological findings to TFs regulation heterochrony and to TFs safeguarding post-mitotic identity that we have been able to satisfy these concerns.

Referee #3 (Remarks to the Author):

In humans, most cortical interneurons (INs) originate from the ganglionic eminences in the ventral forebrain and migrate to the cortex. However, recent work has identified a small but significant population of locally generated interneurons that arise in the dorsal telencephalon alongside excitatory neurons (ENs). In this study, Ding and colleagues show that these interneurons are generated from the same progenitors that give rise to excitatory neurons in the developing cortex. This suggests a broader neurogenic potential of radial glial cells (RGs) than previously thought. The authors here screened 44 transcription factors to determine the molecular

program to generate ENs or INs and found that NR2E1, ARX, SOX2, and ZNF219 play an important role. The presence of cortical-generated INs has been demonstrated in humans, but here they show that this can be extended to other primates. Overall, this study improves our understanding of the regulatory networks that control cell fate decisions during human neurogenesis and suggests that this may also play an important role in neurodevelopmental processes and disorders.

We thank the reviewer for these kind words and thoughtful comments.

Here are my suggestions to strengthen this study:

Selection of transcription factors: I understand how the TFs were initially selected, but I'm a bit puzzled because some very well-known, important, TFs are missing, e.g. PAX6, GLI3, HOPX, EOMES. On the contrary, some only neuronal TFs (TBR1, SATB2) are included while the fate of RGs is investigated.

Our approach is data-driven where TFs predicted to drive the strongest transcriptional changes along the differentiation trajectory were selected.

However, as prediction methods are in their infancy, we agree with the reviewer that our system will benefit from including well known marker TFs, and we can include PAX6, GLI3 and EOMES in revision. As most of the marker TFs have not been functionally characterized at this stage of human cortical development, these experiments will add additional interest, but will not serve specifically as positive controls.

Model system: I'm slightly surprised by the 2D approach when we now have many ways to look at these events in 3D. For example, why not use human fetal slices directly? Alternatively, the authors could use the fetal brain organoid model (FeBO) recently developed by the Artegiani lab. From their data, it appears that identity and developmental stages are maintained when FeBOs are grown in 3D. In addition, the authors should provide evidence for the precise identity of these cells in 2D (I find it hard to find these data, day0 cells are very few in the single cell RNA-seq in Fig. 1B). What is the exact identity of these cells? How can they specifically show that they are only cortical and that there are no ventral RGs?

We agree that the 3D organotypic slice may represent a more physiological cultural system. We have now performed a targeted screen paired with lineage tracing and observed highly correlated gene expression response upon TF perturbations in 2D and 3D (Review Figure 15).

Meanwhile, we would like to note limitations in alternative 3D systems that motivated our decision to establish a 2D system.

- Limited scalability across individuals in 3D models: Organotypic slices cannot be cryopreserved and slice and FeBO models cannot be cultured in pools to scale across individuals while mitigating batch effects, in contrast to cryopreserved chunks followed by batch controlled 2D cultures
- Limited capability for cell fate engineering in 3D models: Even if CRISPRi lentivirus infection is localized to radial glia, differentiation can still occur prior to CRISPR activity resulting in heterogeneous starter populations and reducing sensitivity for detecting cell fate consequences in RG lineages. For example, Cas9 activity from viral libraries typically takes ~7 days in human RG, while differentiation can occur after one cell cycle (24-48 hrs). In contrast, the 2D culture system allows us to expand cells as RG to ensure gene repression occurs in a homogenous starter population. RG from our system could then be transplanted to 3D systems, including in vivo mouse, to further harness 3D in vivo environments, building on our system.

Review Figure 15: Reproducibility in organotypic slice culture. (Left) Schematics of slice culture experiments, with UMAPs showing cell type annotation, individual, species and perturbations. (Right) Scatterplot showing correlations of gene expression changes in primary 2D and 3D slice culture.

For figure 1, the identities of D0 cells are available in the barplot Fig1c, we apologize that it was too small in the initial manuscript and attach here a larger version together with dot plots and UMAPs showing identities of D0 cells:

Review Figure 16: Molecular identities of differentiation D0 cells. (Left) Barplot showing cell type compositions in D0 in two individuals. (Middle) Dotplot showing marker gene expression and UMAPs showing cell type distributions on D0 and D7. (Right) Dotplot showing marker gene expression on D0 with bar plots representing absolute cell numbers in each cell type.

95.3% (94.3% in GW16 and 96.4% in GW18) of 21151 cells collected at D0 were assigned to cluster radial glia class (RG_DIV and RG/Astro) (Review Figure 16), a small population expressing markers for OPCs (OLIG2, BCAS1). OPCs are fate restricted progenitors that do not give rise to OPC-neurons, therefore their presence does not account for the EN/IN phenotypes we discuss in this study.

Timing: It would be nice to have a rationale for selecting the fetal time period that the authors decided to start with. Is this the time when the INs are generated? Why are so many INs generated at the end? Shouldn't they be a tiny fraction of the general neuronal population?

We agree with the reviewer that we should emphasize on our rationale for experimental design as part of expanding on our biological findings. We chose to focus on mid-gestation, starting with GW16-18 in the initial screen, because these stages represent the peak of cortical neurogenesis. Further analysis revealed that by including later timepoints, we could follow sequential changes in cell fate potential giving us resolution to explore regulatory mechanisms controlling the timing of these temporal changes.

A similar ratio of IN was reported in Delgado et al., Nature 2022 (~20%) and in this study (~17%). While we do observe an increase in IN populations in 2D in comparison to slice culture, this change in cell proportions does not impact our conclusions on the roles of TF perturbations when compared with NT controls in the same system. On the other hand, equally represented EN/IN populations allowed us to sensitively detect compositional changes under TF perturbation.

Virus selection: Lentiviruses infect all cells, including differentiated cells. The authors use a protocol enriching for progenitors, but this does not exclude the presence of neurons. Therefore, it would be important to show the % of cells in the progenitor state, and the % of differentiated cells (ENs and INs) with some IHC. In addition, the use of a retrovirus could be a better choice, because it excludes infection of postmitotic cells since it only infects dividing cells.

We agree with the reviewer that it is important to ensure the homogeneity of the starter populations. As reasoned above, single cell analysis revealed the predominance of radial glia cells on D0. We observed similar results at the protein level using flow cytometry for marker quantification (Review Figure 17). In revision, we can include this data and add more biological replicates.

Review Figure 17: Marker quantification of differentiation D0 cells at the protein level. Barplot showing marker populations in D0 in two individuals quantified by flow cytometry, Consistent results from the targeted screen paired with lineage tracing also provides evidence for a homogeneous starter population since any clones that contained differentiated cells would be removed from clonal analysis.

At D0, the authors use HOPX and KI67 to define cycling RGs. why HOPX? why not PAX6? PAX6 should define RGs and HOPX bRGs according to previous papers by the same authors. Some cell clusters are not clear to me. Why do IPCs express TBR1? Why is PAX6 in INs? Some cells are not FOXG1, what are they?

Cell type annotation is defined based on marker gene expression, differential gene expression analysis between clusters and reference mapping to the in vivo atlas. HOPX and MKI67 were shown for visualization but were not the only criteria that we used to define cycling RG. We agree with the reviewer that specific RG subtype, such as vRG, can be HOPX negative, we apologize for not making that clear and have attached a dotplot with additional marker panels to support our cell type annotations (Review Figure 18).

Most cell types in our culture are FOXP1 positive, while the negative populations include OPC and oligodendrocytes, vascular and microglia (**Review Figure 18**).

While PAX6 is highly expressed in RG, it is also expressed in LGE-derived INs, which has been reported in multiple studies including Keefe, Steyert & Nowakowski 2025 Nature, Schmitz et al 2022 Nature.

TBR1 is expressed in excitatory IPCs at the transcript level based on in vivo atlas in Wang 2025 Nature (**Review Figure 19**).

Review Figure 18: Cell type annotation of D7 cell types. Dotplot showing cell type marker expression in each cell type. Additional RG (VIM, GLIS3, SLC1A3), EN_IPC (PPP1R17, SSTR2) and IN markers (DLX2, GAD1) were included to support cell type annotations.

Review Figure 19: TBR1 expression in Wang 2025 in vivo atlas. Dotplot showing excitatory IPC markers and TBR1 expression in each cell type annotated in Wang 2025 Nature.

The authors nicely link the TFs to neurodevelopmental disorders but have not explored this more in-depth. For example, recent work has focused attention on ASD and hypothesized that E/I imbalance is critical. It would be nice to test this hypothesis in a disease context using dorsally patterned cerebral organoids derived from ASD patients.

We agree with the reviewer that it is an intriguing hypothesis to the role of E/I imbalance in ASD. In fact a few of our top candidates (ARX, CTCF, PHF21A) have been identified with causal mutations in ASD. However, since this study focuses on the effects of TF perturbation on normal cortical development, applying patient-derived organoids are beyond the scope of current study. We also highlight several important advantages of the primary cell system (e.g., **Review Figures 3-6**) that would support studying ASD-linked mutations in primary cell models as a complement to organoid tools.

The authors show that more than 25% DEGs are common for more than 1 TFs. This is very intriguing and could be extended a bit with GO and validated in some way, e.g. IHC...

We agree with the reviewer that it is worth investigating and validating the convergent DEGs. We have included GO analysis in Fig. 2d. In review, we have added validation that DEGs detected in 2D also emerge in slice culture, supporting the generalization of these findings.

The perturbation results are interesting as well as unexpected as it seems that so many of these TFs regulate the fate of INs. Interesting because in the cortex these are only a very tiny fraction and therefore it is perplexing that this population is so highly orchestrated by so many TFs. However, since the authors focus on ARX, NR2E1, and ZNF219, they could expand and show the expression of these TFs at the protein level. IHC and double staining with some of the known TFs (PAX6, HOPX, DLX....) would strengthen these findings. KD of ARX results in IN with GE identity. The authors may want to follow up on this point and compare the ARX INs with LMO GE-derived INs. Perhaps KD of these (LMO, RIC3) genes would allow to define their function in this subtype.

We agree with the reviewer that confirming the expression of our top candidates at the protein level together with other cell type markers can include those validations in revision.

In the new data that we collected in organotypic slice culture, we have observed consistent phenotypes in ectopic IN clusters in ARX (LMO1/RIC3) and SOX2 (CALB2/SOX2) KD, suggesting in vivo relevance of our findings in 2D systems (Review Figure 20).

We thank the reviewer's suggestion on performing LMO1 and RIC3 KD. These genes are only induced upon ARX knockdown in post-mitotic interneurons (which represent a subset of cells infected with the ARX sgRNA). This limitation to a subset of GFP labeled cells plus the early stages of maturation that we are studying may make it hard to functionally characterize the IN subpopulations, though we will attempt dual infection in slice culture.

However, as LMO1 is a transcriptional co-factor, with potentially strong consequences on gene expression, the reviewer raises an exciting experiment that we are now attempting to perform dual ARX and LMO1 knockdown and examine whether LMO1 induction functionally mediates the ectopic cell state that we observe.

Review Figure 20: Ectopic IN subtypes in ARX and SOX2 KD in 3D slice models. (Left)MAPs showing distribution of IN subtypes under TF perturbations and distributions of subtypes across perturbations in human, macaque and 3D human samples, (Right) Scatterplot of gene expression log fold change in human, macaque and 3D human samples.

Lineage: the same progenitors can give rise to excitatory and inhibitory neurons in the developing cortex. This is very intriguing and the authors should confirm these results with live imaging. Ideally, this should be performed in a 3D human slice by infecting a RG and following up divisions and fate.

How can the authors be sure they are infecting a single cell and not multiple cells with the same virus? how many cells are infected with more than 1 viral particle (different virus)? Can they check after a few hours?

We agree with the reviewer that it is important to more extensively assess the shared origins of dorsal INs and ENs. Detailed investigation on shared origins of cortical ENs and INs can also be found in Keefe, Steyert & Nowakowski 2025, now accepted in Nature, where 3D human culture coupled with lineage tracing, as well as in vivo IHC validation were performed across midgestation to show presence of cortical derived inhibitory neurogenesis.

The STICR library includes sufficient barcode diversity to rule out infecting multiple cells with the same barcode. Delgado et al. 2021 Nature calculates that the viral library could be used to label more than 250,000 cells before reaching an estimated barcode collision rate of around 0.5%. Each batch in this study captures an average of 32,917 cells resulting from 7155 mother cells, which results in 0.0143% collision rate. We therefore reasoned it is unlikely that barcode clashing has biased our conclusions.

In terms of infecting the same cell with multiple barcodes, 66.7% cells in this study were detected to have single clonal barcodes. Presence of multi-infected cells will not affect our conclusions due to the following reasons:

- Cells encoding more than one barcode will be recognized and assigned to the same clone with other cells with the same expression patterns. Double labeling, though rare would add stringency to clonal assignments.
- Clones that have cells with conflicted sgRNA assignments were removed from downstream analyses.

Clonal size is also an issue without live imaging: how can the authors control for clone size? Some cell types could be more or less sensitive to isolation and the clones could be larger.

We agree with the reviewer that we cannot speak for clone size *in vivo* due to possibly disproportionate cell loss during dissociation and single cell capture. We were very careful throughout the manuscript that we don't draw any conclusions on the proliferative capacity or actual clone sizes of control RG but only compare topology of fate biases between control and perturbations that were processed and captured in a pooled manner. We agree that the actual clone sizes on differentiation D7 are most likely larger than what we observe from the single cell data, but cell loss does not account for changes in clonal fate decisions that we observe under TF KD.

Evolution: considering the proportion of EN/IN cortex-derived neurons, isn't 47% unexpectedly high as a common origin of ENs and INs in macaques?

We thank the reviewer for pointing out the potential species differences in clonal composition. When comparing within humans, we observed that ENs and INs are more clonally related at later stages due to higher abundance of inhibitory neurogenesis. In GW21-22 human samples, 48% clones that have ENs also contained inhibitory neurons, similar to what we have observed in macaques at comparable stages in the 2D culture model (Review Figure 21).

Review Figure 21: Clonal composition in humans and macaques. (Left) Upset plot of cell class compositions in multi-class clones in early and late mid gestational human samples and stage-matched macaque samples. (Right) Clonal coupling between cell types.

The human-macaque-specific responses to TFs (ARX and ZNF219) are very interesting. It would be nice to functionally prove the function of these TFs in evolution.

The responses to ARX and ZNF219 are largely conserved between humans and macaques in both cell type composition (**Review Figure 10**) and gene expression responses (extended data Fig. 8j). Against this background of conservation, we note that a subset of gene expression responses are also lineage-specific, indicating that our approach can be applied to study the evolution of membership in conserved GRNs. Future studies incorporating comparable multiomic data in rhesus macaque could assess whether altered enhancers containing motifs for the upstream TFs in only one species mediate the subset species-specific responses.

We are grateful for constructive feedback from reviewers and have revised the manuscript based on new data generated and analyzed during review. The major comments related to better demonstrating the technological novelty of our screening system to study human neurogenesis and to extending the biological novelty illuminated by the system. Below, we summarize major revisions addressing these concerns, ahead of the detailed point-by-point response:

The **technological novelty** of our system enables dissection of mechanisms underlying cell cycle dynamics and cell fate choice not possible with existing human cortical CRISPR systems by:

- **Increasing the fidelity to normal cortical development.** We have now performed extensive comparisons across methods highlighting the improved fidelity of our primary culture neurogenesis model revealing more precise cell type homologies (**Review Figure 1A**), reduced metabolic stress including glycolysis, ER and oxidative stress than iNeurons, iPSC- and primary- derived cortical organoids (**Review Figure 1B**).
- **Improving the reproducibility of cortical patterning.** We have now compared the primary samples that we use, which have been patterned for 16-22 weeks *in vivo* to organoids. We find that organoids, which endure long-term stress, show less than 50% cortical patterning in recent screens (PMID: 36198796) and in a comprehensive atlas of organoids (PMID: 39567792). Similarly, induced neurons, which forego normal patterning, show less than 50% CNS patterning with cells best resembling PNS neurons (PMID: 34358451) (**Review Figure 1C**).
- **Improving the scalability across individuals.** We have now expanded our screens to 21 biological replicates across 2 species and 3D organotypic slice culture model now included with our new experiments, demonstrating the scalability of our approach and potential to harness diverse genetic backgrounds and the fidelity of *in vivo* patterning (**Review Figure 1D**).
- **Enabling sensitive detection of cell fate consequence.** The week-long radial glia (RG) expansion phase that we implemented allows for CRISPRi activity to be restricted to the progenitor stage rather than manifesting in cells differentiated to various stages (**Review Figure 1E**). We have now performed additional validation experiments for PAX6 and EOMES, which have known cell fate consequences, as suggested, supporting the sensitivity of our system (**Review Figure 1F**). We further show, as suggested, that our 2D system further recapitulates gene expression and compositional consequences of TF knockdown in gold-standard human organotypic slice culture (**Figure 1G, J**), while also uniquely providing the RG expansion point for localizing CRISPRi gene repression to RG.
- **Increasing temporal precision.** Organoid RG show variable maturation stages in the same organoid (PMID: 30735633; 31996853), while induced neurons short circuit physiological RG altogether. In contrast, our system allows us to survey defined *in vivo* maturation stages and recapitulates a known sequence of temporal cell fate decisions at the individual clone level (**Review Figure 1H-I**), a feature we now harness to enhance the biological novelty of our findings (below).

These technological advances enable **biologically novel findings** that we expanded in revision based on reviewer comments. In addition to our findings about the convergence of neurodevelopmental disease-linked genes downstream of multiple TFs (hub effector gene model) and of a critical window for engineering EN/IN plasticity, relevant for disorders driven by E/I imbalance, we have now expanded our findings to illuminate biological novelty relate to RG maturation, neuron subtype-specification, and disease gene mechanisms.

- **TFs regulating heterochrony:** We surveyed additional stages of maturation (9 total samples spanning 6 weeks of *in vivo* patterning) combining perturb-seq with lineage tracing. Joint profiling of gene expression and lineage allowed us to reconstruct sequential stages of RG maturation and cell fate choice recapitulating a known sequence of development events. Combining lineage and perturbation data revealed that ARX and NR2E1 have opposing roles in regulating the developmental tempo and cell fate choice in the human cortex. In both human and rhesus, ARX knockdown consistently transitioned RG to more immature lineage potentials while NR2E1 consistently transitioned NR2E1 to

more mature stages (**Figure 1H-I**). We added an updated Figure 4 focused on the biological theme of conserved TF mechanisms regulating heterochrony of cell fate choice revealed by the study.

- **TFs safeguarding IN Identity:** Within the IN lineage, we discovered a conserved role of ARX and SOX2 in safeguarding normal post-mitotic IN identity. Our findings add a novel cellular mechanism to ARX-linked lissencephaly and epilepsy by revealing that in addition to impaired migration, inhibitory neurons with reduced ARX adopt an ectopic cell state – conserved across species – and by providing a set of the earliest dysregulated genes in human primary cells rather than mouse models. To buttress the finding that these stable and conserved ectopic states exist outside of normal morphospace, we compared with existing mammalian brain atlases, and found only a modest correlation with known developmental populations. We further repeated this experiment in 3D organotypic slice culture, which corroborated the 2D findings and supported the prediction from lineage tracing that this ectopic state is manifest by postmitotic effects on subtype specification, rather than by effects on RG. (**Figure 1J**).
- **Epistasis between ARX and LMO1 driving ectopic IN state:** We have now demonstrated that the transcriptional consequences and ectopic interneuron state caused by loss of ARX are driven, at least in part, by aberrant expression of LMO1, normally repressed by ARX. By performing dual repression of ARX and LMO1, we found that ARX, LMO1 double KD enriched for an intermediate IN state, representing a partial rescue of gene expression changes driven by ARX (**Figure 1J,K**), now presented in a new Figure 5 focusing on the biological theme of TFs safeguarding post-mitotic inhibitory neuron identity.

A. Improved Fidelity

B. Reduced cellular stress

C. iNeuron and Organoid Heterogeneity vs. CNS

D. Scale across individuals

E. Sensitive RG fate detection

F. Validation with known targets

G. Reproducible in organotypic slice culture

H. RG lineage progression at the clonal level

I. Opposing effects on maturation by ARX and NR2E1

J. Conserved ectopic state

K. Partial rescue by ARX, LMO1 dual repression

Review Figure 1: Technological and biological novelty of this study.

A. Pearson correlation of top 25 cell marker gene expression defined in Wang 2025 in vivo atlas using data in this study (left) and in He 2024 organoid atlas (right), with a boxplot showing quantitative differences between the two systems.

B. Gene expression scores of 5 types of cell stress (glycolysis, ER stress, reactive oxygen species (ROS), apoptosis and senescence) in 6 different datasets including vivo human atlas (PMID: 39779846), 3D organotypic slice culture, 2D primary RG culture, iNeurons (PMID: 34358451), primary brain organoids FeBO (PMID: 38194967) and organoids (PMID: 39567792) obtained from published studies.

C. Left: Region annotation of public organoid datasets used in He et al., 2024 (PMID: 39567792), obtained by reference mapping to a first trimester human brain atlas in Braun et al., 2023 (PMID: 37824650); Right (Lin et al., 2021, PMID: 34358451, Fig. 3E): Induced neurons and organoids show a substantial fraction of non CNS cells.

D. Biological replicates included in this study. New samples added during review are highlighted in blue, bringing the total from 12 to 21 individuals (and 2 species) across experiments coupling CRISPRi and scRNA-seq, compared with typically one individual in organoid and induced neuron studies.

E. Left: Schematics showing timeline of cell differentiation and CRISPRi in 2D and 3D organotypic slice culture; Right: robust KD at the protein level in 2D RG before differentiation.

F. Changes in SOX2, EOMES and NEUROD2 expressing populations in PAX6 and EOMES KD. Increase in SOX2 and decrease in EOMES and NEUROD2 populations were detected in both perturbations under paired two-sided Wilcoxon tests across 4 individuals.

G. Correlation of log fold change of differentially expressed genes under perturbations identified in 2D and 3D organotypic slice culture, and (bottom right) cell type abundance under perturbations in slice culture.

H. Top: UMAPs of clones colored by pseudotime and fate biases; Bottom: Pseudotime distribution in clones grouped by stages and perturbation.

I. Left: Distributions of clonal cluster fractions in 8 human individuals in NR2E1 and ARX KD in comparison to NT, grouped by clonal clusters, highlight opposing roles of ARX and NR2E1 in regulating cell fate potential; Right: Clonal pseudotime distribution between species, suggesting conserved properties in rhesus macaque.

J. Distributions of IN subclusters in each TF perturbation in human (top) and macaque (middle) 2D and in human slice culture (bottom), colored by IN subtypes, highlighting conservation of the ectopic IN_LMO1/RIC3 subtype mediated by ARX repression.

K. Left: Schematics for gene expression regulation under NT, ARX KD and ARX, LMO1 double KD (dKD). Middle: UMAPs for differential abundance testing in IN subtypes using Milo in NT, ARX KD and dKD. Neighborhood graph of differential abundance testing colored by log₂FC (FDR=0.05). Right: Beeswarm plots showing differential abundance in neighbourhoods at the clonal cluster level.

Point-by-point:

Referee #1 (Remarks to the Author):

We thank the reviewer for their comments - deeper analysis of the fidelity of our primary cell CRISPRi model for studying human neurogenesis has allowed us to much more clearly convey the technological novelty of our new approach. And we apologize for miscommunications in our methodology in the manuscript. We now clarify the concerns about experimental clone size, statistics, and fidelity below, which strengthen our conclusions.

Ding et al. constructed TF-enhancer-gene eGRNs to identify 44 TFs predicted to have strong impacts on transcriptomes when perturbed. Using CRISPRi knockdown in human primary radial glia (RG) cultures, they studied the transcriptional and cell fate consequences of these TFs in cortical neurogenesis. Their findings reveal multiple TFs that bias differentiation between excitatory neuron and interneuron lineages. Additionally, they identified effector genes that were convergently perturbed across multiple TF knockdowns, highlighting potential roles in neurodevelopmental disorders.

While this work provides insights and novelties of a human RG culture system, concerns remain regarding the model validity and power sufficiency. The 7-day RG culture generates small clones (~4.6 cells), considerably lower than the estimated naïve clone size (~15 cells). This discrepancy raises questions about whether the culture accurately models in vivo conditions and perturbation relevance.

We agree with the reviewer that further addressing the fidelity of our model to normal development is an important component of presenting our study. We now include evidence for substantial improvements in the

fidelity and cultural health of our model by benchmarking to other models including iNeurons, iPSC and primary cell derived organoids. We respectfully note that the comment on smaller clone size reflects a property of the incomplete sampling of clones by 10x scRNAseq and the shorter timing of our experiment rather than reduced fidelity of our model.

To clarify the timing: our culture system involves both a 7 day RG expansion culture after viral infection to ensure that CRISPRi knockdown is achieved prior to differentiation and cell fate choice (improving the sensitivity to detect cell fate phenotypes), followed by an additional 7 days in conditions permissive of both self-renewal and differentiation to study how GRNs influence these dynamics. Our system further enables surveys across multiple stages of maturation that recapitulate known temporal changes in fate potential.

Importantly, the comparison to naive clone size suggested by the reviewer as flagging the accuracy of our model is not an apples-to-apples comparison because we are sampling lineages rather than comprehensively measuring them. The need to perform sampling rather than detecting entire clones arises from at least 50% cell loss during FACS, encapsulation for scRNA-seq by 10x Genomics, joint amplification of sgRNA and cell lineage barcodes, and conservative assignments of these to cell barcodes. The clone size we detect here is akin to genes detected in scRNAseq as a technical metric and not a statement of the absolute number of events.

Therefore, the clone size in a 14 day post infection experiment followed by FACS and droplet-based scRNA-seq is not a fair comparison to naive clone size, which itself would depend on stage and duration examined.

Nevertheless, we appreciate the sentiment to more rigorously evaluate the extent to which our system improves on existing models in terms of physiological relevance and we do so in several ways, described below, that strengthen the manuscript.

Additionally, weak statistical significance in several analyses warrants further examinations (e.g., DEGs are not significant using traditional tools like DESeq2; the tool they used, TRADE, does not account for statistical dispersion; TRADE specifies that null hypothesis testing which the authors omitted and may have inflated the p-values; several analyses used a lenient FDR < 0.1 cutoff).

We agree with the reviewer that it is important to ensure statistical power in analyses used in this study and that traditional tools like DESeq2 are ideal for these analyses. We apologize for any misunderstanding arising from our use of the TRADE workflow.

To clarify, all the DEG results (logFC and FDR corrected p values) and related panels used in this study are identical to DESeq2 output and account for statistical dispersion (Review Figure 2). While TRADE estimates an overall transcriptome-wide impact based on logFC and unadjusted p value from DESeq2, it does not modify p-values for individual genes. The TRADE output for significant DEGs are therefore identical to the initial DESeq2 results without inflation. This was simply a communication issue on our part.

We apologize for the miscommunication and have removed TRADE from our pipeline to simplify the workflow and minimize confusion. All analyses included in the revised manuscript directly use DESeq2 output.

Review Figure 2: Identical p values obtained from TRADE and DEseq2 output. Red dotted line denotes $y=x$.

Major:

1. One of the major novelty is the use of primary RG cell culture – but the validity remains additional evidence. Similar screens (at much higher scale) has been done by iPSC-derived NPCs and neurons (Tian Nat Neuro 2021): what are the unique advantages of this RG system?

We thank the reviewer for encouraging us to emphasize the unique advantages of the screening system we introduce, and we agree that further evidence was required. We have now performed extensive additional analysis along multiple dimensions of fidelity, reproducibility, cell health and stress response to address this point, and we highlight improvements below.

Tian Nat Neuro 2021 performed a neuronal survival screen on induced neurons from a single individual with no phenotypes measured for cell type composition or gene expression. On the other hand, Zheng et al., 2024 introduced a new perturb-seq system in vivo in mouse and screened a small number of genes, but provided more physiological relevance and richer scRNAseq data.

A unique feature of our screen – that we now communicate more clearly – is that we inactivate genes specifically in radial glia and then allow differentiation dynamics and cell fate to occur normally, and we can do this at scale across individuals, species, and precise maturation stages with distinct fate potentials, features that we now better illustrate, including through expanded biological findings.

Other CRISPR systems do not offer this feature of CRISPR-based repression restricted to progenitors. Induced neuron systems drive artificial cell cycle exit and glutamatergic neuron cell fate making it impossible to study how perturbations affect either of these cell behaviors. Induced neurons also represent an artificial low-fidelity patterning system that produces a combination of peripheral and central nervous system neurons, with mixed identities, limiting the physiological relevance, while organoid models also show inconsistent patterning, lower fidelity, and increased metabolic stress (e.g., PMID: 34358451, 30735633; 31996853; 39567792; Review Figures 3-6). In vivo studies in mouse models represent a high level of fidelity to organismal development, but cannot address human-specific phenotypes. In addition, viral delivery of CRISPR machinery in vivo may also permit normal differentiation of infected cells to occur prior to expression, translation, and activity of Cas9, potentially affecting the sensitivity of the system for composition phenotypes in radial glia lineages.

We optimized our system to detect the consequences of perturbations in radial glia on differentiation dynamics and cell fate choices known to occur in vivo. Our innovation to expand radial glia at the progenitor stage for one week in 2D culture allows us not just to localize infection to radial glia, but also to ensure that knockdown occurs at the radial glia stage (which takes ~1 week to achieve in human cells with current CRISPR tools, (e.g., Gilbert al., 2014; PMID: 25307932) (**Review Figure 1E**, Extended Data Fig. 1c). This approach allowed us to identify sgRNAs affecting cell cycle, differentiation dynamics, and excitatory vs. inhibitory neuron cell fate choice – all known to occur in vivo – that would be difficult to detect if CRISPR machinery was induced at later and variable stages of these dynamic processes. We now perform additional experiments on genes with known composition effects, including PAX6 and EOMES to further validate the system (**Review Figure 1F**, Extended Data Fig. 1c). Together with the permissive differentiation conditions that we optimized here, the combination of innovations in the 2D system allows us to detect cell fate consequences not possible in any existing human neural perturb-seq system (**Review Table 1**).

We have further performed perturb-seq experiments in organotypic 3D slice culture during review to compare with the 2D system. The 3D system confirms the physiological relevance of the cell type-specific gene expression consequences we observe in 2D. Both the 2D and 3D primary models show reduced metabolic stress compared to organoid models (see below, **Review Figure 7**), further highlighting the physiological relevance of our system. The strong correlation of gene expression consequences between the 2D and 3D organotypic models (see below, **Review Figure 8**) described below further adds confidence to the 2D studies that maximize sensitivity for RG lineage phenotypes.

Review Table 1. Different systems for Perturb-seq

Systems	Induced Neurons	Organoids	Primary RG 2D	Organotypic slice	in vivo Mouse
Fidelity.to.in.vivo	-	+	++	+++	++++
Human.genetic.background	+	+	+	+	-
RG.maturation	-	+	+++	+++	+++
Recapitulation.of.cell.fate.choice	-	+	+++	++	++
Scalability.across.genetic.background	++	++	+++	+	-
Cell.health	++	+	+++	+++	++++
Genetic.epigenetic.stability	+	+	++++	++++	++++

Table 1. Comparison of different CRISPR systems used to study cortical development.

h

This study- 2D

He et al., 2024 - Organoids

Review Figure 3 (Extended Data Fig. 1h): Comparison of cell type fidelity in the primary 2D RG system developed in this study and iPSC-derived organoids. Pearson correlation of top 25 cell marker gene expression defined in Wang 2025 (PMID: 39779846) in vivo atlas using data in this study (left) and in He 2024 organoid atlas (right). Cell type annotation was done using reference mapping of both datasets to in vivo atlas.

o The cell types in the RG culture appear significantly different from naïve human cortex. Pearson correlations in Fig. S1e are weak ($r \approx 0.2-0.4$), suggesting substantial divergence from naïve, in-vivo states. A differential expression analysis comparing this culture with naïve cortical tissue would help clarify these differences? How different are these cells from naïve states, and are they too different to be useful for perturbation studies?

We agree with the reviewer on the importance of measuring fidelity and thank the reviewer for encouraging us to expand this section. The reviewer may have misread the heatmap in Fig. S1e. The r values for the maximum correlation coefficients in major cell types are actually 0.72 (RG/Astro), 0.68 (EN), 0.50 (IN) with an average value of 0.60 across all cell types. These correlation values are high, despite correlating whole cell scRNAseq data from our study to single nucleus multiome data collected in another lab, and should be considered conservative given differences in cellular compartment, chemistry, and laboratory.

Based on the reviewer's comment, we have more directly compared the fidelity of our system to a comprehensive atlas of organoid datasets (PMID: 39567792). When using predicted cell type annotation through reference mapping from Wang 2025 Nature (PMID: 39779846), we observe an average Pearson correlation R of 0.62 to corresponding cell populations in vivo, representing a near 2-fold increase in fidelity compared to 0.34 in organoid datasets from He 2024 Nature (Review Figure 4), which typically fail to consistently pattern to telencephalon (Review Figure 5, 6). Similarly, induced neurons, frequently used to

study cortex, best resemble the peripheral nervous system, with only a low fraction patterning to any central nervous system regions (Review Figure 5). Additionally, the primary cell culture model we introduce substantially reduces endoplasmic reticulum stress, ectopic glycolysis and oxidative stress that characterize organoid models (Review Figure 7).

Thus, more in-depth analysis encouraged by the reviewer has now allowed us to provide quantitative support for the substantial improvements in fidelity and reproducibility of human cortical models our system offers, while also enabling gene inactivation of target genes in radial glia themselves to sensitively measure cell fate consequences.

Review Figure 4 (Extended Data Fig. 1j): Pearson correlation to predicted cell types in this study compared to organoid models. Boxplot to show distribution of Pearson correlation to predicted cell types (diagonal values from Review Figure 3).

[REDACTION]

Review Figure 5: Presence of mispatterned non-telencephalic cells in organoid models and induced neurons. Left (reproduced from Fleck et al., 2022, PMID: 36198796, Fig. 1d) with example patterning variation. Middle (reproduced from Lin et al., 2021, PMID: 34358451, Fig. 3C), highlighting that most iNeurons across individuals best correspond to the peripheral nervous system and retina with partial brain correspondence, Right (Lin et al., 2021, Fig. 3E) show that induced neurons and organoids show a substantial fraction of non CNS cells.

Review Figure 6: Variation and heterogeneity in patterning efficiency in organoid models. Barplot showing composition of cell identities in data from He at al., 2024 Nature, PMID: 39567792. Cells were reference mapped to an in vivo first trimester human atlas described in Braun et al., 2023, PMID: 37824650, Science for region annotation. Note that ‘Telencephalon-containing’ includes cells that are mapped to telencephalon as well as forebrain, brain, and head dissections that include telencephalon. Presence of mispatterned cells (Cerebellum, Diencephalon, Hindbrain, Medulla, Midbrain, Pons) and variation between studies were observed across studies.

Review Figure 7 (Extended Data Fig. 1k): Comparison of metabolic stress in human model systems. Violin plots of gene expression scores of 5 types of cell stress (glycolysis, ER stress, reactive oxygen species (ROS), apoptosis) in 6 different datasets including vivo human atlas, primary organotypic slice culture, 2D

primary RG culture, iNeurons(Lin et al. 2021), primary brain organoids FeBO (Hendriks et al. 2024) and organoids(He et al. 2024) obtained from published studies.

o The clone size in STICR experiments (~4.6 cells) is much smaller than expected for naïve RG, which is concerning. How might this affect differentiation potential and phenotypic robustness?

Delgado et al. (PMID: 34912114) observed a median of 23 cells per clone after 6 weeks of culture. It is unsurprising that the clone size is lower in our experiment, sampling after 2 weeks in culture.

We further note that the clone size detected depends on the fraction of all GFP cells collected during FACS, which can differ between experiments based on the sorting efficiency and the capacity to load sorted cells into the 10x instrument.

Our study does something that has never been done before in human cortical neurogenesis - combine perturb-seq and lineage tracing. This requires sorting for dual infected cells, and maintaining the low multiplicity of infection to ensure a single perturb-seq lentiviral particle per cell means that we are sorting for ~1-2% of co-infected cells in the system. As such, this reduces the efficiency of capturing the entire lineage tree for each clone. In addition, conservative assignments of both sgRNAs and lineage barcodes to cells can further reduce the detected clone size.

Instead of estimating overall in vivo clone size, we use the topologies of observed lineage trees to identify how perturbations affect clonal fate biases. As a testament to the fidelity of our system to in vivo development and as an example of biological findings from lineage trees, we have been able to capture the hypothesized transcriptional profile of human “tri-IPCs” - radial glia in mixed inhibitory and oligo lineages as molecularly distinct from radial glia in excitatory only lineages, recapitulating the proposed markers from mouse experiments PAX6, GLI3, EGFR, and OLIG2 as the top distinguishing features between these clone types (PMID: 32234482), and supporting a novel homology between human and rodent development.

Further, we have been able to identify perturbations whose clonal fate phenotypes provide answers to developmental mechanisms underlying the cell composition effects that we observe in the initial screen.

In addition, sampling clonal lineages allowed us to study how perturbations affect the cell fate choice over developmental time.

o The dominance of interneuron populations also raise concerns about potential subpallial contamination during dissection or artifacts from RG dissociation. How do the dorsal INs compare with other IN populations (LGE/CGE-derived INs) in gene expression and relative composition? How do they align with existing data (e.g., Delgado et al.)?

Contamination of mis-identified tissue is unlikely to account for our results for several reasons:

-First, contamination cannot account for the presence of EN clones and EN/IN mixed clones, which are only produced in cortex.

-Second, contamination cannot account for the reproducible temporal gradient we observe that the fraction of mixed clones and IN biased clones increases with sample age. Collection of an additional 9 individuals further supports this age-dependent pattern, highlighting a biologically relevant pattern that may correspond to the shift late in mouse cortical neurogenesis to produce inhibitory neurons known to migrate to olfactory bulb.

Indeed, we find the first clonal lineage evidence of a shared transcriptional state between primate IN-biased clones and the reported tri-IPCs analyzed by lineage tracing in mouse.

-Third, sgRNA composition effects that we detect are compared to a baseline in the same pool and would be robust to possible contamination.

-Fourth, all samples show a high fraction of cells expressing cortical markers (e.g., EMX2, PAX6 etc).

-Fifth, we have now included combined lineage tracing and perturbation data performed in cortical organotypic slices, which shows strong consistency with the 2D data in the level of baseline lineage topology and response to perturbation (**Review Figure 8d-f**). Notably, in the organotypic slice culture data, cortical born INs are transcriptionally distinct from MGE and CGE-derived inhibitory neurons, further arguing against subpallial contamination.

We appreciate the reviewers comment to compare with other CGE/LGE populations. Below, we highlight molecular differences between IN subtypes identified in our organotypic cortical slice culture (**Review Figure 8**):

The Immature locally born IN cluster (IN_local), clonally coupled to cortical RG and immature ENs, is adjacent to 2 subtypes of cortical INs that have OB or CGE like molecular identities, respectively.

IN_local highly expresses dorsal marker PAX6 together with other LGE/CGE markers MEIS2, CALB2 and PROX1, as well as immature markers (ST18, NPY) compared to other IN subclusters, but lowly express migration markers (ERBB4, CXCR4). These observations are in line with existing data from Delgado 2022 (PMID: 34912114), and Keefe, Steyert & Nowakowski 2025 (in press, **Review Figure 9**).

Review Figure 8 (Extended Data Fig. 9a-g): Organotypic cortical slice culture and molecular identities of different IN subtypes.

- Experimental design for targeted TF perturbation and lineage tracing in organotypic slice culture of 4 human individuals. CRISPRi and STICR lentivirus was locally co-injected to the germinal zone to enrich for RG populations.
- UMAPs showing cell type annotations and individuals (top right), with a UMAP of newborn progeny after filtering for cells from multicellular clones on the bottom right. Note that large proportions of postmitotic ENs were labeled and captured but not clonally linked to other cell types.
- Dotplot of the expression of cell type markers for the assigned clusters (left), with a barplot showing numbers of cells detected in the cluster (right). The size of each dot denotes the fractions of cells in the group where the gene is expressed and the color denotes mean gene expression in the group.
- Scatterplot showing log2FC in human 2D and slice for DEGs identified in ARX, NR2E1, SOX2 and ZNF219 KD. Pearson correlation coefficients and p values were calculated in each TF perturbation and labeled together with regression line equations. Red dashed line represents $Y=X$. An average Pearson correlation $R = 0.6$ was observed across perturbations.
- Barplot showing cell class compositions of cells in multicellular clones in each perturbation.
- Heatmap for lineage coupling score matrix in major cell types in the RG lineage. IN_local represents locally derived INs in culture that shares lineage with RG and ENs.
- Dot plot showing markers for different IN subtypes.

g

Review Figure 9 (Keefe, Steyert & Nowakowski 2025 Nature, In Press, Fig. 3g, j): Molecular identities of different IN subtypes in organotypic cortical slice culture. (left) Schematics for experimental design in Keefe, Steyert & Nowakowski 2025 (right) Dotplot showing marker gene expression in locally derived INs (IN_local) compared to GE derived ones.

o How does this culture compare to iPSC-derived NPCs or neurons, where Perturb-seq has been widely applied? Does it reveal phenotypes that iNs cannot?

We thank the reviewer for raising this question as the richer neurogenesis phenotypes across stages of radial glia maturation represent important advantages of our system.

Perturb-seq has been widely used in induced neurons and is possible in early NPCs and organoids, though silencing of CRISPR machinery during longer differentiation experiments has limited its use. However, these iPSC systems have major limitations for interrogating the dynamics of neurogenesis. As mentioned above, induced neurons have limited fidelity to normal development with NGN2 neurons often producing a mix of CNS and PNS neurons (**Review Figure 5**). The process of forced transcription also overrides normal development making it difficult to study cell cycle and cell fate choices in an artificial model. Meanwhile, early NPCs are transcriptomically distinct with maturation times though to reflect *in vivo* timescales (PMID: 34535062), and radial glia can vary in maturation stage, even in the same organoid variable (PMID: 30735633; 31996853) making it difficult to model mid-fetal cortical neurogenesis, a developmental stage when important neural cell fate events, including the transition from excitatory neurogenesis to inhibitory neurogenesis and gliogenesis take place.

In contrast, our system enables sensitive detection of perturbations that affect cell fate choice at defined stages of radial glia maturation that recapitulate cell fate choice and lineage plasticity that occurs *in vivo*.

Our system also compares favorably along several dimensions:

iPSC-derived models experience cell culture acquired genetic and epigenetic changes that can confound results (PMID: 28445466; PMID: 28017796; PMID: 35885940; PMID: 34407428) and the engineering of individual cell lines has typically limited screens to a single individual. This is problematic as screens are performed in a single genetic background, limiting generalization.

Even for commonly used iPSC lines, the genetics is often compromised. The KOLF2.1J iPSC line, introduced to standardize iPSC lines and overcome these problems has CNVs causing haploinsufficiency in 3 neurodevelopmental disorder genes (PMID: 37425875). The line CW20107 selected by SSpsygene and multiple CIRM projects for expensive engineering shows subclonal aneuploidy of Chr1 duplications plus additional subclonal aberrations. Clones of the Tuba1-GFP and the commonly used WTC-11 iPSC line in circulation for induced neuron experiments also have structural variants and/or aneuploidy that increases with prolonged passaging (PMID: 35885940).

In contrast, our primary cells do not go through clonal bottlenecks or prolonged cell culture (19 days in culture) versus low-efficiency clonal isolation and typically 15 - 50 passages prior to even starting a differentiation experiment.

In a previous attempt to overcome the limitations of screening in a single genetic background, we spent several years engineering identical CRISPRi machinery into multiple safe harbor sites in iPSCs from 6 human and 6 chimpanzee, and one orangutan individual (PMID: 37343560). While this worked for iPSCs, most lines showed variable silencing of CRISPRi machinery after 2-3 weeks of neural differentiation (and some lines showed P53 hypofunction), making it difficult to scale CRISPR-based approaches by engineering iPS lines.

In contrast, our introduction of an all-in-one direct capture vector and the primary cell system allows us to scale our approach to many individuals and multiple species. Here we have performed experiments, including Perturb-seq and arrayed CRISPRi followed by flow cytometry, in 26 individuals from two species. Our ability to cryopreserve primary samples also lets us compare different age samples in common culture and cell capture

batches further mitigating technical batch effects, while surveying a greater and more well-defined landscape of normal human development.

o Additional validations, such as tissue-level imaging, donor-to-donor reproducibility, and assessments of culture health, would bolster confidence in this system.

We appreciate the reviewer's comment to assess reproducibility and culture health. We have calculated glycolysis, ER stress and oxidative stress scores and directly compared them with other model systems raised by the reviewers, including human datasets in vivo, in iNeurons and in iPSC or primary cell derived organoids (**Review Figure 7**). We found that the cultural system presented in this study had lower metabolic stress than iPSC-derived neurons, iPSC-derived 3D organoids and primary FeBO 3D organoid models.

As discussed above, organotypic slice culture data reproducing phenotypes in 2D also supports the fidelity of our system. Also note here that previous studies have recapitulated relevant cell type behaviors such as outer radial glia mitotic somal translocation in 2D primary cultures (PMID: 25088420; PMID: 26406371).

Review Figure 7 (reproduced from above): Comparison of metabolic stress in human model systems.

Violin plots of gene expression scores of 5 types of cell stress (glycolysis, ER stress, reactive oxygen species (ROS), apoptosis) in 6 different datasets including vivo human atlas, primary organotypic slice culture, 2D primary RG culture, iNeurons(Lin et al. 2021), primary brain organoids FeBO (Hendriks et al. 2024) and organoids(He et al. 2024) obtained from published studies.

We also thank the reviewer for raising the importance of emphasizing donor-to-donor reproducibility in our study. Throughout the initial study, we included 9 human individuals and 3 macaque individuals in the single cell experiments, and 9 human individuals (6 not included in the single cell workflow) in the flow cytometry validation. During review, we repeated our single cell workflow on another 5 individuals (human, n=4, 3 not included in the flow cytometry; macaque, n=1) in 2D and 4 human individuals in 3D slice culture spanning the stage of excitatory and inhibitory neurogenesis and early gliogenesis (**Review Figure 1D**). Cross validation from a total of 26 individuals (human, n=22, macaque, n=4) confirmed that the phenotypes that we are reporting are robust across genetic backgrounds. Biological replicates are already included as a parameter in our models, and we observe consistency across sgRNAs (**Review Figure 13**, see below), cultural systems (2D versus slice, **Review Figure 8d,e**), species (human versus macaque, Extended Data Fig. 6k,m,n) and batches (2D initial screen and 2D targeted screen, Extended Data Fig. 6k,l), with n>=4 individuals in all experiments.

2. Statistical power and sampling size in the study is another concern. The study's design (400~500 cells/gene; Fig. S1i) translates to ~56 cells/perturbation/cell type across eight cell types—considerably lower than recent screens (e.g., Replogle et al., Cell, 2022; ~200 cells in a genome-wide screen; Li et al., Nature, 2023; ~80 cells per perturbation per cell type in organoids). Given this low sample size, can the authors justify the screen's statistical power? How are the effect observed across batches/replicates? Perturb-seq downsampling analysis is a routine – is the current cell number well powered to reach conclusions? Additional replicates or cell numbers may improve reproducibility.

We thank the reviewer for raising the importance of reproducibility.

The initial screen in this study collected an average of 600 cells per perturbation and 75 cells per cell type. This number is greater than Li et al., Nature, 2023(PMID: 37704762) which reported 64 cells per cell type across 17 cell types.

Effects of perturbation for both cell type composition and gene expression were calculated accounting for biological replicates. Four biological replicates were used for the initial screen and 8 more for the targeted lineage tracing study. In revision, we further added 4 more human individuals, 1 more macaque individual for 2D experiments, and 4 more human individuals for validation in organotypic slice culture. We observed consistency between individual replicates (Review Fig 10), batches (Extended Data Fig. 6k,l), sgRNAs (Review Figure 13) and species (Extended Data Fig. 6k,m,n) and cultural systems (Review Figure 8d,e) across the 21 individuals for which we collected single cell data.

We have implemented downsampling in the differential composition analysis in this study and here further show the robust detection of consistent phenotypes across conditions tested (200-600 cells/gene) (Review Figure 11), indicating that our initial screen is well powered to detect effects of different perturbations. The validation of these phenotypes in secondary screening from independent samples further supports the reproducibility and generalizability of our findings.

Review Figure 10: Reproducibility across biological replicates. Cluster-free differential abundance testing using Milo at the individual level using sgRNA as replicates. Nodes are neighborhoods with sizes representing the number of cells in each neighborhood. The edges depict the number of cells shared between neighborhoods and the layout of nodes is determined by the position of the neighborhood index cell in the UMAP. The nodes are colored by their log₂FC when perturbed.

Review Figure 11: Robustness to downsampling. Left (Extended Data Fig. 3b): Scatterplots of logFC of cell abundance in downsampled populations (x axis) and in all cells collected under each perturbation (y axis). Each condition was randomly downsampled to 300, 400 and 600 cells for this testing. Red dashed line represents $Y=X$. Right (Extended Data Fig. 4d): Cluster-free differential abundance testing using Milo after downsampling to 200-600 cells shows consistency in perturbations that have strong (ARX, NR2E1) or weak (POU2F1, BHLHE22) compositional phenotypes.

3. I am concerned about the DEG statistical tests for several reasons, as the authors used a somewhat unusual pipeline:

- o For DESeq, the authors only check that more than 1 cell is has the perturbed condition within the cell type they are looking for – this is a very low cell number threshold..
- o The authors then send a few outputs to TRADE: only taking in the un-adjusted P-value and does its own FDR and Bonferroni correction, but this does not account for statistical dispersion? It also cannot account for the low statistical power that has been already introduced.
- o All the figure panels are made with the TRADE output, and not the DESeq output. I wonder how many statistically relevant genes that DESeq identifies? If at all, do they overlap with the TRADE analysis?
- o TRADE itself specifies that null hypothesis testing should be done with a jackknife sterr method, which involves rerunning the pipeline many times with 1 batch leave-out. The authors don't seem to do that anywhere in their code though, and this may lead to inflated statistical values.

We agree with the reviewer the importance of ensuring proper statistical testing for our analysis and thank the reviewer for the detailed, thoughtful comments. We apologize for the miscommunications of our workflow:

1. We applied a threshold of minimum 2 cells per replicate for pseudobulking and minimum 2 replicates for DESeq2 statistical testing in each cell type and perturbation. Also note that conditions that have 2 or lower cells apply to rare cell populations that are not the main focus of this study (microglia, vascular, etc) and therefore does not impact our conclusions. We have now rerun this analysis with a minimum cell threshold of 5 and biological replicates of 2. We observed consistent results and have updated Fig. 2 and other related panels in the revised manuscript. All the new analyses added during revision have been set to 5 cells and 2 biological replicates as minimum threshold.

2. The TRADE output is identical with the DEseq2 output and therefore the FDR adjusted p-values are not inflated.
3. All the figure panels use p-values that are identical to DEseq2 output.
4. The null hypothesis testing that the reviewers mentioned applies to TRADE estimates of “overall transcriptional impacts” which we did not include in this study.

We apologize for the confusion and have updated the methods to improve clarity of our workflow.

Review Figure 2 (reproduced from above): Correlation plot to show identical p values obtained from TRADE and DEseq2 output.

4. Flow cytometry validation (Fig. S6) seems contradictory to the Perturb-seq results. Several single-cell phenotypes fail to replicate in flow assays. Notably, DLX2 perturbation does not alter interneuron composition, and Neurod2 excitatory neuron phenotype is inconsistent with single-cell data. Can the authors address these discrepancies?

We agree on the importance of validation by orthogonal assays, but the reviewer has misread Figure S6. Flow cytometry validation was only performed in perturbation of top TF candidates ZNF219, ARX and NR2E1. DLX2 perturbation was not performed. We have updated the figure panel and legend to improve clarity (Extended Data Fig. 5).

DLX2 positive populations were stained to quantify the changes in inhibitory populations in each KD condition and consistency was observed in all conditions. For example, the decrease in the EOMES+ and NEUROD2+ populations, and the increase of DLX2+ populations in NR2E1 KD mimics the decreased EN/IN ratio observed in the initial Perturb-seq screen and the follow-up lineage tracing experiments in both species. Additionally, 2 other differentiation timepoints were included to investigate the temporal dynamics of cell fate changes under perturbation, reinforcing consistency along the RG differentiation trajectory and informing the temporal onset of these composition changes.

5. The experimentally measured TF impacts vs. eGRN predictions are inconsistent: the 44 TFs were selected for their predicted strong influence on transcription, yet the observed effects (Fig. 2b) appear weaker than expected. Could this be due to insufficient statistical power or the culture system failing to replicate naïve tissue states (where the eGRNs were derived)?

The reviewer asks an important question raised by our study about why all 44 TFs do not have a strong influence on cell fate. We detect different numbers of altered genes depending on the TF perturbation and we only report on a subset with strong composition consequences.

As shown above, our screen is well powered to detect composition and gene expression changes under TF perturbations. Similarly, our system is closer to in vivo metabolic state than organoid models and better recapitulates the in vivo cell types where the eGRNs were derived.

Although statistical power is not an issue based on downsampling our strong hits, we expect that additional screening would add confidence to smaller effect perturbations.

One possible explanation for why more TFs do not cause strong cell fate consequences is that CRISPRi produces quantitative repression and not complete inactivation. Given the difference in knockdown efficiencies, we do not make strong claims that the absence of strong phenotypes for some TFs means those genes are unimportant.

However, we must note that the in silico perturbation tools are still in their infancy and may not account for compensation/redundancy, which is why performing the experiments is important.

We have included this discussion in the revised manuscript (line 153-156).

6. The study reports divergent phenotypic effects between human and macaque cultures. Could these differences stem from technical factors (e.g., RG survival biases due to culture conditions or cryopreservation conditions, or general age differences in donor tissues)? Testing key hits (e.g., NR2E1, ARX) across multiple developmental stages may help clarify whether these effects are age-dependent.

We thank the reviewer for the constructive feedback and agree that confounding effects from stage differences can lead to phenotypic differences we observe in the system. As mentioned above, during review, we have expanded our observation of single cell lineage tracing study to 9 independent human individuals from broader developmental stages and confirmed the stage-dependent effects of RG lineage plasticity and responses to TF perturbation in this study. For example, we identified ARX and NR2E1's capability to alter developmental tempo across multiple cell lineages, while this RG lineage plasticity for EN/IN balance appears to peak between GW16-19 (**Review Figure 12, Extended Data Fig. 7i**).

Review Figure 12 (Fig. 4): Heterochronic RG lineage progression under TF perturbations

a. Stacked barplot of distributions of clonal cluster fractions along human developmental stages in NT, NR2E1, ARX, SOX2, ZNF219 and NEUROD2 KD, colored by clonal clusters. Legend is shown in Fig. 4c.

b-d. UMAPs of human clones colored by (b) pseudotime, (c) fate biases and (d) individuals.

e. Ridge plots of pseudotime distribution in clones grouped by stages.

f. Boxplot to show median value of clonal pseudotime in each individual under TF perturbation, highlight opposing roles of ARX and NR2E1 in regulating maturation of cell fate potential. Paired Wilcoxon test was performed to test for significance.

g. Heatmap to show changes of marker gene expression along the clonal pseudotime.

h. Violin plots to show marker gene expression under ARX and NR2E1 perturbations.

GW, gestational week; sgRNA, single guide RNA; TF, transcription factor; FACS, fluorescence-activated cell sorting; NT, non-targeting; RG, radial glia; EN, excitatory neuron; IN, inhibitory neuron

*, $p < 0.05$; **, $p < 0.01$; ***, $p < 0.001$

When performing comparative studies in macaques, we have minimized the confounding effects through matching sgRNA, cryopreservation and cell culture conditions, as well as by matching developmental stages. We reported largely conserved phenotypes in cell type composition and correlated gene expression responses (Extended Data Fig. 6k-o). Against the background of conserved responses, we highlight the subset of divergent gene expression responses upon TF perturbation as a new approach to functionally dissect effector gene evolution in conserved GRNs. Importantly, the gene divergence was calculated in each cell type under multiple biological replicates (humans, $n=9$; macaques, $n=4$) and stages (humans, GW16-22; macaques, PCD60-80) in both species, therefore is not confounded by composition differences across stages.

In the revision, we mainly focused on conserved human and macaque phenotypes that illuminate mechanisms of cortical development, but we do note that comparative perturb-seq can also be applied to functionally dissect differences in regulatory networks across species.

7. gRNA Variability is another issue. Given the observed variability in gRNA effects, per-guide analyses would be beneficial. Do different guides targeting the same TF yield consistent phenotypes?

Consistent phenotypes between sgRNAs sharing the same gene target were observed at the level of cell type composition (Extended Data Fig. 4a, Extended Data Fig. 8d,f) and gene expression (**Review Figure 13**).

Review Figure 13 (Extended Data Fig. 3a): Consistency between sgRNA. Scatterplots of logFC of DEGs identified in each class in the sgRNA level (x axis) and gene level (y axis). Significant DEGs in the gene level were used for this comparison and red dots denote DEGs that were detected as significant in at least 2 conditions. Red dashed line represents $Y=X$.

Minor:

- The 44 TFs were selected based on their broad transcriptional impact but are not necessarily causal drivers of differentiation. Pseudotime trajectory analyses could provide insights into their effects on differentiation dynamics beyond discrete cell-type compositions.

We thank the reviewer for the suggestion and agree that finer resolution of changes will help uncover effects of TF perturbations. We applied Milo for cluster free differential abundance testing, which provided finer resolution

and does not rely on discrete cell type clustering (Fig. 3a, Extended Data Fig. 4). We also included pseudotime trajectory analysis for the clonal clusters which identified opposing effects of NR2E1 and ARX in modulating RG lineage progression (**Review Figure 12**).

- The cross-species comparisons in Fig. 4 are not entirely convincing. Some patterns appear conserved, while others do not. Statistical correlation analyses would help support the conservation hypothesis.

We thank the reviewer for the suggestion. Correlation analysis at the level of gene expression responses (Extended Data Fig. 6n) and cell type composition responses generally supports the conservation hypothesis (**Review Figure 14**). While we do observe some differences in the cell type level, we reasoned that it can be due to accelerated differentiation/maturation in macaques compared to humans over the course of differentiation in culture (as the absolute number of differentiation days was used for both species, despite controlling for sample starting age) and observed an overall consistency between the two species.

Review Figure 14 (Extended Data Fig. 6m): Correlation of compositional changes in humans and macaques. Scatter plot of estimated coefficients from DCATS in human (x axis) and macaque (y axis). Pearson correlation coefficient and p value were calculated in each TF perturbation and labeled together with regression line equations. Extended Data Fig. 6n shows similar results for gene expression responses (Pearson R = 0.5 to 0.58).

- Lines 933-935: What method was used for in silico perturbation, and was it validated against in-vitro or in-vivo datasets?

We applied the SCENIC+ workflow for in silico perturbation and have updated the Methods to clarify this point (line 1059). In the original publication, the authors used in vivo datasets in human and mouse brains and observed the predicted results consistent with previous results. While useful for prioritization, these prediction models are still being established and sensitive to the duration of dynamic processes like differentiation, and we therefore focused the study on experimental results.

Referee #2 (Remarks to the Author):

In the manuscript entitled Dissecting Gene Regulatory Networks Governing Human Cortical Cell Fate, the authors investigate the transcription factor (TF) circuits that regulate human cortical neurogenesis. Using single-cell CRISPRi screening in a human primary culture system, they examine the functional roles of 44 TFs in determining radial glia (RG) lineage progression. Their findings reveal key regulators such as NR2E1, ARX, and ZNF219, which influence the balance between excitatory and inhibitory neuron (EN/IN) differentiation.

They also identify effector genes implicated in neurodevelopmental disorders and demonstrate conserved mechanisms of RG lineage plasticity across primates. This study offers a framework for dissecting gene regulatory networks underlying human cortical cell fate decisions.

The manuscript presents a solid study with several strengths, including the use of valuable human fetal cells to investigate authentic gene regulatory networks controlling human cortical specification. The combination of cutting-edge technologies, such as Perturb-seq and lineage barcoding, provides important insights into the developmental origins and lineage plasticity of cortical cell types. Additionally, the cross-species comparison between human and macaque offers an interesting perspective on the conservation of lineage plasticity. However, the study lacks the level of novelty required for publication in Nature for the following reasons:

1. Limited technological innovation –While the Perturb-seq approach is a powerful tool, its application has already been demonstrated in various biological contexts. In this study, the reviewer does not see significant methodological advancements beyond what has already been published. It should also be noted that number of genes targeted (44) is relatively limited by today's standards. Without major technical improvements or novel adaptations, the study does not provide a sufficient technological breakthrough to justify publication in a high-impact journal.

We thank the reviewer for raising the important point that we need to demonstrate the technological innovation of our system, which we believe is qualitatively new, enabling analysis not possible in any existing system.

We did not sufficiently discuss and quantitatively benchmark the advantages of our system compared to the limitations of existing systems in our initial submission, nor did we discuss the methodological advances we made.

In brief: Organoids and induced neurons show patterning heterogeneity, limited fidelity to primary tissue, ectopic glycolysis pathway activation, endoplasmic reticulum stress and oxidative stress pathway upregulation. Induced neurons artificially short circuit normal developmental events, making it impossible to study cell cycle, physiological differentiation dynamics, or cell fate choice (while also showing variable patterning to peripheral nervous system glutamatergic neurons). Organoids show limited and variable maturation, while exhibiting necrotic cores, and frequently silence CRISPR machinery. No existing system allows restricting CRISPR-based gene restriction to RG to study lineage consequences. Neither induced neurons nor organoids have been scaled across multiple genetic backgrounds or applied to study temporal changes in cell fate potential.

We address all these limitations and now provide analyses to demonstrate the improvements (as well as the novel biological findings they enable). Reproduced from above:

The **technological novelty** of our system enables dissection of mechanisms underlying cell cycle dynamics and cell fate choice not possible with existing human cortical CRISPR systems by:

- **Increasing the fidelity to normal cortical development.** We have now performed extensive comparisons across methods highlighting the improved fidelity of our primary culture neurogenesis model revealing more precise cell type homologies (Review Figure 1A), reduced metabolic stress including glycolysis, ER and oxidative stress than iNeurons, iPSC- and primary- derived cortical organoids (**Review Figure 1B**).
- **Improving the reproducibility of cortical patterning.** We have now compared the primary samples that we use, which have been patterned for 16-22 weeks *in vivo* to organoids. We find that organoids, which endure long-term stress, show less than 50% cortical patterning in recent screens (PMID: 36198796; PMID: 39567792) or absence of mature EN populations(PMID: 38194967) (**Review Figure**

18, see below). Similarly, induced neurons, which forego normal patterning, show less than 50% CNS patterning with cells best resembling PNS neurons (PMID: 34358451) (**Review Figure 1C**).

- **Improving the scalability across individuals.** We have now expanded our screens to 21 biological replicates across 2 species and 3D organotypic slice culture model now included with our new experiments, demonstrating the scalability of our approach and potential to harness diverse genetic backgrounds and the fidelity of in vivo patterning (**Review Figure 1D**).
- **Enabling sensitive detection of cell fate consequence.** The week-long radial glia (RG) expansion phase that we implemented allows for CRISPRi activity to be restricted to the progenitor stage rather than manifesting in cells differentiated to various stages (**Review Figure 1E**). We have now performed additional validation experiments for PAX6 and EOMES, which have known cell fate consequences, as suggested, supporting the sensitivity of our system (**Review Figure 1F**). We further show, as suggested, that our 2D system further recapitulates gene expression and compositional consequences of TF knockdown in gold-standard human organotypic slice culture (**Figure 1G, J**), while also uniquely providing the RG expansion point for localizing CRISPRi gene repression to RG.
- **Increasing temporal precision.** Organoid RG show variable maturation stages in the same organoid (PMID: 30735633; 31996853), while induced neurons short circuit physiological RG altogether. In contrast, our system allows us to survey defined in vivo maturation stages and recapitulates a known sequence of temporal cell fate decisions at the individual clone level (**Review Figure 1H-I**), a feature we now harness to enhance the biological novelty of our findings (below).

2. Moderate biological impact – The findings are not sufficiently groundbreaking, as out of the 44 TFs analyzed, only ZNF219 is identified as a novel regulator of human corticogenesis. The roles of NR2E1 and ARX in cortical development have already been established, and while the authors provide additional details on lineage specification and inhibitory neuron subtype regulation, these findings do not represent a major conceptual leap in the field without additional follow up work on the mechanism of action.

We thank the review for raising this limitation with the initial submission, which motivated us to extend the biological novelty. Our initial manuscript was organized around the novelty of the screen design combining perturb-seq and lineage tracing in primary cortical tissue. We have now performed additional experiments to extend the biological findings and restructure the manuscript to include two figures devoted to important biological findings enabled by our system. Figure 4 addresses **TFs regulating developmental tempo** with conserved phenotypes across primates, and Figure 5 addresses **TFs safeguarding post-mitotic inhibitory neuron maturation**, now isolating a major mechanism of action of ectopic IN states mediated by ARX KD.

By performing new experiments during review (9 total samples spanning 6 weeks), we discovered that the effects of ARX and NR2E1 extend beyond balancing EN/IN neurogenesis to affect additional cell fate choices. Synthesizing these findings - enabled only by combining perturb-seq, lineage tracing, and human radial glia from defined in vivo maturation stages - suggests opposing roles for ARX and NR2E1 in regulating the developmental tempo of the human cortical neurogenesis (**Review Figure 12**). Identifying transcription factor regulators of developmental tempo contributes to an emerging field of inquiry in developmental biology in which regulators of chromatin state, metabolism, and protein turnover have recently been implicated. We further identified a critical temporal window for EN/IN plasticity of RG that can be engineered through TF perturbation, providing clinically relevant insights for disorders driven by E/I imbalance. Below we attach an updated Figure 4 focusing on the biological theme of conserved TF mechanisms regulating heterochrony of cell fate choice revealed by the study.

Review Figure 12 (Fig. 4, reproduced from above): Heterochronic RG lineage progression under TF perturbations

a. Stacked barplot of distributions of clonal cluster fractions along human developmental stages in NT, NR2E1, ARX, SOX2, ZNF219 and NEUROD2 KD, colored by clonal clusters. Legend is shown in Fig. 4c.

b-d. UMAPs of human clones colored by (b) pseudotime, (c) fate biases and (d) individuals.

e. Ridge plots of pseudotime distribution in clones grouped by stages.

f. Boxplot to show median value of clonal pseudotime in each individual under TF perturbation, highlight opposing roles of ARX and NR2E1 in regulating maturation of cell fate potential. Paired Wilcoxon test was performed to test for significance.

g. Heatmap to show changes of marker gene expression along the clonal pseudotime.

h. Violin plots to show marker gene expression under ARX and NR2E1 perturbations.

GW, gestational week; sgRNA, single guide RNA; TF, transcription factor; FACS, fluorescence-activated cell sorting; NT, non-targeting; RG, radial glia; EN, excitatory neuron; IN, inhibitory neuron

*, $p < 0.05$; **, $p < 0.01$; ***, $p < 0.001$

In Figure 5, we focus on **TFs safeguarding IN Identity**. Within the IN lineage, we discovered a conserved role of ARX in safeguarding normal post-mitotic IN identity. Loss of ARX has long been linked to impaired inhibitory neuron migration and epilepsy in ARX-linked lissencephaly. Our findings add a novel cellular mechanism to this literature by revealing that inhibitory neurons with reduced ARX adopt an ectopic cell state – conserved across species – and by providing a set of the earliest human dysregulated genes. During review, we have confirmed that the ectopic state is recapitulated in organotypic slice culture and provided additional support that the state is mediated post-mitotically.

To examine whether these stable and conserved clusters represent alternative *in vivo* cell types or ectopic states outside of normal morphospace, we mapped the INs in our dataset to existing mammalian brain atlas (PMID: 37824652). We found the cluster formed by repression of ARX had only moderate similarity to MGE derived INs (Extended Data Fig. 9p), while not sharing core regulatory genes such as LHX6. This finding supports the concept that removal of transcription factor identity guardrail can cause cells to differentiate into stable ectopic basins of attraction outside of normal morphospace and that these ectopic states can be conserved across species, despite not being adopted in normal development.

To identify mechanisms underlying ARX mediated IN state alterations, we performed dual repression of ARX and LMO1. We found that repression of LMO1 in ARX KD partially rescues the ectopic state by enriching an intermediate IN state and reversing ARX mediated gene upregulation (Review Figure 15f-i, Extended Data Fig. 10f-h). This finding adds to our knowledge about the molecular mechanisms mediating pathologies in neurodevelopmental disorders caused by mutations to ARX by highlighting a key target in a physiological model of human development.

Review Figure 15 (Fig. 5): ARX and LMO1 double KD partially rescues ARX-mediated ectopic IN state.

a. UMAP of integrated human and macaque IN clusters identified in 2D from Fig. 4c.

- b. Dotplot showing marker gene expression in each IN cluster (left), with a barplot showing numbers of cells detected in the cluster (right). The size of each dot denotes the fractions of cells in the group where the gene is expressed and the color denotes mean gene expression in the group.
- c. Stacked barplot of distributions of IN subclusters in each TF perturbation in human (top) and macaque (middle) 2D and in human slice culture (bottom), colored by IN subtypes.
- d. UMAPs for differential abundance testing in IN subtypes using Milo in ARX KD at the gene level in human (top) and macaque (bottom). NT cells were subset based on the number of perturbed cells to balance the cell number between conditions. (left) UMAP of subset NT and perturbed cells from D7, colored by perturbation condition and labeled by IN subcluster. (right) Neighborhood graph of differential abundance testing colored by \log_2FC .
- e. Heatmaps of \log_2FC in top DEGs in IN_ *LMO1/RIC3* cluster compared to other IN subtypes in human (left) and macaque (right). Dot/asterisks indicate significant changes identified from DEseq2.
- f. Schematics for gene expression regulation under NT, ARX KD and ARX, *LMO1* double KD (dKD).
- g. UMAPs for differential abundance testing in IN subtypes using Milo in NT, ARX KD and dKD. (left) UMAP of subset NT and perturbed cells colored by perturbation conditions. (right) Neighborhood graph of differential abundance testing colored by \log_2FC (FDR=0.05) under different contrasts: ARX KD versus NT; dKD versus ARX KD and dKD versus NT.
- h. Beeswarm plots showing differential abundance in neighbourhoods identified in Fig. 5g at the IN subtype level. Neighbourhoods that had significant changes between conditions (FDR=0.05) were colored by their \log_2FC .
- i. Dotplot showing gene expression changes in IN_ *LMO1/RIC3* marker under ARX, *LMO1* double KD.
- j. Graphical summary of findings: Human cortical RG gives rise to EN, IN and oligodendrocytes sequentially along developmental stages. NR2E1 KD promotes the RG lineage progression while ARX KD delays the transition and impairs normal IN subtype specification. ZNF219 KD promotes differentiation of both EN and IN with an overall preference to EN.

As an additional aspect of biological novelty, our experiments reveal that genes not previously characterized in cortical development, have strong transcriptional and cell type consequences when repressed in the RG lineage, including ZNF219 and PHF21A, providing additional mechanisms for inquiry in clinical cases of PHF21A haploinsufficiency. In Figure 2, we synthesized our results across transcription factors to advance a concept that neurodevelopmental and neuropsychiatric disorder-linked genes, including schizophrenia (SCZ) and major depression disorder(MDD), show increased connectivity in functionally defined gene regulatory networks.

We thank the reviewer for the constructive feedback and have sought to improve the manuscript by adding experiments and analysis to focus on biologically novel findings about heterochrony, cell fate potential, and safeguarding normal developmental morphospace.

3. Lack of internal controls – The study would benefit from including perturbations of well-characterized TFs known to regulate human cortical cell fate. This would serve as an internal benchmark to validate the experimental system and provide a clearer functional framework for interpreting the role and timing of the novel TFs being investigated.

We thank the reviewer for this suggestion and have expanded our use of internal benchmarks during revision. Initially, we included NEUROD2 as an internal control, even using the NEUROD2 sgRNAs to optimize our silencing and differentiation time course and confirm protein knockdown by flow cytometry prior to perturb-seq experiments to maximize sensitivity for detecting altered RG differentiation sensitivity. However, we did not use

the term positive control in the study, because the genes largely have not been studied with this developmental resolution during primary human cortical neurogenesis.

During revision, we performed an arrayed screen with a flow cytometry readout targeting PAX6 and EOMES (**Review Figure 16**), and observed depletion of EN lineage (EOMES+, NEUROD2+) with enrichment of SOX2+ RG and IN populations. These phenotypes are consistent with previous reports in other model systems (mouse and iPSC-based studies) suggesting EOMES and PAX6 are essential for EN differentiation (PMID: 28057268; 26304102). We have included these results in the updated Extended Fig. 1d, and agree that internal benchmarks using well studied genes provide additional confidence in the new phenotypes that we report.

Review Figure 16 (Extended Data Fig. 1d): Arrayed KD of previously characterized TFs EOMES and PAX6. Barplot of fold changes in fractions of SOX2, EOMES and NEUROD2 expressing populations obtained from flow cytometry in PAX6 and EOMES KD at the sgRNA level in comparison to NT on D7 from n=4 individuals x 2 sgRNAs per perturbation. Dots represent biological replicates from independent individuals and sgRNAs. Increase in SOX2 and decrease in EOMES and NEUROD2 populations were detected in both perturbations under paired two-sided Wilcoxon tests.

Minor comments include:

- ARX and SOX2 Perturbations (line 183-): The formation of new clusters (“ectopic”) following ARX and SOX2 knockdown is interesting, but the characterization of these clusters is limited. It would strengthen the paper to discuss whether these ectopic clusters have any known in vivo counterparts or if they represent entirely novel cell states. Additional marker analysis could help clarify this.

We agree - we investigated the similarity of these clusters to other cell types in the developing human cortex as well as developing macaque telencephalon using public atlas data from Wang 2025 Nature (PMID: 39779846) and Micali 2023 Science (PMID: 37824652). Here we attach results from reference mapping to in vivo datasets and marker expression. We found that while the ectopic clusters define unique transcriptomic states that is

distinguishable from other IN subtypes observed in NT controls, they still showed no clear correspondence to other in vivo IN subtypes identified in macaque telencephalon at both transcriptomically and marker expression level, instead retaining general CGE/LGE features while occupying distinct transcriptional states (**Review Figure 17**).

Specifically, we find only moderate similarity between IN_LMO1/RIC3 and MGE derived INs were observed, consistent with previous observations that Arx ablation in mice enriched for MGE INs at the expense of CGE INs (PMID: 38895467), but the ectopic cluster is missing expression canonical MGE markers LHX6, SST suggests that these populations do not have a clear physiological counterpart.

Review Figure 17: Molecular identities of ectopic clusters in ARX and SOX2 KD. Heatmap of results from reference mapping of IN clusters to in vivo atlas of the developing human cortex (Wang, top left) and developing macaque telencephalon (Micali, top right), with dot plot showing marker gene expression of known IN subtypes.

- Lineage Tracing Experiment: While the lineage tracing approach is technically impressive, its contribution to the overall conclusions of the paper feels limited. Consider elaborating on how this experiment adds new insights beyond what is already provided by the perturbation and transcriptional analyses.

We agree with the reviewer and this fits the constructive theme of these comments that we need to better illustrate the biological novelty of our study rather than just the technical achievements. As our figures were previously structured around technical techniques, we realize that biological insights were underdeveloped and harder to appreciate.

In our revisions, we have restructured and expanded the Figure 3 results to better frame how we used lineage tracing, beyond supporting the presence of mixed EN/IN clones in human and macaque cortical development, to more directly address the question of what developmental mechanisms underlie the observed EN/IN composition changes in our initial screen.

Lineage tracing allowed us to distinguish between possibilities that perturbations alter EN/IN composition by affecting RG producing EN-biased clones, mixed EN/IN clones, and/or IN-biased clones. We performed experiments across 4 additional individuals (bringing the total for this experiment to 8 individuals for clonal analysis spanning neurogenesis) and updated our analysis to more clearly answer this question. We find that NR2E1 repression acts mainly by shifting from EN-biased to IN-biased clones to increase IN composition,

while ARX repression acts by increasing the fraction of EN-biased clones and reducing mixed clones to increase EN composition. ZNF219 KD increased EN ratio through increasing mixed clones at the expense of IN-biased clones. This application of lineage tracing with perturb-seq illuminates the mechanism by which TF perturbations alter lineage plasticity.

During the revision, we increased the depth of our analysis, leveraging an increased sample size at defined maturation stages, to further apply, lineage tracing to reconstruct the pseudotime of radial glia cell fate choices during cortical development, confirming sequential changes in cell fate potential with increased age and revealing the transcriptomic profile of the “tri-IPC” resembles that of mouse suggesting re-use of mouse cortical olfactory bulb progenitor programs.

Coupling this pseudotime analysis with perturbations allowed us to extend beyond the EN/IN fate bias observations and to examine whether distinct TF perturbations altered developmental tempo of human radial glia cell fate choices. This analysis further illuminates NR2E1 and ARX mechanisms by supporting opposing roles in developmental tempo across multiple cell fate choices spanning excitatory, inhibitory neurogenesis and gliogenesis (Review Figure 12).

- Gestational Week (GW) Sample Variation: The inclusion of samples across a broad range of gestational weeks provides developmental context, but the potential impact of this variation on the results isn't fully addressed. It would be helpful to discuss whether differences in cell composition or gene expression could be influenced by developmental stage, and how this might affect the interpretation of TF perturbation outcomes.

We agree with the reviewer that variation along developmental stages can impact effects of TF perturbation. As mentioned above, by increasing our sample size during review and harnessing the reproducibility of in vivo RG maturation, we have now turned this scalability across ages into an advantage to expand our biological findings.

Indeed, when we extended the targeted screen to late mid-gestation (GW22), changes in clone composition and perturbation effects were noted. During review, we have collected 4 more individuals spanning early to late mid-gestation, making a total of 9 individuals (GW16, n=1; GW17, n=2; GW18, n=1; GW19, n=2; GW21, n=2, GW22, n=1) to test the stage-dependent effects of top TF candidates on RG lineage plasticity.

We observe consistent phenotypes with data presented in the manuscript. Moreover, sampling densely from stages spanning excitatory, inhibitory neurogenesis and oligodendrogenesis further revealed opposing roles of NR2E1 and ARX KD on the developmental tempo of cortical RG, impacting gliogenesis beyond EN/IN balance.

Review Figure 12 (Fig. 4, reproduced from above): Heterochronic RG lineage progression under TF perturbations

a. Stacked barplot of distributions of clonal cluster fractions along human developmental stages in NT, NR2E1, ARX, SOX2, ZNF219 and NEUROD2 KD, colored by clonal clusters. Legend is shown in Fig. 4c.

b-d. UMAPs of human clones colored by (b) pseudotime, (c) fate biases and (d) individuals.

e. Ridge plots of pseudotime distribution in clones grouped by stages.

f. Boxplot to show median value of clonal pseudotime in each individual under TF perturbation, highlight opposing roles of ARX and NR2E1 in regulating maturation of cell fate potential. Paired Wilcoxon test was performed to test for significance.

g. Heatmap to show changes of marker gene expression along the clonal pseudotime.

h. Violin plots to show marker gene expression under ARX and NR2E1 perturbations.

GW, gestational week; sgRNA, single guide RNA; TF, transcription factor; FACS, fluorescence-activated cell sorting; NT, non-targeting; RG, radial glia; EN, excitatory neuron; IN, inhibitory neuron

*, $p < 0.05$; **, $p < 0.01$; ***, $p < 0.001$

Overall, while the study is rigorous and contributes valuable data to the field, its lack of significant novelty limits its suitability for Nature journal.

We thank the reviewer for their comments and agree that the organization of the initial submission around techniques rather than insights limited the novelty. We hope that by extending the biological findings to TFs regulation heterochrony and to TFs safeguarding post-mitotic identity through new experiments and analysis that we have been able to satisfy these concerns.

Referee #3 (Remarks to the Author):

In humans, most cortical interneurons (INs) originate from the ganglionic eminences in the ventral forebrain and migrate to the cortex. However, recent work has identified a small but significant population of locally generated interneurons that arise in the dorsal telencephalon alongside excitatory neurons (ENs). In this study, Ding and colleagues show that these interneurons are generated from the same progenitors that give rise to excitatory neurons in the developing cortex. This suggests a broader neurogenic potential of radial glial cells (RGs) than previously thought. The authors here screened 44 transcription factors to determine the molecular program to generate ENs or INs and found that NR2E1, ARX, SOX2, and ZNF219 play an important role. The presence of cortical-generated INs has been demonstrated in humans, but here they show that this can be extended to other primates. Overall, this study improves our understanding of the regulatory networks that control cell fate decisions during human neurogenesis and suggests that this may also play an important role in neurodevelopmental processes and disorders.

We thank the reviewer for these kind words and thoughtful comments.

Here are my suggestions to strengthen this study:

Selection of transcription factors: I understand how the TFs were initially selected, but I'm a bit puzzled because some very well-known, important, TFs are missing, e.g. PAX6, GLI3, HOPX, EOMES. On the contrary, some only neuronal TFs (TBR1, SATB2) are included while the fate of RGs is investigated.

We agree that including additional well known TFs as anchors, if not positive controls, would be helpful to contextualize results with previous studies. Our initial prioritization approach was data-driven where TFs predicted to drive the strongest transcriptional changes along the differentiation trajectory were selected. However, as prediction methods are in their infancy, we agree with the reviewer that our system will benefit from including well known marker TF.

We have now included results from EOMES and PAX6 KD in revision in Extended Data Fig. 1d (**Review Figure 16**). Arrayed screening with a flow cytometry readout quantifying abundance of SOX2, EOMES and NEUROD2 positive cells showed effects of both EOMES and PAX6 KD on decreasing EN differentiation and enriching for SOX2+ RG and possibly INs. These phenotypes are consistent with previous reports in other model systems (mouse and iPSC-based studies)(PMID: 28057268; 26304102).

Review Figure 16 (Extended Data Fig. 1d, reproduced from above): Arrayed KD of previously characterized TFs EOMES and PAX6. Barplot of fold changes in fractions of SOX2, EOMES and NEUROD2 expressing populations obtained from flow cytometry in PAX6 and EOMES KD at the sgRNA level in comparison to NT on D7 from n=4 individuals x 2 sgRNAs per perturbation. Dots represent biological replicates from independent individuals and sgRNAs. Increase in SOX2 and decrease in EOMES and NEUROD2 populations were detected in both perturbations under paired two-sided Wilcoxon tests.

Model system: I'm slightly surprised by the 2D approach when we now have many ways to look at these events in 3D. For example, why not use human fetal slices directly? Alternatively, the authors could use the fetal brain organoid model (FeBO) recently developed by the Artegiani lab. From their data, it appears that identity and developmental stages are maintained when FeBOs are grown in 3D.

We agree that the 3D organotypic slice may represent a more physiological cultural system. We have now performed a targeted screen paired with lineage tracing in 3D organotypic slice culture. We observed consistent composition changes and correlated gene expression changes upon TF perturbations in 2D and 3D (Review Figure 8d-e).

Meanwhile, we would like to note limitations in alternative 3D systems that motivated our decision to establish a 2D system.

- **Limited scalability across individuals in 3D models:** Organotypic slices cannot be cryopreserved and slice and FeBO models cannot be cultured in pools to scale across individuals while mitigating batch effects, in contrast to cryopreserved chunks followed by batch controlled 2D cultures.
- **Limited capability for cell fate engineering in 3D models:** Even if CRISPRi lentivirus infection is localized to radial glia, differentiation can still occur prior to CRISPR activity resulting in heterogeneous starter populations and reducing sensitivity for detecting cell fate consequences in RG lineages. For example, Cas9 activity from viral libraries typically takes ~7 days in human RG, while differentiation can occur after one cell cycle (24-48 hrs). In contrast, the 2D culture system allows us to expand cells as RG to ensure gene repression occurs in a homogenous starter population. RG from our system could

then be transplanted to 3D systems, including in vivo mouse, to further harness 3D in vivo environments, building on our system.

Additionally, we note that the FeBO dataset is characterized by a significant increase in cellular stress response, including aggregate scores for glycolysis, ER stress, reactive oxygen species, and apoptosis on par with organoids (Review Figure 7) and a lack of EN populations (Review Figure 18), therefore limiting its capability to study the EN lineage and supporting the increased physiological relevance of the screening system we introduce for studying neurogenesis.

Review Figure 8 (Extended Data Fig. 9a-g, reproduced from above): Organotypic cortical slice culture and molecular identities of different IN subtypes.

- Experimental design for targeted TF perturbation and lineage tracing in organotypic slice culture of 4 human individuals. CRISPRi and STICR lentivirus was locally co-injected to the germinal zone to enrich for RG populations.
- UMAPs showing cell type annotations and individuals (top right), with a UMAP of new born progenies after filtering for cells from multicellular clones on the bottom right. Note that large proportions of postmitotic ENs were labeled and captured but not clonally linked to other cell types.
- Dotplot of the expression of cell type markers for the assigned clusters (left), with a barplot showing numbers of cells detected in the cluster (right). The size of each dot denotes the fractions of cells in the group where the gene is expressed and the color denotes mean gene expression in the group.
- Scatterplot showing log₂FC in human 2D and slice for DEGs identified in ARX, NR2E1, SOX2 and ZNF219 KD. Pearson correlation coefficients and p values were calculated in each TF perturbation and labeled together with regression line equations. Red dashed line represents Y=X. An average Pearson correlation R = 0.6 was observed across perturbations.
- Barplot showing cell class compositions of cells in multicellular clones in each perturbation.

- m. Heatmap for lineage coupling score matrix in major cell types in the RG lineage. IN_local represents locally derived INs in culture that shares lineage with RG and ENs.
- n. Dot plot showing markers for different IN subtypes.

Review Figure 7 (reproduced from above): Comparison of metabolic stress in human model systems.

Violin plots of gene expression scores of 5 types of cell stress (glycolysis, ER stress, reactive oxygen species (ROS), apoptosis) in 6 different datasets including vivo human atlas, primary organotypic slice culture, 2D primary RG culture, iNeurons(Lin et al. 2021), primary brain organoids FeBO (Hendriks et al. 2024) and organoids(He et al. 2024) obtained from published studies.

[REDACTION]

Review Figure 18: Limited EN populations in FeBO.

Top (reproduced from Hendriks et al., 2024, PMID:38194967, Figure 3H,I): Predominance of HES1, HOPX and MKI67 positive RG populations (cluster1-13) in dorsal FeBO.

Bottom: Barplots showing cell type compositions in different conditions in FeBO (PMID: 38194967). (left to right) expansion media 3 months; maturation media 3 months; expansion media 6 months; maturation media 6 months. Data was reference mapped to the Wang et al., 2025 Nature (PMID: 39779846) in vivo human atlas for cell type annotation. While emergence of dividing progenitors and newborn excitatory and inhibitory IPCs were observed in maturation media, overrepresentation of astrocytes and absence of postmitotic EN in all conditions were noted.

In addition, the authors should provide evidence for the precise identity of these cells in 2D (I find it hard to find these data, day0 cells are very few in the single cell RNA-seq in Fig. 1B). What is the exact identity of these cells? How can they specifically show that they are only cortical and that there are no ventral RGs?

The identities of D0 cells are available in the barplot Fig1c, we apologize that it was small in the initial manuscript and have updated and added additional UMAPs and dotplots to highlight populations from different timepoints and their transcriptional identities (Review Figure 19). All the updates can be found in the revised version of the manuscript (Fig. 1c,d and Extended Data Fig. 1f). In addition to reference mapping, the

expression of cortical markers restricted from ventral RG such as GLI3 and EMX2 further supports the cortical identity.

Review Figure 19: Molecular identities of differentiation D0 cells. (Left) UMAPs and barplot showing cell type compositions in D0 and D7. (Right) Dotplot showing marker gene expression.

95.3% (94.3% in GW16 and 96.4% in GW18) of 21151 cells collected at D0 were assigned to cluster radial glia class (RG_DIV and RG/Astro) (Review Figure 19), a small population expressing markers for OPCs (OLIG2, BCAS1, < 5%). OPCs are fate restricted progenitors that do not give rise to neurons, therefore their presence does not account for the EN/IN phenotypes we discuss in this study.

As discussed below, quantification of marker expression at the protein level through flow cytometry (Review Figure 20, see below) further validated homogeneity of RG populations on D0 with less than 5% populations expressing lineage specific markers, including DLX2 which marks ventral RG.

Timing: It would be nice to have a rationale for selecting the fetal time period that the authors decided to start with. Is this the time when the INs are generated? Why are so many INs generated at the end? Shouldn't they be a tiny fraction of the general neuronal population?

We agree with the reviewer that we should emphasize on our rationale for experimental design as part of expanding on our biological findings. We chose to focus on mid-gestation, starting with GW16-18 in the initial screen, because these stages represent the peak of cortical neurogenesis. Further analysis revealed that by including later timepoints, we could follow sequential changes in cell fate potential giving us resolution to explore regulatory mechanisms controlling the timing of these temporal changes. As primary cells enable comparing cells with well defined maturation stages, we expanded the temporal sampling in revision to assess the effects on the maturation of radial glia cell fate choices.

In terms of IN production at later stages of human cortical development, a similar ratio of IN was reported in Delgado et al., Nature 2022 (PMID: 34912114) (~20%) and in this study (~17%). We also would like to note that changes in cell proportions at the baseline level do not impact our conclusions on the roles of TF perturbations when compared with NT controls in the same system. On the other hand, equally represented EN/IN populations allowed us to sensitively detect compositional changes under TF perturbation.

Virus selection: Lentiviruses infect all cells, including differentiated cells. The authors use a protocol enriching for progenitors, but this does not exclude the presence of neurons. Therefore, it would be important to show the % of cells in the progenitor state, and the % of differentiated cells (ENs and INs) with some IHC. In

addition, the use of a retrovirus could be a better choice, because it excludes infection of postmitotic cells since it only infects dividing cells.

We agree with the reviewer that it is important to ensure the homogeneity of the starter populations. As reasoned above, single cell analysis revealed the predominance of radial glia cells on D0.

To further address this point, we also quantified the cell marker positive populations at the protein level. Using flow cytometry, we quantified SOX2, EOMES, NEUROD2 and DLX2 positive populations in NT control cells throughout the course of differentiation. We observed less than 5% populations expressing lineage specific markers EOMES, NEUROD2 or DLX2 on D0 (before differentiation) (Review Figure 20).

Review Figure 20 (Extended Data Fig. 1b): Marker quantification of differentiation D0 cells at the protein level. Barplot of SOX2, KI67, EOMES and DLX2 positive populations across the differentiation time course (D0, D2, D4, D7) in cells expressing all-in-one CRISPRi vectors with NT sgRNAs. Quantification was done through flow cytometry. Dots represent biological replicates from independent individuals and sgRNAs. D0, n=2; D2, n=3 x 2 sgRNAs; D4, n=3 x 2 sgRNAs; D7, n= 6 x 2 sgRNAs.

At D0, the authors use HOPX and KI67 to define cycling RGs. why HOPX? why not PAX6? PAX6 should define RGs and HOPX bRGs according to previous papers by the same authors. Some cell clusters are not clear to me. Why do IPCs express TBR1? Why is PAX6 in INs? Some cells are not FOXG1, what are they?

Cell type annotation is defined based on marker gene expression, differential gene expression analysis between clusters and reference mapping to the *in vivo* atlas in Wang et al., 2025 (PMID: 39779846). HOPX and MKI67 expression were shown for visualization but were not the only criteria that we used to define cycling RG. We agree with the reviewer that specific RG subtype, such as vRG, can be HOPX negative, we apologize for not making that clear and have attached a dotplot with additional marker panels to support our cell type annotations (Review Figure 21).

We did not use PAX6 to define RG because while PAX6 is highly expressed in RG, it is also expressed in LGE-derived INs, which has been reported in multiple studies including Schmitz et al 2022 Nature (PMID: 35322231), and Keefe, Steyert & Nowakowski 2025 Nature, where DLX2 and SCGN cells coexpressing PAX6 found *in vivo* in midgestation cortex (Review Figure 22).

TBR1 is expressed in excitatory IPCs at the transcript level based on *in vivo* atlas in Wang 2025 Nature (PMID: 39779846) (Review Figure 23).

Most cell types in our culture are FOXG1 positive, while OPCs and oligodendrocytes have lower expression, and as expected, FOXG1 is absent from vascular cells and microglia (Review Figure 21).

**Review Figure 21: Cell type annotation of D7 cell types.** Dotplot showing cell type marker expression in each cell type. Additional RG (VIM, GLIS3, SLC1A3), EN_IPC (PPP1R17, SSTR2) and IN markers (DLX2, GAD1) were included to support cell type annotations.

**Review Figure 22: PAX6 expression in IN populations (Keefe, Steyert, and Nowakowski, Nature 2025, in press).** Immunocytochemical labeling of PAX6 with DLX2 at outer subventricular zone (oSVZ) and SCGN at cortical plate (CP).

Review Figure 23: TBR1 expression in Wang 2025 in vivo atlas. Dotplot showing excitatory IPC markers and TBR1 expression in each cell type annotated in Wang 2025 Nature (PMID: 39779846).

The authors nicely link the TFs to neurodevelopmental disorders but have not explored this more in-depth. For example, recent work has focused attention on ASD and hypothesized that E/I imbalance is critical. It would be nice to test this hypothesis in a disease context using dorsally patterned cerebral organoids derived from ASD patients.

We agree with the reviewer that the role of E/I imbalance in ASD is an intriguing hypothesis to examine in human cellular models. In fact, candidates for which we observed strong transcriptional and cell type composition phenotypes (ARX, CTCF, PHF21A) have been identified as causal mutations in ASD. However, since this study focuses on the effects of TF perturbation on normal cortical development, applying patient-derived organoids are beyond the scope of current study. Nevertheless, we have added a new Figure 5, in which we use a dual sgRNA approach to study molecular mechanisms mediating ectopic IN subtype specification in the context of ARX loss of function. We also highlight several important advantages of the primary cell system (e.g., **Review Figures 3-7**), including cellular health enabling more physiological analysis of neurogenesis, defined stages of maturation enabling studies of heterochrony, and scalability across individuals enabling studies in low and high polygenic risk score backgrounds. Together, these advantages will support studying ASD-linked genes in primary cell models as a complement to organoid tools, a goal that we are pursuing as the resource of cryopreserved individuals expands to a population scale.

The authors show that more than 25% DEGs are common for more than 1 TFs. This is very intriguing and could be extended a bit with GO and validated in some way, e.g. IHC...

We agree - the finding of distinct properties for convergent TFs highlights adds to our knowledge of how neurodevelopmental gene networks are organized and linked to disease. We have sought to further investigate and validate the phenomenon of convergent DEGs and now organize Figure 2 around this phenomenon. We have included GO analysis in Fig. 2d and new analysis investigating the association of DEGs between disease related genes and the number of convergence in TF perturbations (Fig. 2c,f). Fig. 2f in particular shows that the significance of the neuropsychiatric enrichments increases with increased convergence, adding additional support to the interpretation. We attempted to validate convergent DEGs downstream of multiple TFs through IHC but could not find validated antibodies. Instead, we have added validation that DEGs detected in 2D also emerge in organotypic slice culture (**Review Figure 8d**), supporting the generalization of these findings.

The perturbation results are interesting as well as unexpected as it seems that so many of these TFs regulate the fate of INs. Interesting because in the cortex these are only a very tiny fraction and therefore it is perplexing that this population is so highly orchestrated by so many TFs. However, since the authors focus on ARX, NR2E1, and ZNF219, they could expand and show the expression of these TFs at the protein level. IHC and double staining with some of the known TFs (PAX6, HOPX, DLX....) would strengthen these findings. KD of ARX results in IN with GE identity.

We thank the reviewer for the thoughtful comment. We agree that confirming the expression of our top candidates at the protein level together with other cell type markers would strengthen our conclusion. We have now included IHC results for ARX (**Review Figure 24**, Extended Data Fig. 10m-p), using an antibody validated by our KD experiment (Extended Data. Fig1c).

Review Figure 24: ARX antibody validation and IHC.

(Left) Barplot of fold changes in fractions of ARX positive populations compared to NT in each sgRNA to confirm efficient KD in ARX KD and validate the ARX antibody, from n=4 human individuals.

(Right) (top) Immunocytochemical labeling of ARX with HOPX and KI67 at subventricular zone. (bottom) Immunocytochemical labeling of ARX with DLX2 and SCGN at cortical plate, scale bar: 50 μ m.

The additional experiments in review might also help address the observation that the EN/IN ratio could be influenced by multiple TFs even though the overall cortical born IN population is low. We find that, at least for NR2E1 and ARX, the influence on IN fate is part of a larger role for these TFs in shifting the developmental timing of cell fate potential with broader impacts on the extent of excitatory neurogenesis and onset of gliogenesis (Figure 4). The switch to IN production in the neurogenesis window that we surveyed served as a conspicuous readout reflecting this broader effect on maturation.

The authors may want to follow up on this point and compare the ARX INs with LMO GE-derived INs. Perhaps KD of these (LMO, RIC3) genes would allow to define their function in this subtype.

In the new data that we collected in organotypic slice culture, we have observed consistent phenotypes in ectopic IN clusters in ARX (LMO1/RIC3) and SOX2 (CALB2/SOX2) KD, suggesting in vivo relevance of our findings in 2D systems (Review Figure 8c).

We thank the reviewer for the suggestion to perform LMO1 and RIC3 KD - this experiment was really powerful for illuminating the molecular mechanism through which ARX acts in INs. Of these two candidates, we focused on LMO1 because it is a transcriptional coactivator that we predicted could have many downstream targets while RIC3 is an acetylcholine receptor chaperone. Because these genes are only induced upon ARX knockdown in post-mitotic interneurons (which represent a subset of cells infected with the ARX sgRNA), we performed dual repression of ARX and LMO1, and we investigated the effects of LMO1 in mediating ectopic IN cluster driven by ARX KD. In brief, we found that ARX, LMO1 double KD partially rescued the ectopic IN phenotype by reverting ARX-mediated transcriptional changes including RIC3, IQCH and EFNA5. We have now added a new Figure 5 focusing on TF safeguarding IN identity (Review Figure 15) and we are grateful for the reviewers suggestion.

Review Figure 15 (Fig. 5): ARX and LMO1 double KD partially rescues ARX-mediated ectopic IN state.

f. Schematics for gene expression regulation under NT, ARX KD and ARX, LMO1 double KD (dKD).

g. UMAPs for differential abundance testing in IN subtypes using Milo in NT, ARX KD and dKD. (left) UMAP of subsetted NT and perturbed cells colored by perturbation conditions. (right) Neighborhood graph of differential abundance testing colored by log₂FC (FDR=0.05) under different contrasts: ARX KD versus NT; dKD versus ARX KD and dKD versus NT.

h. Beeswarm plots showing differential abundance in neighbourhoods identified in Fig. 5g at the IN subtype level. Neighbourhoods that had significant changes between conditions (FDR=0.05) were colored by their log₂FC.

i. Dotplot showing gene expression changes in IN_LMO1/RIC3 marker under ARX, LMO1 double KD.

j. Graphical summary of findings: Human cortical RG gives rise to EN, IN and oligodendrocytes sequentially along developmental stages. NR2E1 KD promotes the RG lineage progression while ARX KD delays the transition and impairs normal IN subtype specification. ZNF219 KD promotes differentiation of both EN and IN with an overall preference to EN.

Lineage: the same progenitors can give rise to excitatory and inhibitory neurons in the developing cortex. This is very intriguing and the authors should confirm these results with live imaging. Ideally, this should be performed in a 3D human slice by infecting a RG and following up divisions and fate. How can the authors be sure they are infecting a single cell and not multiple cells with the same virus? how many cells are infected with more than 1 viral particle (different virus)? Can they check after a few hours?

We agree with the reviewer that it is important to more extensively assess the shared origins of dorsal INs and ENs. Detailed investigation on shared origins of cortical ENs and INs can also be found in Keefe, Steyert & Nowakowski 2025, now accepted in Nature, where 3D human culture coupled with lineage tracing, as well as in vivo IHC validation (**Review Figure 22**) were performed across midgestation to show presence of cortical derived inhibitory neurogenesis (IN_{local}).

Similarly, we have now performed combined lineage tracing and TF perturbation in the organotypic slice culture system and observed a similar population (IN_{local}), where lineage coupling with RG and EN was observed (**Review Figure 8f**), supporting shared lineage between cortical ENs and INs.

The STICR library includes sufficient barcode diversity to rule out infecting multiple cells with the same barcode. Delgado et al. 2021 Nature (PMID: 34912114) calculates that the viral library could be used to label more than 250,000 cells before reaching an estimated barcode collision rate of around 0.5%. Each batch in this study captures an average of 32,917 cells resulting from 7155 mother cells, which results in 0.0143% collision rate. We therefore reasoned it is unlikely that barcode clashing has biased our conclusions. Furthermore, combinatorial labeling with sgRNAs further increased library diversity and clones including cells with conflicted sgRNA assignments were removed from downstream analyses.

In terms of infecting the same cell with multiple barcodes, 66.7% cells in this study were detected to have single clonal barcodes. Presence of multi-infected cells will not affect our conclusions due to the following reasons:

- Cells encoding more than one barcode will be recognized and assigned to the same clone with other cells with the same expression patterns. Double labeling, though rare, would add stringency to clonal assignments.
- Clones that have cells with conflicted sgRNA assignments were removed from downstream analyses.

Clonal size is also an issue without live imaging: how can the authors control for clone size? Some cell types could be more or less sensitive to isolation and the clones could be larger.

We agree with the reviewer that we cannot speak for clone size in vivo due to possibly disproportionate cell loss during dissociation and single cell capture. We were very careful throughout the manuscript that we don't draw any conclusions on the proliferative capacity or actual clone sizes of control RG but only compare topology of fate biases as a relative measure between control and perturbations that were processed and captured in a pooled manner. We agree that the absolute clone sizes on differentiation D7 are most likely larger than what we observe from the single cell data, but cell loss does not account for changes in clonal fate decisions that we observe under TF KD with equivalent pooled screening and processing conditions.

Evolution: considering the proportion of EN/IN cortex-derived neurons, isn't 47% unexpectedly high as a common origin of ENs and INs in macaques?

We initially presented this percentage to support the presence of mixed EN/IN clones in our dataset as this phenomenon had not yet been established in rhesus macaque, but we appreciate that the numerator and denominator are confusing. We observed that 47% of EN-containing multi-class (as opposed to EN restricted

clones) also contained INs. This number is similar to what we observe in humans. For example, at comparable stages in humans (GW21-22), 48% EN-containing multi-class clones also contained INs. (Review Figure 25). As those percentages were restricted to multi-class EN clones, we have now presented the simpler overall percentages in the updated text to avoid confusion, “The most abundant multi-class clones contained RG and ENs (35%) and RG and INs (20%), but, consistent with recent studies, we also observed around 15% clones containing both ENs and INs (Fig. 3d)” (Line 228-230). The percentage of IN containing clones may not be unexpectedly high as this late stage of neurogenesis follows >2 months of predominantly excitatory neurogenesis in humans, but represents an exciting window for studying developmental cell fate transitions related to radial glia maturation and also includes the emergence of tri-IPCs and onset of gliogenesis.

Review Figure 25: Clonal composition in humans and macaques (Extended Data Fig. 7e,f,h). (e-f) Clonal coupling between cell types and upset plot of cell class compositions in multi-class clones in macaque. (h) Upset plot of cell class compositions in multi-class clones and clonal coupling between cell types in early (top) and late (bottom) mid gestational human samples.

The human-macaque-specific responses to TFs (ARX and ZNF219) are very interesting. It would be nice to functionally prove the function of these TFs in evolution.

The responses to ARX and ZNF219 are largely conserved between humans and macaques in both cell type composition (Extended Data Fig. 6k,m) and gene expression (Extended data Fig. 6n). Against this background of conservation, we note that a subset of gene expression responses are also lineage-specific, indicating that our approach can be applied to study the evolution of membership in conserved GRNs. Future studies incorporating comparable multiomic data in rhesus macaque could assess whether altered enhancers containing motifs for the upstream TFs in only one species mediate the subset species-specific responses. With our revisions, we focus on the properties of connected genes in developmental gene networks, the lineage mechanisms underlying EN/IN cell fate differences, effects of TFs on developmental tempo and RG cell fate potential, and mechanisms by which ARX safeguards subtype specification. As our group is very

interested in the evolution of developmental processes, we hope to employ comparative loss of function profiling in the future to systematically study cis-trans interactions in gene network evolution.

Point-by-point:

Referee #1 (Remarks to the Author):

I appreciate the authors for their sincere responses to our questions. In the revised manuscript, the authors compared their system with other human models, strengthening their claim of higher level of fidelity and cell health in their system. Additionally, the authors performed further statistical analysis on the data, including the comparisons to DESeq2 results, phenotypic reproducibility across replicates and downsampling experiments that demonstrated conserved patterns in the data.

We thank the reviewer for their kind words and for motivating these additional analyses.

The novelty of their culture system and the biological discovery, yet, remain limited. It is already well established that neural progenitor cells (NPCs) can be derived from ES or iPSCs, a routine practice to form 3D organoids (Li et al., *Development*, 2009; Duan et al., *Stem Cells Translational Medicine*, 2015; Gunhanlar et al., *Molecular Psychiatry*, 2017; Scholz et al., *Frontiers in Molecular Biosciences*, 2022). NPCs derived from different ESC lines are also known to exhibit radial glia signatures as well as the heterogeneity of radial glia – so the novelty point of this culture system is likely overstated (Duan et al., *Stem Cells Translational Medicine*, 2015; Luciani et al., *Nature Communications*, 2024). By inducing the CRISPR editing system at either the ESC/iPSC/NPCs, it is already possible to investigate gene functions during early developmental stages, such as the radial glia.

We agree that iPSCs and organoids are an important model system, particularly for early stages of neurodevelopment. However, the primary culture system addresses unmet needs related to limitations of organoid models (PMID: 31996853; 32888425). Organoids present limited fidelity to in vivo cell types, display variable patterning and heterogeneous progenitor maturation, exhibit elevated cellular stress and mature slowly, with limited gliogenesis for 120 days, and pattern variably across individuals (e.g., PMID: 30735633; 32521257; 40864552). In addition, iPSC culture enriches for somatic mutations, including in tumor suppress and proliferation genes, and epigenetic changes with non-genetic effects controlling over half the variation in gene expression in cell culture (PMID: 28445466; 35176222; 28489815; 28388430; 29208628). These limitations make it challenging to reproducibly study features such as cell fate changes during radial glia maturation. Our benchmarking indicates improvements in fidelity, reproducibility, maturation, and multiple metrics of cell stress in both the 2D and 3D culture approaches that we present (Extended Data Fig. 1i, k, n). In addition, our findings leverage the ability to more precisely assign primary cell maturation stages (with a chronological ground-truth) and access to late stages of neurogenesis including the switch to gliogenesis (variable

and cumbersome to reach in organoids) as well as the reproducibility across individuals to identify genes that shift maturation stage (as assessed molecularly and by lineage relationships) as well as additional genes influencing cortical development. Finally, experiments in primary culture are closer to a ground-truth and may help to support improvements in organoid models, underscoring the additional value of introducing a primary cell CRISPR screening system.

Furthermore, the roles of NR2E1 and ARX in neurodevelopment have been extensively characterized in the last two decades (Roy et al. *The Journal of Neuroscience*, 2004; Friocourt et al., *The Journal of Neuroscience*, 2008; Colasante et al., *Cerebral Cortex*, 2013; Joseph et al., *iScience*, 2021; Kandel et al., *PNAS*, 2022); and LMO1 has been shown to be directly repressed by ARX in interneuron development (Fulp et al., *Human Molecular Genetics*, 2008).

We thank the reviewer for pointing out existing studies on roles of NR2E1 and ARX in neurodevelopment. Our study reveals phenotypes that are consistent with previous observations, but also extends roles of NR2E1 and ARX in regulating RG maturation and cortical developmental tempo, affecting multiple cortical derived lineages including ENs, INs and oligodendrocytes. This is beyond the effects on progenitor proliferation or differentiation that have been characterized before.

Moreover, the ectopic IN cell type induced by ARX has also not been reported or characterized before. Previous studies have reported upregulation of *Lmo1* and migration defects, but it was unclear if these were normal cells improperly migrating or cells with ectopic transcriptional states, and these effects had been determined in mouse and MGE, but not in the human and developing cortex. We believe our findings of the ectopic transcriptional state forming an entirely different transcriptional type outside of cell atlases and of the impacts in human and cortex add novelty on top of the existing studies. Further, by integrating lineage tracing with ARX perturbation, we show that ARX KD post-mitotically induced ectopic IN subtype characterized by LMO1 and RIC3 expression, a phenotype independent of its roles in balancing EN/IN subtype and regulating RG maturation.

Besides the cellular phenotypes, molecular mechanisms downstream of ARX are also largely elusive. While numerous studies (PMID: 18799476, 22252899, 24122442) including Fulp et al., 2008 have shown that LMO1 expression is directly repressed by ARX, it remains to be tested whether LMO1 upregulation mediates the cellular phenotypes caused by ARX KD. By performing dual repression of ARX and LMO1, we show that LMO1 repression partially rescued the cortical derived ectopic IN clusters induced through ARX KD, directly linking LMO1 expression to functional cellular phenotypes downstream of ARX.

Referee #2 (Remarks to the Author):

The authors have addressed my comments and concerns. In addition to reinforcing the technological novelty of their CRISPRi screening platform, they have now significantly enriched the manuscript with compelling biological findings. These additions greatly enhance the depth and relevance of the study, transitioning it from a primarily methodological contribution to one that also provides meaningful insights into human neurogenesis.

The integration of functional perturbations, lineage tracing, and cross-species validation adds considerable strength to the biological conclusions. I find that the manuscript now offers a well-rounded and impactful contribution, combining technical innovation with mechanistic discovery.

Referee #3 (Remarks to the Author):

The authors carefully addressed all the points I raised in my initial assessment.